# A Survey of Linear Attention: Algorithm, Theory, Application, and Infrastructure

### Abstract

Large Language Models (LLMs) have proven effective in understanding and generating extremely long contexts. Recently, linear attention mechanisms have garnered significant attention, as they can largely reduce the quadratic computational complexity of traditional attention mechanisms to linear complexity relative to token sequence length, thus balancing effectiveness and efficiency in LLM training and inference. This survey mainly focuses on a broad spectrum of linear attention techniques, including traditional linear attention methods, state space model (SSM) series, and linear recurrent neural networks (RNNs). These methods enable implicit historical information integration via state propagation, and achieve an approximately constant memory footprint as well as linear time complexity in sequence modeling tasks. Beyond algorithmic designs and model architectures, we further explore the characteristics, challenges, and successful applications of linear attention from a more comprehensive perspective. We also discuss the essential factors for practical hybrid frameworks, robust and efficient infrastructure, and scenario-specific features of downstream tasks, which jointly contribute to the successful deployment of linear attention mechanisms.

## 1 Introduction

The scaling laws of Large Language Models (LLMs) (Kaplan et al., 2020; Hoffmann et al., 2022; Sun et al., 2025d) have revealed the correlations between LLM performance and model/data scales, a breakthrough that has ushered in a new era of scaling up model architectures and data volumes to enhance LLM capabilities. Recently, the emerging trend of test-time scalingdriven primarily by long-context Chain-of-Thought (CoT) reasoning and reinforced by Reinforcement Learning (RL)has further pushed the boundaries of model capabilities, albeit at the expense of increased inference costs (DeepSeek-AI et al., 2025; OpenAI et al., 2024). The application of high-performance LLMs to generate reasoning traces and execute concrete actions by acting as intelligent agents has also garnered significant attention (Nakano et al., 2022; Yao et al., 2023; OpenAI, 2025). However, these trends inevitably drive up training and inference costs, as they entail scaled-up model sizes and lengthier sequence generations under quadratic attention complexity.

In this context, efficient LLMs have gradually drawn increasing interest from both academia and industry. Various efficiency-driven techniquessuch as linear attention (Katharopoulos et al., 2020; Qin et al., 2022b), Mixture-of-Experts (MoE) (Sun et al., 2024b; OpenAI et al., 2025), KV cache compression (Ge et al., 2023; Dai et al., 2024), and quantization (Frantar et al., 2023; Ma et al., 2024c)have been proposed and subsequently adopted in mainstream LLMs. With respect to sequence length, the computational bottleneck of traditional Transformers stems primarily from the quadratic complexity $O(N^2)$ relative to sequence length $N$, particularly under the current paradigm of extended context windows. **Linear attention**, which aims to reduce this quadratic complexity to linear $O(N)$, is an important research direction of efficient LLMs. We use the term in a broad operational sense: a class of sequence-modeling mechanisms that achieve linear-time inference by maintaining a fixed-size recurrent state for context compression and update. Under this view, state space models (SSMs) (Gu & Dao, 2024; Dao & Gu, 2024) and linear RNNs (Qin et al., 2023b; Beck et al., 2024) can be viewed as related parameterizations under a shared write/read recurrence. This unified view also provides the basis for discussing hybrid architectures, where linear modules are combined with full attention for competitive large-scale performance. Conventional linear attention methods can be roughly

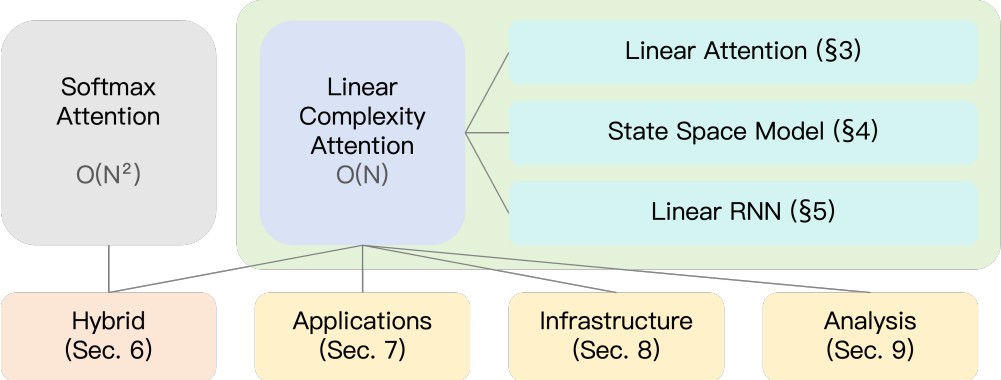

Figure 1: Overview of the survey. Softmax attention scales as $O(N^2)$ with a cache that grows with sequence length, whereas linear-complexity models scale as $O(N)$ by maintaining a fixed-size state that is written and read at each step. Under our unified view, this family has three instances: linear attention (Sec. 3), state space models (Sec. 4), and linear RNNs (Sec. 5), which differ primarily in their state parameterization and update/read-out rules; Tab. 1 compares these rules. The same write/read view organizes the later chapters: hybrid architectures (Sec. 6), applications (Sec. 7), infrastructure (Sec. 8), and analysis (Sec. 9).

categorized into three families: (a) approximation-based methods, which employ low-rank projections or kernel feature maps to approximate the softmax kernel with linear complexity (Katharopoulos et al., 2020; Arora et al., 2024b; Qin et al., 2022b); (b) gating-based methods, which replace the attention map with recurrent-style modeling and gating mechanisms for linear-time selective memory (He et al., 2025; Yang et al., 2024a; Sun et al., 2023b; Qin et al., 2024d); and (c) test-time training methods, which frame the recurrent state update as an online optimization process that adapts to contextual information (Sun et al., 2025e; von Oswald et al., 2025). Together with the SSM- and RNN-based models above, these methods constitute the linear-complexity design space covered in this survey.

Compared to full attention, linear attention offers clear advantages in training and inference efficiency for long contexts, yet it often lags in long-context retrieval and in-context learning (Wang et al., 2025a). This motivates hybrid full/linear architectures that combine the two (Lenz et al., 2025; MiniMax et al., 2025), where variations in the linear module, integration structure, and hyperparameters lead to different performance trade-offs. For practical deployment, LLMs equipped with linear attention have been validated across diverse domains, including NLP (Pitorro et al., 2024; Zhang et al., 2024d; Do et al., 2025), computer vision, speech, and multimodal tasks (Zheng & Wu, 2024; Xing et al., 2024; Li et al., 2024b; Erol et al., 2024), time series analysis (Patro & Agneeswaran, 2024a; Wu et al., 2024; Yuan et al., 2025), and AI4Science (Wang et al., 2024e; Yue & Li, 2024; Zhao et al., 2024b), where it can be tailored to each domain's challenges. Robust infrastructure support further serves as a fundamental guarantee for the efficiency advantage of linear attention (Yang et al., 2024a; Qin et al., 2024d), underpinning its stable deployment and competitive performance across downstream tasks (Qwen, 2025; Team et al., 2025a).

To evaluate linear attention against full attention, numerous studies have conducted in-depth quantitative analyses that yield valuable insights for designing practical (hybrid) architectures, focusing on core challenges such as long-context retrieval, length extrapolation, scaling laws, and a unified framework (Wang et al., 2025a; Schlag et al., 2021; Arora et al., 2024b; Ben-Kish et al., 2025a). Drawing on these findings and our own empirical insights, we summarize key recommendations for the future development of linear attention. Recent large-scale efforts suggest that linear attention, especially when integrated with hybrid architectures, is becoming a viable candidate for industrial-grade efficient LLMs (Team et al., 2025c; Qwen, 2025; Team et al., 2025a; Wang et al., 2025d).

Complementing this momentum, prior surveys have approached efficient sequence modeling from several angles: broad efficient-Transformer taxonomies that predate the selective-SSM era (Tay et al., 2022), treatments centered on a single family such as SSMs and Mamba (Patro & Agneeswaran, 2024b; Qu et al., 2025; Somvanshi et al., 2026; Wang et al., 2024d; Lv et al., 2025), RWKV (Li et al., 2025f), or fast-weight programmers viewed through a neurobiological lens (Irie & Gershman, 2025), and the recent and much broader

survey of Sun et al. (2025a), which spans the entire efficiency stack (linear and sparse modeling, efficient full attention, mixture-of-experts, hybrids, and diffusion LLMs). Building on these efforts, this survey takes the shared memory write/read recurrence as a single organizing axis and follows its consequences in depth, giving delta-rule and test-time-training methods, attention-level hybrids, and supporting infrastructure a more detailed treatment than a single-family or single-perspective organization allows.

Our contributions are threefold. First, we provide a unified view of linear-complexity sequence modeling that relates linear attention, state space models, linear RNNs, and test-time training under a shared memory write-and-read recurrence, yielding a consistent taxonomy of representative architectures that is carried through the later chapters of the survey (Fig. 1). Second, we connect this algorithmic view with practical deployment: hybrid architectures (Sec. 6) show how classical and linear layers are combined in real systems; applications (Sec. 7) show how task characteristics interact with linear designs; and infrastructure (Sec. 8) provides the concrete support that makes linear-time computation practical at scale. Third, we present a consistent terminology and comparison framework that aligns independently developed model families under a shared set of definitions, and distill practical observations on model selection, with particular attention to architectures already validated at large size and training scale (*e.g.*, Qwen3-Next (Qwen, 2025) and Kimi's KDA (Team et al., 2025a)) as well as newer academic designs.

This survey systematically reviews linear attention techniques and presents our in-depth discussions and insights, aiming to serve as a comprehensive reference for the design, adoption, and analysis of linear attention in real-world LLMs. The subsequent sections are organized as follows. In Sec. 2, we introduce the fundamental forms and inherent consistency of generalized linear attention. Sections 3 to 5 then provide detailed analyses of linear attention, the Mamba family, and RNN-based methods, respectively. Section 6 focuses on prevailing full/linear hybrid architectures. Practical applications across downstream tasks are presented in Sec. 7, followed by infrastructure details specific to linear attention in Sec. 8. Section 9 summarizes domain-specific challenges, characteristics, and solutions, along with our insights and recommendations.

## 2 Background

This section provides a foundational overview of conventional softmax attention mechanisms and various alternative architectures with linear computational complexityconsistent with the primary focus of this survey. In addition, we analyze the core distinct design concepts underlying the reviewed sequence models, with the goal of unifying these models under a comprehensive framework.

We begin by introducing the standard softmax attention paradigm, laying bare its quadratic complexity bottleneck and scalability constraints that arise with increasing sequence length. Subsequently, we present a systematic taxonomy of linear-complexity alternatives architectures, including linear attention, state-space models (SSMs), and linear recurrent neural networks (linear RNNs). Each category undergoes rigorous analysis regarding its development trajectory, representational capacity and typical formulation. Despite their disparate origins, these methodologies exhibit convergence: developed from diverse theoretical motivations, they ultimately embody variants of a unified linear recurrent paradigm. Concretely, all such models can be trained efficiently with specified parallel training techniques, whereas during inference, they operate via recurrent processes that act on a compact, fixed-dimensional state encoding historical contextresulting in constant space complexity and linear time complexity. We formalize this unification by proposing a generalized framework within which the reviewed methods are framed as special cases, and we present a systematic comparison in Tab. 1.

### 2.1 Standard Softmax Attention

Softmax attention has emerged as the canonical token-mixing module across a broad range of modern machine learning architecturesparticularly in natural language processing (NLP), computer vision (CV), and other sequential data processing tasks. Given an input hidden state $x$ that is projected into three vectors $Q, K, V \in \mathbb{R}^{N \times d}$, the standard softmax attention for language tasks is defined as:

Table 1: Memory update rules and corresponding objectives for attention variants.

| Type | Model | Memory Update | Memory Read-out |
|---|---|---|---|
| Softmax Attn | Attention (Vaswani et al., 2017) | $S_t = S_{t-1} \cdot \text{append}(k_t, v_t)$ | $o_t = V_t \text{softmax}(K_t^\top q_t)$ |
| | SWA | $S_t = S_{t-1} \cdot \text{append}(k_t, v_t) \cdot \text{drop}(k_{t-w}, v_{t-w})$ | $o_t = V_t \text{softmax}(K_t^\top q_t)$ |
| Linear Attn | LA | $S_t = S_{t-1} + v_t k_t^\top$ | $o_t = S_t q_t$ |
| | LA + normalizer (Qin et al., 2022a) | $S_t = \alpha S_{t-1} + v_t k_t^\top \quad z_t = z_{t-1} + k_t$ | $o_t = S_t q_t / (z^\top q_t)$ |
| | LA + kernel (Katharopoulos et al., 2020) | $S_t = S_{t-1} + v_t \phi(k_t)^\top$ | $O_t = S_t \phi(q_t)$ |
| | Performer (Choromanski et al., 2021) | $S_t = S_{t-1} + v_t \phi(k_t)^\top$ | $o_t = S_t \phi(q_t)$ |
| | Lightning Attn (Qin et al., 2024d) | $S_t = \alpha S_{t-1} + v_t k_t^\top$ | $o_t = S_t q_t$ |
| | RetNet (Sun et al., 2023b) | $S_t = \alpha S_{t-1} + v_t k_t^\top$ | $o_t = S_t q_t$ |
| | ABC (Peng et al., 2022) | $S_t^k = S_{t-1}^k + k_t \phi_t^\top, \quad S_t^v = S_{t-1}^v + v_t \phi_t^\top$ | $o_t = S_t^v \text{softmax}(S_t^k q_t)$ |
| | GLA (Yang et al., 2024a) | $S_t = S_{t-1} \text{diag}(\alpha_t) + v_t k_t^\top$ | $o_t = S_t q_t$ |
| | GSA (Zhang et al., 2024g) | $S_t^k = S_{t-1}^k \text{diag}(\alpha_t) + k_t \phi_t^\top, \quad S_t^v = S_{t-1}^v \text{diag}(\alpha_t) + v_t \phi_t^\top$ | $o_t = S_t^v \text{softmax}(S_t^k q_t)$ |
| | DeltaNet (Yang et al., 2024b) | $S_t = S_{t-1}(I - \beta_t k_t k_t^\top) + \beta_t v_t k_t^\top$ | $o_t = S_t q_t$ |
| | DFW (Mao, 2022) | $S_t = S_{t-1} \odot (\beta_t \alpha_t^\top) + v_t k_t^\top$ | $o_t = S_t q_t$ |
| | GatedDeltaNet (Yang et al., 2024a) | $S_t = S_{t-1}\left(\alpha_t(I - \beta_t k_t k_t^\top)\right) + \beta_t v_t k_t^\top$ | $o_t = S_t q_t$ |
| | RWKV-7 (Peng et al., 2025a) | $S_t = S_{t-1}\left(\text{diag}(\alpha_t) - \beta_t k_t k_t^\top\right) + \beta_t v_t k_t^\top$ | $o_t = S_t q_t$ |
| | Comba (Hu et al., 2025a) | $S_t = S_{t-1}\left(\alpha_t - \beta_t k_t k_t^\top\right) + \beta_t v_t k_t^\top$ | $o_t = S_t(q_t - dk_t)$ |
| | TTT-MLP (Sun et al., 2025e) | $S_t(\cdot) = S_{t-B}(\cdot) - \sum_{i=1}^B \beta_i \nabla_S \mathcal{L}(S_{t-1}, k_t, v_t)$ | $o_t = S_t q_t$ |
| | Titans (Behrouz et al., 2025c) | $M_t = (1 - \gamma_t)M_{t-1} + S_t, \quad S_t = \alpha_t S_{t-1} - \beta_t \nabla_M \mathcal{L}(M_{t-1}, k_t, v_t)$ | $o_t = M_t q_t$ |
| | MesaNet (von Oswald et al., 2025) | $S_t = \alpha_t S_{t-1} + \beta_t v_t k_t^\top, \quad H_t = \alpha_t H_{t-1} + \beta_t k_t k_t^\top$ | $o_t = S_t q_t$ |
| | DeltaProduct (Siems et al., 2025) | $S_t = S_{t-1} \prod_{i=1}^n (I - \beta_{ti} k_{ti} k_{ti}^\top) + \sum_{i=1}^n \prod_{k=i+1}^n (I - \beta_{tk} k_{tk} k_{tk}^\top)\beta_{ti} v_{ti} k_{ti}^\top$ | $o_t = S_t q_t$ |
| | Miras (Behrouz et al., 2025b) | $S_t = \alpha_t S_{t-1} - \beta_t \nabla_S \mathcal{L}(g, M_{t-1}, k_t, v_t)$ | $o_t = S_t q_t$ |
| | Atlas (Behrouz et al., 2025a) | $M_t = \gamma_t M_{t-1} - \eta_t \text{NS-5}(S_t), \quad S_t = \alpha_t S_{t-1} - \beta_t \nabla_M \mathcal{L}(M_{t-1}, k_t, v_t)$ | $o_t = S_t q_t$ |
| SSM | S4 (Gu et al., 2022b) | $S_t = S_{t-1} \odot \exp(-(\alpha 1^\top) \odot \exp(A)) + B \odot (v_t 1^\top)$ | $o_t = (S_t \odot C)1 + d \odot v_t$ |
| | Mamba (Gu & Dao, 2024) | $S_t = S_{t-1} \text{diag}(\alpha_t) + \beta v_t k_t^\top$ | $o_t = S_t q_t + d \odot v_t$ |
| | Mamba2 (Dao & Gu, 2024) | $S_t = \gamma S_{t-1} + v_t k_t^\top$ | $o_t = S_t q_t$ |
| Linear RNN | HGRN (Qin et al., 2023b) | $S_t = \alpha_t \odot e^{i\theta} \odot S_{t-1} + (1 - \alpha_t) \odot v_t$ | $o_t = S_t q_t$ |
| | RWKV-6 (Peng et al., 2024) | $S_t = S_{t-1} \text{diag}(\alpha_t) + v_t k_t^\top$ | $o_t = (S_{t-1} + (d \odot v_t)k_t^\top)q_t$ |
| | HGRN2 (Qin et al., 2024e) | $S_t = S_{t-1} \text{diag}(\alpha_t) + v_t(1 - \alpha_t)^\top$ | $o_t = S_t q_t$ |
| | xLSTM (Beck et al., 2024) | $S_t = f_t S_{t-1} + i_t v_t k_t^\top, \quad z_t = f_t z_{t-1} + i_t k_t$ | $o_t = S_t q_t / \max\{1, |z_t^\top q_t|\}$ |

$$o_i = \sum_{j=1}^i \frac{\exp(q_i^\top k_j)}{\sum_{l=1}^i \exp(q_i^\top k_l)} v_j. \tag{1}$$

Its parallel training formulation is as follows:

$$O = \text{softmax}(QK^\top \odot M) V. \tag{2}$$

$M$ denotes the attention mask designed to enforce causal constraints. Notably, its computational complexity is $O(N^2 d)$, which stems from the requirement to compute an attention score for every token pair in the input sequence. The resultant quadratic scaling in both computation time and memory—driven by the need to store the $N \times N$ attention matrix—renders training models on extremely long sequences prohibitively costly, and constitutes a key limitation of the standard Transformer architecture. During autoregressive inference—such as text generation—tokens are generated sequentially. Even when a key-value cache (KV cache) is employed to speed up inference, the latency of generating each token scales linearly with sequence length, the total computational cost of generating a sequence of length $N$ is still $\sum_i^N O(id) = O(N^2 d)$. Nevertheless, the memory footprint for the KV cache is $O(Nd)$, which can be substantial for long contexts and renders text generation memory-constrained. This total quadratic complexity and per-step cost that scales linearly during inference drive the development of more efficient, linear-time alternatives.

## 2.2 Linear Attention

To circumvent the quadratic-complexity bottleneck of softmax attention, a straightforward conceptual remedy entails removing the softmax non-linearity and eliminating the pairwise query–key interactions. Linear attention models originate from reordering the computation sequence of standard softmax attention through

decomposing the softmax function:

$$o_i \;=\; \frac{\sum_{j=1}^{i} \phi(q_i)^\top \phi(k_j)\, v_j}{\sum_{j=1}^{i} \phi(q_i)^\top \phi(k_j)} = \frac{\left(\sum_{j=1}^{i} v_j\, \phi(k_j)^\top\right) \phi(q_i)}{\left(\sum_{j=1}^{i} \phi(k_j)\right)^\top \phi(q_i)}. \tag{3}$$

In Eq. (3), $\phi$ is a projection function that can eliminate or replace the softmax operation in standard self-attention (Peng et al., 2021; Zhang et al., 2025d; Katharopoulos et al., 2020; Qin et al., 2022b). Since $\phi(K)\phi(V)$ is computed first, no attention matrix is materializedbecause the $QK$ matrix calculation is avoid-edthus reducing the theoretical time and memory complexity of linear attention to $O(N)$, which constitutes the core innovation of this mechanism. Nevertheless, training linear attention models is non-trivial for causal language modeling tasks. Because every output depends on the preceding state, we must either perform sequential training or materialize states of all time stepsincurring a space complexity of $O(Nd^2)$. Researchers have developed specialized techniquessuch as chunk-wise parallelizationthat leverage the linear characteristics of the model and enable efficient training by harnessing the parallel computing capabilities of modern GPUs (Qin et al., 2024d; Yang et al., 2024a; Sun et al., 2023b). We will elaborate on this in detail in Sec. 8.

During inference, the aforementioned equation can be reformulated in a recurrent formulation (Katharopoulos et al., 2020),

$$S_i = S_{i-1} + v_i\, \phi(k_i)^\top, \quad S_0 = 0, \tag{4}$$

$$z_i = z_{i-1} + \phi(k_i), \qquad z_0 = 0, \tag{5}$$

$$o_i = \frac{S_i\, \phi(q_i)}{z_i^\top \phi(q_i)}. \tag{6}$$

If we neglect the denominator term $z$ given that it is independent of query position, linear attention inference can be expressed as:

$$S_i = S_{i-1} + v_i\, \phi(k_i)^\top \in \mathbb{R}^{d_v \times d_k}, \quad o_i = S_i\, \phi(q_i). \tag{7}$$

Linear attention mechanisms fundamentally embody an RNN-style architecture characterized by linear computational complexity. At each temporal step $i$, a fixed-dimensional state $S_i \in \mathbb{R}^{d_v \times d_k}$ is maintainedone that eliminates the need to explicitly store the full history of key-value pairs. This design achieves an inference time complexity of $O(Nd^2)$ and a space complexity of $O(d^2)$. Consequently, linear attention proves particularly effective for processing long sequential contexts, as its memory footprint remains invariant to sequence length while preserving the model's capacity to capture long-range dependencies.

## 2.3 State-space

The state space is a foundational mathematical paradigm originating from Dynamic Systems Theory, which provides a rigorous framework for characterizing the temporal evolution of complex systems. Coupled differential equations are employed to encapsulate both the current state and its propagation dynamics in the context of control systems. Specifically, this paradigm can be formally expressed as:

$$x_t = Ax_{t-1} + Bu_t, \quad y_t = Cx_t + Du_t, \tag{8}$$

where $x_t \in \mathbb{R}^N$ denotes the latent state vector, $u_t \in \mathbb{R}^M$ represents the input signal, and $y_t \in \mathbb{R}^P$ constitutes the observable output at temporal index $t$. This formulation encapsulates a fundamental paradigm in which the input sequence $\{u_t\}$ undergoes transformation via an $N$-dimensional latent manifold before being mapped to the output sequence $\{y_t\}$. Pioneering work by Gu et al. (2020) first proved the feasibility of applying state-space models for large-scale language modeling tasks. Subsequent studies have expanded on this foundation with a suite of algorithmic refinements: principled initializations derived from orthogonal polynomial theory, judicious diagonalization assumptions that boost model expressivity and stability, and FFT-based convolutional and parallel-prefix (scan) algorithms that enable hardware-efficient training. These advances have evolved into a family of deep state-space architectures (Gu et al., 2020; 2022b; Gu & Dao,

2024; Gupta et al., 2022; Gu et al., 2022a; 2023; Dao & Gu, 2024; Smith et al., 2023b) that have addressed the three core goals of computational scalability, numerical stability, and algorithmic efficiency for state-space formulationsthus achieving performance comparable to that of softmax attention mechanisms in language modeling tasks. Similar to linear attention mechanisms, SSMs can also be categorized as a recurrent-style model, where $x_t$ is an updated state vector that occupies a fixed-size memory footprint.

## 2.4 Linear RNN

Prior to the advent of Transformers (Vaswani et al., 2017), recurrent neural networks (RNNs) and their advanced variants served as the dominant paradigm for sequence modeling tasks. Essentially, a traditional RNN model is formally expressed as:

$$h_t = \sigma(W^R h_{t-1} + W^I x_t), \quad y_t = W^O h_t, \tag{9}$$

where $x_t$ denotes the input vector, $h$ represents the evolving hidden state, and $y_t$ denotes the output at time step $t$; the parameter matrices $W^R W^I W^O$ are learnable. $\sigma$ denotes a nonlinear activation function here, typically selected as tanh or sigmoid. However, traditional RNN formulations inherently hinder efficient parallelization due to their temporal nonlinear recurrent dependencies. To circumvent this computational bottleneck, Martin & Cundy (2018); Smith et al. (2023b) proposed removing nonlinearities, thus enabling the use of parallel-prefix (scan) algorithms that deliver significant training speedups. Subsequent studies, including (Orvieto et al., 2023; Qin et al., 2023b; 2024e), have meticulously engineered architectural variants that reintroduce expressive capacity while retaining linear recurrence propertiesultimately achieving performance on par with state-of-the-art Transformer architectures. A typical gated linear RNN model (Orvieto et al., 2023; Qin et al., 2023b; 2024e) can be formally expressed as:

$$g_t = \sigma(W_g x_t + b_g), \tag{10}$$
$$i_t = \tau(W_i x_t + b_i), \tag{11}$$
$$o_t = \sigma(W_o x_t + b_o), \tag{12}$$
$$h_t = g_t \odot h_{t-1} + (1 - g_t) \odot i_t, \quad y_t = h_t \odot o_t, \tag{13}$$

where $\odot$ denotes element-wise multiplication, while alternative formulations utilize structured or fully diagonal recurrent matrices to lower computational complexity. Note that since $g_t$ and $i_t$ depends only on $x_t$, parallel scanning algorithms (Martin & Cundy, 2018; Smith et al., 2023b) can be utilized for efficient parallel training. Still, since linear RNNs originate from traditional RNN models, the latent state $h_t$ serves as a persistent memory vector that aggregates historical context information and undergoes adaptive refinement conditioned on the instantaneous input $x_t$.

## 2.5 A Memory-view Coherent Framework

Based on the foregoing discussions, contemporary linear attention mechanisms, state-space models, and linear RNNs can be formally subsumed within a unified linear attention framework. Concretely, these architectures avoid explicitly materializing pairwise token interactions during training; instead, they implement implicit historical information integration through state propagation. Consequently, during inference, they operate as recurrent processes that act on a compact, fixed-dimensional stateresulting in a constant memory footprint and linear time complexity with respect to sequence length.

Nevertheless, empirical evidence indicates that early linear attention variants demonstrate significantly inferior performance compared to their softmax-attention-based Transformer counterparts (Qin et al., 2022b; Katharopoulos et al., 2020); this performance gap may be fundamentally attributed to the inherent constraints of their historical-information indexing paradigms. In this section, we seek to unify these historical-information indexing paradigms from the perspective of **memory updating and retrieval**.

A standard softmax attention module can be conceptualized as a recurrent model whose memory footprint and per-step cost scale quadratically with a complexity of $O(N^2)$ with respect to sequence length. It assumes an unbounded memory bank: every new token is appended verbatim to the expanding KV cache,

and each query must traverse the entire, ever-expanding context. In contrast, a linear recurrent model implements a form of cognitive compression within a fixed memory budget. The state $S_t \in \mathbb{R}^{d_v \times d_k}$ serves as a compact working memory, which is updated by the incoming token; the token then queries this memory to extract relevant contextual information for next-token predictionmirroring a dynamic neural memory system. Equation (7) reveals that the linear attention mechanism fundamentally reduces to a cumulative summation operation, in which all historical information is aggregated with uniform weights. It becomes evident that as the number of accumulated tokens grows substantially, the contribution of each individual token diminishes asymptotically to infinitesimal levels. Consequently, the fixed-dimensional state proves insufficient for precisely reconstructing any tokena phenomenon that can be intuitively conceptualized as the progressive blurring and eventual obliteration of each token's mnemonic traces.

From the perspective of neural memory systems (Gershman et al., 2025), the core challenge lies in regulating the memory state: compression reduces computation costs yet introduces the risk of progressive information dilution and mutual interference, ultimately leading to performance degradation. Consequently, numerous methods have been proposed to effectively manage and retrieve memory information. We therefore formally define the linear recurrent model as follows:

$$S_t = A_t S_{t-1} B_t + \beta_t v_t k_t^\top \text{ (memory updating)}, \qquad o_t = S_t q_t \text{ (memory read-out)}, \tag{14}$$

where:

- $q_t, k_t \in \mathbb{R}^{d_k}$ and $v_t \in \mathbb{R}^{d_v}$ denote the query, key, and value projection vectors of the current input; these projections may take the form of linear projections or nonlinear feature mappings.

- $S_t \in \mathbb{R}^{d_v \times d_k}$ represents the state (memory) at time step $t$, so that the outer-product write $\beta_t v_t k_t^\top \in \mathbb{R}^{d_v \times d_k}$ and the read-out $o_t = S_t q_t \in \mathbb{R}^{d_v}$ are dimensionally consistent for arbitrary $d_k$ and $d_v$.

- $A_t \in \mathbb{R}^{d_v \times d_v}$ and $B_t \in \mathbb{R}^{d_k \times d_k}$ are left and right gates that modulate forgetting intensityi.e., the proportion of memory to be erased from the current state. These gates can take the form of diagonal matrices, identity matrices, full matrices, selection or shift operations, or rank-1 deflation matrices such as $I - \beta_t k_t k_t^\top$.

- $\beta_t \in \mathbb{R}$ denotes the write strength at time step $t$, *i.e.*, the extent to which input information should be written into the memory.

For instance, Lightning Attention (Qin et al., 2024d;a) and RetNet (Sun et al., 2023b) employ data-independent scalar decay to force the memory to forget distant contexts: $A_t = \alpha_t I,\ B_t = I,\ \beta_t = 1,$ with $\alpha_t$ a preset (data-independent) scalar. In contrast, Mamba (Gu & Dao, 2024) uses a data-dependent diagonal decay $\alpha_t$ computed from the input$(A_t = I,\ B_t = \mathrm{diag}(\alpha_t))$. GLA (Yang et al., 2024a) adopts data-dependent vector gating: $A_t = I,\ B_t = \mathrm{diag}(\alpha_t),\ \beta_t = 1,$ where $\alpha_t$ is derived from the input. DeltaNet (Yang et al., 2024b) and TTT (Sun et al., 2025e) implement more sophisticated delta-rule memory updating mechanisms (Peng et al., 2021; Schlag et al., 2021; Widrow & Hoff, 1988), where $A_t = I$ (or a scalar $\alpha_t$), and $B_t = I - \beta_t k_t k_t^\top$. We comprehensively summarize linear recurrent variants using Eq. (14) in Tab. 1.

## 3 Linear Attention

The self-attention mechanism constitutes the primary computational bottleneck of Transformers, owing to its $\mathcal{O}(N^2)$ time and memory complexity with respect to sequence length. This quadratic scaling severely limits the applicability of Transformers to long-sequence modeling tasks.

To mitigate this limitation, a growing body of research has explored *linear attention* methods, which seek to reduce the computational complexity of attention mechanisms from $\mathcal{O}(N^2)$ to $\mathcal{O}(N)$ while preserving maximal modeling capacity.

Broadly speaking, the literature on linear attention can be categorized into three primary families:

1. **Approximation-based Methods**: These methods approximate the softmax kernel or attention map via low-rank projections, kernel feature mappings, or matrix decomposition techniques (Katharopoulos et al., 2020; Choromanski et al., 2021; Xiong et al., 2021; Peng et al., 2021; Chen et al., 2021; Ma et al., 2021; Qin et al., 2022b; Hua et al., 2022; Duman Keles et al., 2023; Qin et al., 2022a; Zheng et al., 2023; Garnelo & Czarnecki, 2023; Qin et al., 2023a; Arora et al., 2024b).

2. **Gating-based Methods**: These methods replace or augment attention mechanism with recurrent-style updates and gating mechanisms for selective memory management (Sun et al., 2023b; Qin et al., 2024d; Ma et al., 2023; Yang et al., 2025b; Munkhdalai et al., 2024; Karami & Mirrokni, 2025; Yang et al., 2024a; Zhang et al., 2024g; Peng et al., 2022; Qin et al., 2024a;b; He et al., 2025).

3. **Test-time Training Methods**: These methods treat memory state matrices as fast-adaptive weights that are updated based on incoming information via an optimizer. (Sun et al., 2025e; Wang et al., 2025e; Behrouz et al., 2025c;a; von Oswald et al., 2025; Hu et al., 2025a; Behrouz et al., 2025b; Zhong et al., 2025a).

## 3.1 Approximation-based Methods

Approximation-based methods seek to reformulate the attention mechanism into a structure amenable to $\mathcal{O}(N)$ computation by leveraging the algebraic properties of kernel functions. Recall the standard softmax attention formulation:

$$\text{Attn}(Q, K, V) = \text{softmax}\left(\frac{QK^\top}{\sqrt{d}}\right) V, \tag{15}$$

where $Q, K, V \in \mathbb{R}^{N \times d}$ denotes the query, key, and value matrices corresponding to a sequence of length $N$. The quadratic computational cost stems from the explicit construction of the $N \times N$ similarity matrix $QK^\top$.

The core idea of linear attention is to approximate the softmax kernel using a *decomposable feature mapping* $\phi(\cdot)$, formulated as:

$$\exp\left(\frac{q_i^\top k_j}{\sqrt{d}}\right) \approx \phi(q_i)^\top \phi(k_j). \tag{16}$$

Based on this approximation, the attention computation can be rearranged as:

$$\text{Attn}(Q, K, V)_i \approx \frac{\phi(q_i)^\top \left(\sum_{j=1}^{N} \phi(k_j) v_j^\top\right)}{\phi(q_i)^\top \left(\sum_{j=1}^{N} \phi(k_j)\right)}. \tag{17}$$

Crucially, the summations over all keys and values can be updated incrementally rather than recomputed at each step, such that each query can attend to the accumulated state in $\mathcal{O}(d^2)$ time. This reduces the overall computational complexity from quadratic to linear with respect to sequence length. Different works propose diverse instantiations of the feature mapping $\phi(\cdot)$, including random Fourier features for unbiased kernel estimation (Choromanski et al., 2021), deterministic polynomial projections (Schlag et al., 2021), and cosine reweighting schemes that preserve locality (Qin et al., 2022b). Beyond feature mapping design, some works further enhance model stability and fidelity by integrating Nyström decompositions (Xiong et al., 2021; Wang et al., 2020), control variates techniques (Zheng et al., 2023), and normalization schemes to mitigate gradient explosion and attention dilution issues (Qin et al., 2022a). Approximation-based methods thus offer a unifying framework: they preserve the general architecture of Transformer while replacing the quadratic softmax attention mechanism with *kernelized or low-rank alternatives*, thereby enabling efficient training and inference on long sequences.

**Random Feature Kernelization** The earliest line of research observes that the exponential kernel underlying softmax attention can be approximated via random Fourier features. Performer (FAVOR+) (Choromanski et al., 2021) proposes positive orthogonal random features, formulated as:

$$\exp\left(\frac{q^\top k}{\sqrt{d}}\right) \approx \mathbb{E}_{\omega \sim \mathcal{N}(0,I)}\left[\cos(\omega^\top q)\cos(\omega^\top k) + \sin(\omega^\top q)\sin(\omega^\top k)\right]. \tag{18}$$

This yields an unbiased, low-variance estimator for softmax attention. Random Feature Attention (RFA) (Peng et al., 2021) further proposes a recurrent causal formulation, which maintains a dynamic running memory state $(S_t, z_t)$. EVA (Zheng et al., 2023) frames RFA within the control variate framework and refines the estimator by partitioning tokens into subsets, thus reducing variance while preserving linear computational runtime.

**Low-Rank Projection and Decomposition**  Another class of methods compresses the $N \times N$ attention matrix into low-rank surrogate matrices. Linformer (Wang et al., 2020) projects keys and values using low-rank projection matrices $E, F \in \mathbb{R}^{N \times k}$, formulated as:

$$\text{Attn}(Q, K, V) \approx \text{softmax}\left(\frac{Q(EK)^\top}{\sqrt{d}}\right)(FV), \tag{19}$$

where $k \ll N$. Nyströmformer (Xiong et al., 2021), by contrast, selects a set of landmark points $\tilde{K}$ and approximates the attention matrix as:

$$\text{softmax}(QK^\top) \approx \text{softmax}(Q\tilde{K}^\top)\,\text{softmax}(\tilde{Q}\tilde{K}^\top)^+\,\text{softmax}(\tilde{Q}K^\top), \tag{20}$$

where $^+$ denotes the MoorePenrose pseudoinverse. Luna (Ma et al., 2021) proposes a pack/unpack memory mechanism, which effectively implements a structured low-rank factorization strategy shared across model layers.

**Softmax-free Reparameterizations**  A parallel research direction challenges the necessity of the softmax operator itself (Banerjee et al., 2021). SOFT (Lu et al., 2021) replaces the exponential kernel with a Gaussian kernel defined as $k(q, k) = \exp(-\|q-k\|^2)$ a positive semi-definite (PSD) kernel that can thus be approximated via the Nyström method. Skyformer (Chen et al., 2021) maps non-PSD matrices to higher-dimensional PSD spaces prior to approximation. cosFormer (Qin et al., 2022b) proposes a ReLU feature mapping combined with cosine reweighting, formulated as:

$$f_{\cos}(q, k, i, j) = \text{ReLU}(q)^\top \text{ReLU}(k) \cdot \cos\left(\frac{\pi}{2} \cdot \frac{i-j}{M}\right), \tag{21}$$

this kernel can be linearly decomposed into two rank-one terms via Ptolemys identity.

**Normalization and Structural Refinements**  Linear attention mechanisms are often prone to training instability and attention dilution issues. TransNormer (Qin et al., 2022a) introduces NormAttention, a variant where normalization (via LayerNorm or RMSNorm) is applied directly to the attention output to stabilize gradient updates, defined as:

$$\text{NormAttn}(Q, K, V) = \text{XNorm}\left(\frac{Q(K^\top V)}{\|Q\| \cdot \|K\| + \epsilon}\right). \tag{22}$$

This normalization strategy helps mitigate gradient explosion and improves training stability. Building on this foundation, MetaLA (Chou et al., 2024) proposes the first provably theoretically optimal linear approximation of the softmax operator. This model eliminates redundant keys and relies solely on queries and a dynamic decay mechanism for efficient state updates, formulated as:

$$h_t = \Lambda_t h_{t-1} + v_t q_t^\top, \qquad o_t = h_t q_t, \tag{23}$$

where $\Lambda_t$ denotes a dynamic decay operator. MetaLA further integrates self-augmentation and short convolution modules to mitigate attention dilution and effectively capture local contextual information.

In summary, approximation-based methods replace softmax attention with decomposable surrogate modulesvia random feature mappings, low-rank matrix decompositions, alternative kernel functions, or structural refinementsthus enabling linear-time computation while attempting to preserve the model expressivity of full quadratic attention.

### 3.2 Gating-based Methods

Linear attention architectures provide a promising pathway to achieving linear time complexity and constant memory footprint in sequence modeling tasks by refactoring the self-attention mechanism. The core ideafirst formalized in the seminal work *Transformers are RNNs* by Katharopoulos et al. (2020)is that linear attention can be conceptualized as a stateful RNN architecture, formulated as follows:

$$s_i = s_{i-1} + \phi(x_i W_K) \cdot (x_i W_V)^T, \quad z_i = z_{i-1} + \phi(x_i W_K), \quad y_i = f_l \left( \frac{\phi(x_i W_Q)^T s_i}{\phi(x_i W_Q)^T z_i} + x_i \right). \tag{24}$$

This approach circumvents the quadratic computational complexity of standard self-attention by leveraging the associative property of matrix multiplication, thus enabling efficient recurrent inference. This formulation can be derived from softmax attention without the exponential operation. Nevertheless, early linear attention models often demonstrated suboptimal performance in comparison to softmax attention, largely owing to their reliance on naive additive memory updates schemes and constrained memory capacity. In the absence of more nuanced memory control mechanisms, these updates schemes tended to discard informative contextual signals and impair the model's capacity to capture long-range dependencies (Qin et al., 2022a). To mitigate this issue, gating mechanisms have been incorporated into linear attention frameworks to modulate the storage, update, and forgetting of memory states in an autoregressive manner.

**A Taxonomy of Gating Mechanisms.** A gating mechanism is an architectural component that regulates the flow of information and gradient propagation. In the context of linear attention, gating enables models to dynamically regulate their fixed-dimensional recurrent state. These mechanisms can be systematically classified along two primary dimensions:

- **Gating Granularity**: *Scalar gates* apply a scalar value to the entire hidden state, enabling coarse-grained yet computationally efficient control; in contrast, *vector or matrix gates* enable more fine-grained, dimension-wise modulation.

- **Data Dependency**: *Data-dependent gates* are dynamically derived from the input data, enabling adaptive memory management; *data-independent gates*, by contrast, depend on fixed or position-aware values and are typically employed for simple decay mechanisms.

**Data-Independent Gating.** Data-independent gating mechanisms are grounded in the intuition that distant contextual information is gradually downweighted (*i.e.*, decayed) to mitigate interference and prevent state overflow. Retentive Networks (RetNet) (Sun et al., 2023b) realize this via a data-independent decay factor $\alpha$ and a Swish gate within the Multi-Scale Retention module:

$$h_t = \gamma h_{t-1} + f(x_t), \tag{25}$$

where $\gamma \in (0,1)$ enforces exponential decay schedule and $h_t$ denotes the recurrent hidden state vector; MEGA (Ma et al., 2023), by contrast, is built on an Exponential Moving Average (EMA) framework to enhance long-range memory control, formulated as:

$$h_t^{(j)} = \alpha_j \odot u_t^{(j)} + \left(1 - \alpha_j \odot \delta_j\right) \odot h_{t-1}^{(j)}, \tag{26}$$

where $\alpha_j$ governs the forgetting rate, $\delta_j$ tunes the update stability, and the gated term $\alpha_j \odot u_t^{(j)}$ regulates the magnitude of new input updates.

Similarly, Lightning Attention v1 and v2 (Qin et al., 2024d;a) also employ data-independent scalar gating mechanisms, where a fixed scalar gate modulates the trade-off between the contribution of previous memory states and incoming token information. This family of methods embodies a core principle: memory retention can be regulated via simple, input-agnostic scalar gates, enabling efficient modeling of long-range dependencies without incurring additional per-step computational overhead.

**Data-Dependent Scalar Gating.**   This category employs a single input-aware scalar to modulate recurrent memory states. Hu et al. (2025a) propose Comba, a model inspired by closed-loop control theory. The Infini-Attention  (Munkhdalai et al., 2024) mechanism adopts a similar approach with a distinct objective, combining local attention and long-term linear attention to model infinitely long input sequences. It employs a learned scalar gate $\beta$ to fuse the outputs of these two attention branches, acting as a "memory mixer" that dynamically adjusts the emphasis on local versus long-range context; Lattice  (Karami & Mirrokni, 2025), by contrast, adopts orthogonal update mechanisms.

**Data-Dependent Vector and Matrix Gating.**   This class of models employs gates of finer granularity, which are applied at the dimension or feature level to achieve fine-grained modulation of memory states. Decaying Fast Weights  (Mao, 2022) proposes a matrix-valued gate that modulates the hidden state via element-wise multiplication; this design prevents the chaotic mixing of hidden state dimensions. Similarly, RODIMUS* (He et al., 2025) integrates a data-dependent tempered selection (DDTS) mechanism into a linear attention framework to adaptively filter out irrelevant information and achieve semantic compression.

Another seminal work in this area is Gated Linear Attention (GLA) (Yang et al., 2024a), which augments the linear attention update rule with a learnable, data-dependent 2D gating matrix $G_t \in \mathbb{R}^{d \times d}$. Building on this foundation, Gated Slot Attention (GSA)  (Zhang et al., 2024g) integrates gating mechanisms into the Attention with Bounded-Memory-Control (ABC) framework (Peng et al., 2022). ABC provides a unifying paradigm for memory-efficient attention by abstracting memory as a fixed set of slots. It generalizes several approximation strategies: (a) Linformer (Wang et al., 2020), which compresses $N$ tokens into $n$ representations via low-rank projection matrices; (b) *clustering*-based methods, which partition $N$ tokens into $n$ clusters and use cluster centroids as representative tokens; and (c) *sliding-window* attention mechanisms, which maintains a dynamically updated queue of size $n$. Distinct from these strategies, ABC (Peng et al., 2022) itself proposes an iterative compression mechanism that uses a shared learnable matrix $W$ to map $N \to n$ while enabling bounded-memory updates. Within this framework, GSA leverages GLA-inspired gating mechanisms to implement context-aware memory reading and adaptive forgetting.

In parallel, inspired by partial attention mechanisms (Dai et al., 2019; Zaheer et al., 2020), FLASH introduces the Gated Attention Unit (GAU), which incorporates attention with a Gated Linear Unit (GLU). GAU downplays the dominance of the attention mechanism via a GLU-like gating module. Building on this design, TransNormerLLM  (Qin et al., 2024a) employs both a gating strategy for training stability and a simplified GLU (SGLU) to boost computational efficiency. Similarly, LightNet (Qin et al., 2024b) adopts GLU-based gating mechanisms for fine-grained modulation of multi-dimensional feature representations.

Table 2: Comparison of Different Gated Memory Models.

| Model Name | Gating Type | Data Dependency | Core Memory Update Rule | Unique Gating Role |
| --- | --- | --- | --- | --- |
| Gated DeltaNet | Scalar | Data-Dependent | Delta rule based update | Rapid memory clearance and targeted updates |
| Infini-attention | Scalar | Data-Dependent | Local/Global memory fusion | Acts as a mixer for different memory sources |
| RetNet | Scalar | Data-Independent | Exponential decay | Fixed positional decay |
| MEGA | Matrix | Data-Independent | Exponential decay | Based on EMA |
| GLA | Matrix | Data-Dependent | Multiplicative and additive updates | Selective control over hidden state dimensions |
| Decaying Fast Weights | Matrix | Data-Dependent | A pure element-wise operation | Enable more control and efficient parallelization on GPU |
| Gated Slot Attention | Vector | Data-Dependent | Memory slot compression and updates | Memory compression and adaptive forgetting |
| Gated Attention Unit (GAU) | Matrix | Data-Dependent | Attention-based gating | Re-defines attention itself as a gating mechanism |
| TransNormerLLM (SGLU) | Matrix | Data-Dependent | Attention-based gating | Removes the activation function from the original GLU structure |
| Random Feature Attention | Scalar | Data-Dependent | Low-rank kernel update with recency bias | Learns and enhances recency bias |
| Lattice | Scalar | Data-Dependent | Orthogonal update | Stores only novel, non-redundant information |
| LightNet | Vector | Data-Dependent | GLU-based additive update | Manages information flow for multi-dimensional data |
| RODIMUS* | Vector | Data-Dependent | Data-Dependent Tempered Selection (DDTS) | Acts as a smart compressor, autonomously filtering irrelevant information |
| Comba | Scalar | Data-Dependent | Scalar-plus-low-rank state transition | With both state feedback and output feedback corrections |

**Gating and Positional Encodings.**   Positional information is critical for sequence modeling tasks, and linear attention methods have devised strategies to integrate it while preserving linear computational complexity. Among these approaches, Rotary Position Embedding (RoPE) (Su et al., 2024) is particularly well suited for linear attention mechanisms, as it imposes a rotation transformation on the query and key vectors independently prior to the computation of their inner product. This implies that RoPE introduces position-dependent interactions without altering the linear structure of the attention kernel, enabling its seamless integration with linear-time attention mechanisms. It is further recommended to apply RoPE after the kernel function (Chen et al., 2025a) to prevent the kernel function from distorting positional information. In

addition, PaTH Attention (Yang et al., 2025c) is a flexible, data-dependent positional encoding method that employs accumulated products of Householder-like transformations. This approach can be regarded as a form of data-dependent multiplicative gating, as it dynamically adjusts state transitions according to the input sequence. Another representative approach is Linearized Relative Positional Encoding (LRPE) (Qin et al., 2023a), which generalizes the Rotary Position Embedding (RoPE) mechanism. It reformulates position bias terms into query and key factors that are fully compatible with the linear attention kernel, thus extending relative positional encoding to the linear attention regime. Moreover, it supports higher-dimensional variants such as MD-LRPE (Qin et al., 2024b).

Table 3: Comparison of representative positional encoding methods.

| Method | Encoding Type | Key Characteristics |
|---|---|---|
| Absolute PE | Fixed or learned embeddings added to inputs | Provides absolute token positions; simple but lacks relative inductive bias. |
| RoPE | Data-independent unitary rotation (complex multiplication) | Encodes relative positions via rotations; widely used in LLMs but tied to quadratic attention. |
| LRPE | Generalized unitary transformations reformulation | Extends RoPE to linear attention; supports higher-dimensional variants such as MD-LRPE. |
| PaTH | Data-dependent Householder-like transformations accumulated along sequence | Stronger expressivity (up to $NC^1$); improves state tracking and long-range generalization; inference via in-place key updates. |

### 3.3 Test Time Training

A recent and influential line of research reinterprets linear attention from the perspective of test-time learning. The seminal proposal of Test-Time Training (TTT) (Sun et al., 2025e) equips sequence models with hidden states that function as lightweight parametric models, which are updated online during inference by leveraging (key, value) pairs. By updating the hidden state matrix $S$ to approximate $v = Sk$, contextual information can be compressed into $S$, and the product $Sq$ can be used to generate the output $o$.

These hidden states can be conceptualized as model-like memory components, where each state retains a local parameterization to interact with incoming input tokens. The optimization objective of these hidden states typically involves minimizing a task-specific loss or reconstruction error during test time, thereby effectively aligning the memory with the current input context. Through this interaction, the system can suppress redundant or irrelevant information by selectively updating only the most predictive or informative segments of the memory, while preserving other stored content.

This approach distinguishes itself from prior methods (*e.g.*, standard linear attention mechanisms or static memory mechanisms): unlike methods that rely solely on pre-trained weights, the memory here adapts dynamicallyperforming online regression or update steps that enable the model to flexibly integrate new observations without overwriting previously stored relevant information. In essence, TTT bridges the gap between parametric sequence modeling and non-parametric, input-dependent memory, thereby offering a more flexible and context-aware mechanism for sequence processing tasks.

Concretely, each input $x_t$ is projected into the triplet $(k_t, v_t, q_t)$, and the layer defines a self-supervised regression loss function as $\ell(W; x_t) = \|f(k_t; W) - v_t\|^2$; the internal parameters are then updated via a gradient descent step, formulated as:

$$W_t = W_{t-1} - \eta \nabla_W \ell(W_{t-1}; x_t). \tag{27}$$

The predictions are generated as $z_t = f(q_t; W_t)$. In the linear regime, this procedure yields $z_t \approx VK^\top q_t$ a canonical formulation of linear attention.

Building on this foundation, the Test-Time Regression (TTR) (Wang et al., 2025e) framework generalizes this perspective: associative recalla core cognitive ability to learn and retain relationships between distinct entities, even when these entities are not directly correlatedis framed as solving a regression problem during test time. Within this framework, various design dimensions are explored, including the choice of regressor class, weighting scheme, and optimization strategy. TTR further offers an alternative interpretation of numerous well-established architectures, including linear State Space Models (SSMs)which can be regarded as special cases under this regression-based perspective.

Following the delta rule principle, Delta Network (DeltaNet) (Yang et al., 2024b) is proposed as a linear attention model that incrementally updates memory states via a delta-rule-inspired mechanism. The delta ruleoriginally formulated for online weight adaptationupdates parameters as follows:

$$\Delta S_t = \eta \left( v_t - S_t k_t \right) k_t^\top, \tag{28}$$

this update adjusts $S_t$ proportionally to the prediction error $v_t - S_t k_t$. By adopting this principle, DeltaNet continually refines its memory representations and mitigates the severe information-overload issue prevalent in purely additive linear attention models, as each update step is explicitly guided by the error signal. Building on this design, Gated Delta Network (Gated DeltaNet) (Yang et al., 2025b) extends DeltaNet by introducing a data-dependent scalar gate $\alpha_t \in (0, 1)$ a design analogous to that of Mamba. This gate enables flexible memory control: setting $\alpha_t \to 0$ allows for rapid memory clearance, whereas setting $\alpha_t \to 1$ facilitates targeted updates without perturbing other stored informationeffectively combining the advantages of additive updates with selective memory retention. This approach is rooted in the "fast weight programmers" paradigm (Schlag et al., 2021), which replaces purely additive updates with delta rule-inspired update rules that allows the model to learn and correct its key-value associations. This approach has been demonstrated to enhance associative recall performance and is further optimized via chunk paralleling to improve hardware efficiency, as detailed in (Yang et al., 2024b).

Subsequent works explore this design space along complementary research directions. Titans (Behrouz et al., 2025c) introduces a high-capacity contextual memory module that learns to memorize contextual information during test time, enabling efficient recall over extremely long histories via a "Surprise" metric derived from the gradient of the associative memory neural network. Atlas (Behrouz et al., 2025a) builds on this idea with a primary focus on *contextual* memorization. Building on these insights, DeltaProduct (Siems et al., 2025) enhances state tracking capability in linear RNNs by executing multiple gradient descent update steps per token; each update constructs the state-transition matrix as a product of generalized Householder transformations. Additionally, the Mesa layer (von Oswald et al., 2025) formalizes test-time memory updates as an *optimal linear regression problem.* Unlike prior approaches that rely on incremental updates, the Mesa layer computes the linear mapping that minimizes the cumulative regularized squared error over all historical inputs, achieving one-shot associative memory capabilities. Its parallelized implementation leverages conjugate gradient methods to enable stable and efficient updates, while preserving the capability to dynamically forget obsolete information or integrate new inputs. In contrast, instead of using small sequence chunks, Zhang et al. (2025c) proposes a large chunk test-time training paradigm that significantly improves hardware utilization and facilitates the scaling of nonlinear state dimensions.

From a broader theoretical perspective, recent research examines Transformer architectures from the perspective of associative memory, drawing inspiration from human cognition mechanisms. Two core aspects are explored in relevant studies (Behrouz et al., 2025b; Zhong et al., 2025a): *memory capacity*, which involves the effectiveness of softmax attention and reinterpreting Feed-Forward Networks (FFNs) as associative memory components; (2) *memory update*, which provides a unified framework for understanding how architectural variants evolve their internal knowledge representations. Behrouz et al. (2025b) and Zhong et al. (2025a) focus on these two aspects, aiming to explore the core architectural and memory-related design principles.

### 3.4 Other Methods

Recent research has proposed several innovative attention mechanisms that depart from traditional softmax-based paradigms, with the goal of enhancing model efficiency and scalability in Transformer architectures. Tensor Product Attention (TPA) (Zhang et al., 2025f) utilizes tensor decompositions techniques to compactly

represent queries, keys, and values, thereby reducing memory overhead and enabling the efficient processing of longer sequences. 2-Simplicial Attention (Roy et al., 2025) generalizes standard dot-product attention to trilinear functions formulations, achieving enhanced token efficiency. Garnelo & Czarnecki (2023) explores the KVQ space, identifying that certain modelssuch as those formulated based on least squares optimizationcan generalize linear attention mechanisms and provide computationally efficient alternatives with equivalent complexity.

# 4 State Space Models and Mamba Series

State Space Models (SSMs) have emerged as a scalable, linear-time alternative to attention mechanisms, evolving from control-theoretic frameworks into high-performance sequence modeling architectures. Their development can be categorized into three major research directions:

- **Foundational linear time-invariant (LTI) SSMs and Structured Models.** Early research established the core recurrentconvolutional duality of SSMs and principled long-range memory. This includes the formal formulation of continuous and discrete SSMs, the HiPPO framework (Gu et al., 2020; 2023), the convolutional perspective introduced by LSSL (Gu et al., 2021), and the structured S4 model family (Gu et al., 2022b). Subsequent architectural simplifications, including GSS (Mehta et al., 2023), H3 (Fu et al., 2023), DSS (Gupta et al., 2022), S4D (Gu et al., 2022a), and S5 (Smith et al., 2023b) enhanced efficiency, expressiveness, and hardware compatibility.

- **Selective and Input-Dependent SSMs (the Mamba Paradigm).** Mamba (Gu & Dao, 2024) proposed Selective SSMs, where parameters are dynamically conditioned on input tokens, enabling dynamic information retention while preserving strict $O(L)$ computational complexity via a hardware-aware parallel scan algorithm. Variants such as Liquid-S4 (Hasani et al., 2023) explore richer adaptive dynamics behaviors; SSD (Structured State Space Duality) (Dao & Gu, 2024) unifies selective SSMs with attention mechanisms and inspires the design of hybrid models such as Mamba-2.

- **Architectural and Multidimensional Extensions.** Beyond 1D sequential data, SSMs have been extended to spatial and spatiotemporal data modalities. ConvSSM (Smith et al., 2023a) establishes a formal connection between convolution dynamics and SSMs, while MambaMixer (Behrouz et al., 2024) adapts selective mixing mechanisms to images and video data via tokenchannel dual SSM operations. These extensions demonstrate how core SSM principles generalize naturally to computer vision and multimodal modeling tasks.

## 4.1 State Space Model (SSM)

A standard continuous-time State Space Model (SSM) is defined by a pair of linear Ordinary Differential Equations (ODEs), which map an input function $u(t)$ to an output function $y(t)$ via a latent state $x(t) \in \mathbb{R}^N$, formulated as:

$$x'(t) = Ax(t) + Bu(t), \quad y(t) = Cx(t) + Du(t). \tag{29}$$

Here, $A \in \mathbb{R}^{N \times N}$ denotes the state transition matrix, while $B \in \mathbb{R}^{N \times 1}$, $C \in \mathbb{R}^{1 \times N}$, and $D \in \mathbb{R}^{1 \times 1}$ represent projection matrices. For deployment in deep learning frameworks, this continuous-time system must be discretized. A step size $\Delta$ is introduced to transform the continuous parameters $(A, B)$ into their discrete counterparts $(\bar{A}, \bar{B})$. A canonical discretization method is the Zero-Order Hold (ZOH) scheme, which yields the following discrete-time recurrence relations:

$$\bar{A} = \exp(\Delta A), \quad \bar{B} = (\exp(\Delta A) - I)A^{-1}B, \tag{30}$$

$$x_k = \bar{A}x_{k-1} + \bar{B}u_k, \quad y_k = Cx_k + Du_k. \tag{31}$$

This discretized formulation exhibits a crucial recurrent-convolutional duality. It can be computed in a **recurrent** fashion, where the equations define a sequential model analogous to a Recurrent Neural Network

(RNN). This recurrent mode is highly efficient for inference, as each step only requires a simple state update, leading to $\mathcal{O}(1)$ per token computational complexity. Alternatively, the recurrent formulation can be unrolled to reveal its dual nature as a **convolutional** model. By assuming a zero initial state $x_{-1} = 0$, the output can be expressed as a function of the entire input sequence, given by:

$$y_k = C\bar{A}^k\bar{B}u_0 + C\bar{A}^{k-1}\bar{B}u_1 + \cdots + C\bar{B}u_k + Du_k = (\bar{K} * u)_k. \tag{32}$$

This corresponds to a convolution operation, where $\bar{K}$ denotes a structured convolutional kernel of length $L$, defined as $\bar{K}_i = C\bar{A}^iB$. This convolutional representation is pivotal to efficient model training, as it enables parallel training across all time stepsanalogous to Convolutional Neural Networks (CNNs), and is typically accelerated via highly efficient Fast Fourier Transform (FFT) algorithms. The core challenge thus shifts to identifying optimal $(A, B, C)$ matrix configurations that enable the model to effectively capture long-range sequence dependencies. This is precisely the problem that the HiPPO framework is designed to tackle.

## 4.2 The Theoretical Cornerstone: The HiPPO Framework

The core challenge for SSMs lies in designing the state transition matrix $A$ to effectively compress long sequential histories into the finite-dimensional latent state $x(t)$. The HiPPO (High-order Polynomial Projection Operator) framework (Gu et al., 2020) offered a fundamental solution to this problem. Its core idea is to reframe the notion of "memory" as an online function approximation task: identifying a function $g_t(\cdot)$ that optimally approximates the historical trajectory of an input function $f(\cdot)|_{(-\infty, t]}$. HiPPO achieves this by projecting the historical input function onto a basis of orthogonal polynomials (*e.g.*, Legendre polynomials). The latent state $x(t) \in \mathbb{R}^N$ is defined as the coefficient vector of this projection, and the framework derives a linear ODE that governs the evolution of these coefficients, formulated as:

$$\frac{dx}{dt}(t) = A(t)x(t) + B(t)f(t). \tag{33}$$

However, the original HiPPO framework was formulated for time-varying systems. Gu et al. (2023) established a crucial theoretical bridge to time-invariant SSMs. This work introduced the Generalized Orthogonal State Space Model framework and proved that employing a constant HiPPO matrix in an LTI SSM is equivalent to projecting the input sequence onto a fixed, exponentially weighted basis. This work justified the adoption of the now-standard LegS state transition matrix, which is optimal for a Legendre polynomial basis under an exponential decay metric, and whose elements are precisely defined as follows:

$$A_{nk} = -(2n+1)^{1/2}(2k+1)^{1/2} \cdot \begin{cases} 1 & n > k \\ \frac{n+1}{2n+1} & n = k \\ 0 & n < k \end{cases}, \quad B_n = (2n+1)^{1/2}. \tag{34}$$

This theoretical framework provides a rigorous mathematical guarantee for the long-range dependency modeling capabilities of SSMs and clarifies that the discretization step size $\Delta$ directly governs the length of the dependency horizon ($\approx 1/\Delta$).

## 4.3 Evolution of LTI Models: From Structured to Simple

**LSSL: The Bridge from Theory to Practice** The HiPPO framework laid the theoretical foundation for the design of SSM matrices. However, a critical practical bottleneck persisted: how to efficiently train such models within standard deep learning training paradigms. A naive direct implementation of the SSM's recurrent formulation, while efficient for stepwise inference, is inherently sequential in nature. This characteristic precludes efficient parallelization across the time dimension, creating a major training bottleneck on modern hardware (*e.g.*, GPUs) that excels at parallel computation. The Linear State Space Layer (LSSL) (Gu et al., 2021) established the critical bridge from theory to practice by addressing this training efficiency bottleneck. It rigorously proved that for any LTI SSM, the recurrent computation is mathematically equivalent to a global convolution operation. Specifically, the entire output sequence can be computed in parallel by convolving the input sequence with a single, large convolutional kernel derived from the SSM's core parameters $(A, B, C)$.

This "convolutional view" represented a breakthrough. It recast the training process from a sequential loop that cannot be parallelized into a single, highly parallelizable global convolutional operation. This enables ultrafast model training via standard acceleration techniques such as the Fast Fourier Transform (FFT). By decoupling the training and inference modalities, LSSL demonstrated that a single model could benefit from both parallel training (via convolution) and efficient autoregressive inference (via recurrence), thereby resolving a key trade-off in sequence modeling tasks.

**S4: The Structured State Space Milestone**  While LSSL demonstrated theoretical viability, the computational overhead associated with dense HiPPO matrices $A$ remained a practical bottleneckparticularly for the computationally expensive matrix exponentiation step required during the discretization process. The S4 (Structured State Space) model (Gu et al., 2022b) addressed this challenge via an ingenious structural design strategy. S4's core innovation was to impose a Diagonal Plus Low-Rank (DPLR) structure constraint on the HiPPO matrix, formulated as $(A = \Lambda - PP^*)$. This particular structural constraint, while serving as a highly effective approximation, renders the state transition matrix mathematically tractable. It enables efficient computation of the discretized convolutional kernel $\bar{K}$ without explicitly materializing the dense matrix form, instead leveraging a specialized algorithm rooted in generating function theory. This enables fast parallel training with $\mathcal{O}(L \log L)$ computational complexity via the FFT acceleration. This structured design approach served as a cornerstone for subsequent SSM model architectures, laying the foundation for the S6 (Selective Structured State Space) layerlater integrated into the Mamba architecture.

**Bridging the Gap to Language Modeling**  While S4 and its variants delivered promising performance on long-range sequence modeling benchmarksespecially for continuous datatheir success did not readily generalize to natural language modeling tasks. These early LTI SSMs initially struggled to match the performance of Transformers in this domain, which spurred a wave of follow-up research aimed at closing the performance gap. One notable endeavor in this direction was H3 (Hungry Hungry Hippos) (Fu et al., 2023), which enhanced model expressiveness by mimicking the QKV mechanism of self-attention via multiplicative interactions. Its core computation involves stacking a local shift SSM (with state matrix $A_{\text{shift}}$ acting as a shift operator) and a long-range diagonal SSM (initialized via the HiPPO framework), formulated as:

$$y_i = Cx_i + Du_i \odot (Q \cdot \text{SSM}_{\text{diag}}(\text{SSM}_{\text{shift}}(K) \odot V)). \tag{35}$$

To further improve computational efficiency, H3 also introduced FLASHConva hardware-aware convolution algorithm tailored for long-sequence convolutions. Similarly, the Gated State Space (GSS) model (Mehta et al., 2023) incorporated gating mechanisms analogous to those of Gated Linear Units (GLUs) to accelerate model training. Notably, GSS also challenged the prevailing reliance on HiPPO initialization by demonstrating that simplified random initialization strategies could yield comparable performancesuggesting that the model's architectural design was the key to its effectiveness.

**The Push for Simplicity: DSS and S4D**  A major subsequent research direction focused on simplifying the complex DPLR structure of S4. The Diagonal State Space (DSS) model (Gupta et al., 2022) proposed a radical simplification, proving that a purely diagonal state space modelwith state matrix $(A = \text{diag}(\lambda_1, \ldots, \lambda_N))$possesses sufficient modeling capacity. This reduced the parameter count from $\mathcal{O}(N^2)$ to $\mathcal{O}(N)$ and dramatically simplified model implementation. Building on this simplification, S4D (Gu et al., 2022a) explored strategies for effectively parameterize and initializing such diagonal models. It theoretically proved that a diagonal approximation of the HiPPO matrix is asymptotically equivalent to the original full-rank matrix and proposed simple yet effective closed-form initialization schemes, such as: S4D-Lin variant : $\lambda_n = -1/2 + i\pi n$. These studies verified that most of the modeling power of SSMs can be preserved in a far simpler, more accessible diagonal parameterization.

**S5: Improving Hardware-Friendliness**  The S5 model (Smith et al., 2023b) further boosted computational efficiency by replacing the parallel Single-Input Single-Output (SISO) models of S4 with a single Multiple-Input Multiple-Output (MIMO) modeland crucially, replacing FFT-based convolution operations with a parallel scan algorithm. By leveraging the associative property of the linear recurrence relations, S5 computes the output via a hardware-efficient, tree-structured reduction operation. This efficiency gain is

enabled by diagonalizing the state transition matrix $A = V\Lambda V^{-1}$, which simplifies the recurrent operator to facilitate efficient parallel scan execution.

## 4.4 Mamba Series: The Paradigm Shift to Selective and Dynamic SSMs

**Mamba: Content-Aware Selection through Input-Dependent Parameters** All the aforementioned SSMs fall under the category of Linear Time-Invariant (LTI) models. Mamba (Gu & Dao, 2024) ushered in a fundamental paradigm shift by proposing the ***Selective State Space Model (Selective SSM)***. Its core innovation lies in breaking the LTI constraint by making the model's parameters dependent on the input sequence. Specifically, Mamba parameterizes the time step $\Delta$ and the matrices $B$ and $C$ as functions of the current input token $u_k$, formulated as:

$$\Delta_k = \text{softplus}(\text{Linear}_\Delta(u_k)), \quad B_k = \text{Linear}_B(u_k), \quad C_k = \text{Linear}_C(u_k). \tag{36}$$

Applying the Zero-Order Hold (ZOH) discretization rule then yields time-varying matrices $\bar{A}_k = e^{\Delta_k A}$ and $\bar{B}_k = (\Delta_k A)^{-1}(e^{\Delta_k A} - I)B_k$, resulting in a time-varying state update equation:

$$x_k = \bar{A}_k x_{k-1} + \bar{B}_k u_k, \quad y_k = C_k x_k. \tag{37}$$

This input-dependent mechanism empowers the model to selectively forget or retain contextual information based on input content. However, the abandonment of time invariance renders the efficient convolution-based training method inapplicable. To address this challenge, Mamba proposed a hardware-aware parallel scan algorithm. This algorithm enables the exact parallel computation of recurrent updates while maintaining a strict $\mathcal{O}(L)$ linear computational complexity with respect to sequence length. The combination of content selectivity and linear-time training makes Mamba exceptionally well-suited for constructing large-scale language model capable of handling extremely long contextsa domain where the quadratic computational cost of Transformers becomes prohibitive. Crucially, this content-selective mechanism was proven to be the key to achieving high performance. At the multi-billion parameter scale, Mamba became the first SSM-based architecture to achieve performance comparable toand in some cases surpassinghighly optimized Transformer models, positioning itself as a credible and highly efficient alternative for next-generation foundation models.

**Alternative Input-Dependent Dynamics** Mamba's success sparked widespread interest in exploring other forms of input-dependent dynamics mechanisms. The Liquid-S4 model (Hasani et al., 2023) integrated concepts from Liquid Time-Constant (LTC) networks by making the continuous-time state transition matrix $A$ itself a function of the input. In its linearized formulation, the state transition is governed by $\dot{x}(t) = [A + Bu(t)]x(t) + Bu(t)$. This dynamic causality, when combined with S4's DPLR structure constraint, yields a model that is inherently adaptive to heterogeneous time-series data.

**Unifying SSMs and Attention: The Structured State Space Duality (SSD)** The theoretical connection between SSMs and attention mechanisms was rigorously clarified by the Structured State Space Duality framework (Dao & Gu, 2024). The seminal paper *Transformers are SSMs* (Dao & Gu, 2024) reveals that the recurrent computation of a selective SSM (*i.e.*, $h_t = A_t h_{t-1} + B_t x_t, y_t = C_t^\top h_t$) is mathematically equivalent to multiplication by a large, implicit matrix with a semi-separable structure. This groundbreaking discovery fundamentally unifies SSMs and structured attention models under a common mathematical framework.

This duality is not merely a theoretical curiosityit serves as the direct technical foundation for the next generation of hybrid models, namely Mamba-2. Mamba-2 operationalizes this theoretical insight by processing sequences in blocks and decomposing the computation into two complementary steps. Specifically, within each block, it employs an efficient Mamba-style parallel scan for state evolution, thereby maintaining linear-time computational complexity. For cross-block interactions, it then computes an explicit, dense attention-like matrix. This quadratic operation, however, is applied only to the compressed states representations of each block rather than the entire token sequence, thereby rendering the computationally expensive operation manageable. This hybrid design enables Mamba-2 to strategically combine the linear-time efficiency of SSMs with the strong expressive power of quadratic attention mechanismsapplied at a much smaller scale. By providing both a unified theoretical framework and a hardware-friendly implementation

(the block-based SSD algorithm), this work paves the way for a new frontier of highly efficient yet powerful hybrid model architectures.

**Alternative Theoretical Foundations**  Recent research has explored novel theoretical underpinnings for SSM-based sequence modeling. Longhorn (Liu et al., 2025) reframes the SSM state update process as the closed-form solution to an online learning problem. The state transition is derived from an implicit optimization objective for associative recall, yielding a parameter-free adaptive forgetting mechanism:

$$S_{t,i} = (I - \varepsilon_{t,i} k_t k_t^\top) S_{t-1,i} + \varepsilon_{t,i} k_t x_{t,i}, \quad \text{where} \quad \varepsilon_{t,i} = \frac{\beta_{t,i}}{1 + \beta_{t,i} k_t^\top k_t}. \tag{38}$$

In a distinct research direction, Oscillatory State-Space Models (LinOSS) (Rusch & Rus, 2025) challenge the dominance of standard first-order ODEs. Inspired by cortical oscillations mechanisms in biological neural systems, LinOSS is built on a second-order ODE system formulated as $(y''(t) = -Ay(t) + Bu(t) + b)$. This system exhibits energy conservation and ensures stability under much weaker constraints (*e.g.*, $A \geq 0$), potentially unlocking greater modeling expressiveness for complex sequence data.

### 4.5  Architectural and Application Extensions: Breaking the 1D Barrier

Although SSMs were initially designed for 1D sequence modeling, recent research efforts have extended their core principles to higher-dimensional data such as images and video. ConvSSM (Smith et al., 2023a) demonstrated that a pointwise (1×1) convolution evolving over time is mathematically equivalent to an LTI SSM, thereby enabling spatial tensors $\mathcal{X}_t \in \mathbb{R}^{H \times W \times P}$ to be treated as SSM states tensors. By fusing spatial convolution with temporal recurrent computation via a parallel scan operator, ConvSSM enables efficient spatiotemporal modeling with linear computational complexity with respect to sequence length. Building upon this insight, MambaMixer (Behrouz et al., 2024) extends Mambas content-selective mechanism to multi-dimensional signals via a novel dual-mixing architecture: a selective token-mixing block that deploys bidirectional SSMs across spatial tokens, and a selective channel-mixing block that employs the identical selective mechanism across feature channels. Collectively, these extensions demonstrate how SSMs can naturally generalize beyond 1D sequence modeling and function as versatile building blocks for high-dimensional perception tasks.

## 5   Recurrent Neural Networks

Early research on linear attention emphasized its intimate connection to recurrent formulations (Katharopoulos et al., 2020), demonstrating that the kernelized accumulation of keyvalue pairs can be rephrased as a recurrence relationmathematically equivalent to a *Recurrent Neural Networks (RNNs)* with matrix-valued hidden states. An RNN can be defined as a discrete-time dynamical system parameterized by $\theta$, which maps the current inputstate pair to an output and an updated state, formulated as: $f_\theta(x_t, h_t) = (y_t, h_{t+1})$, where $x_t$ denotes the input, $h_t$ the hidden state, $y_t$ the output, and $\theta$ the set of learnable parameters. To elucidate why linear attention naturally connects to recurrent formulations, this chapter reviews the evolutionary trajectory of RNNs and highlights how their historical challenges and technical innovations have motivated the development of modern linear, gated, and associative sequence modeling architectures. The literature surveyed in this chapter is organized into four thematic sections:

1. **Vanilla RNNs and Classical Limitations**: This section covers early recurrent models (Elman, 1990; Jordan, 1986), and the fundamental challenges of vanishing/exploding gradients and sequential computational bottlenecks (Bengio et al., 1994; Hochreiter et al., 2001; Pascanu et al., 2013).

2. **Gating-based Recurrent Models**: These models stabilize training and enable long-range memory via gating mechanisms (Hochreiter & Schmidhuber, 1997; Cho et al., 2014; Qin et al., 2023b).

3. **Linear RNNs and Parallelizable Recurrences**: These models eliminate or restructure nonlinear components to enable scan-parallel computation (Blelloch, 1990; Balduzzi & Ghifary, 2016; Bradbury et al., 2017; van den Oord et al., 2016; Lei et al., 2018; Martin & Cundy, 2018), including modern instantiations (Katsch, 2024; Qin et al., 2024e; Peng et al., 2023; 2024).

4. **Fast Weight Programmers Perspective**: This section covers the development of classical fast-weight systems (Schmidhuber, 1992; 1993; Irie et al., 2022; 2023; Schlag & Schmidhuber, 2018; Irie et al., 2021), their reinterpretation as formulations equivalent to linear attention (Katharopoulos et al., 2020; Schlag et al., 2021; Sun et al., 2023b), and modern extensions (Neil et al., 2017; Sun et al., 2023b; Mao, 2022; Sun et al., 2025e; Yang et al., 2025b; Siems et al., 2025; Qin et al., 2024e; Beck et al., 2024; Dao & Gu, 2024; Schlag et al., 2021; Team et al., 2025a; Yang et al., 2025a; Sun et al., 2025e).

## 5.1 Vanilla RNNs and Their Limitations

Recurrent Neural Networks (RNNs) (Elman, 1990; Jordan, 1986) emerged in the late 1980s as one of the earliest architectures tailored for sequential data modeling. Their architecture maintains a hidden state vector $h_t$, which serves as a fixed-dimensional summary of all preceding input information. The standard RNN update rule is formulated as:

$$h_t = \sigma(W^R h_{t-1} + W^I x_t), \quad y_t = W^O h_t, \tag{39}$$

where $h_t \in \mathbb{R}^{d_{\text{out}}}$ denotes the hidden state, $x_t \in \mathbb{R}^{d_{\text{in}}}$ the input, and $\sigma(\cdot)$ a nonlinear activation function. The weight matrices $W^R \in \mathbb{R}^{d_{\text{out}} \times d_{\text{out}}}$ and $W^I \in \mathbb{R}^{d_{\text{out}} \times d_{\text{in}}}$ correspond to the recurrent and input weight parameters, respectively.

Despite its inherent simplicity and computational efficiency, this formulation is plagued by three well-documented bottlenecks: (a) **Training instability:** Specifically, Backpropagation Through Time (BPTT) is prone to vanishing and exploding gradients (Bengio et al., 1994; Hochreiter et al., 2001; Pascanu et al., 2013), thus limiting the effective range of dependency modeling. (b) **Sequential bottleneck:** Each state update depends directly on its immediate predecessor, which hinders parallelization along the sequence length dimension and limits scalability on parallel computing modern hardware. (c) **Representation bottleneck:** The fixed-dimensional hidden vector must encode the entire sequence history, leading to information interference and lossy retrieval of distant context. These limitations have spurred a series of technical innovations: gating mechanisms that enhance training stability, linear RNNs that simplify recurrence to improve parallelizability, fast weight programmers that expand hidden states into more expressive memory representations and establish direct connections to linear attention, alongside various refinements targeting the mitigation of the sequential bottleneck for parallel training.

## 5.2 Gating Mechanism

Early RNNs were plagued by the well-documented vanishing and exploding gradient problems, which hindered their ability to capture long-range sequence dependencies. This issue was effectively mitigated with the advent of *gating mechanisms*. The Long Short-Term Memory (LSTM) architecture (Hochreiter & Schmidhuber, 1997) emerged as the canonical solution, augmenting the vanilla recurrent update with three specialized gates (input, forget, and output) that modulate the flow of information within the network. Formally, the computation of an LSTM cell is defined as:

$$f_t = \sigma(W_f x_t + U_f h_{t-1} + b_f), \quad i_t = \sigma(W_i x_t + U_i h_{t-1} + b_i), \tag{40}$$

$$o_t = \sigma(W_o x_t + U_o h_{t-1} + b_o), \quad \tilde{c}_t = \tanh(W_c x_t + U_c h_{t-1} + b_c), \tag{41}$$

$$c_t = f_t \odot c_{t-1} + i_t \odot \tilde{c}_t, \qquad h_t = o_t \odot \tanh(c_t). \tag{42}$$

Among these gates, the forget gate proved critical: its adaptive control over cell state preservation stabilized model training and enabled the capture of long-range dependencies. The Gated Recurrent Unit (GRU) (Cho et al., 2014) proposed a streamlined alternative that employs only two gates: a reset gate and an update gates. The update gate serves a role analogous to the LSTMs forget gate, dynamically balancing the integration of old and new information. Both architectures demonstrate that gating mechanisms induce "slow modes" in recurrent dynamics, thereby enabling the modeling of long-term memory.

Gating mechanisms have since been extended to a diverse range of variants for modern recurrent neural networks. HGRN (Qin et al., 2023b) proposed a hierarchical gated architecture, demonstrating how gates

could be modulated across network depth to distribute memory management responsibilities: lower layers are biased toward forgetting to model short-term dynamics, while higher layers prioritize information retention to capture long-term dependencies. Griffin (De et al., 2024) has demonstrated how gating can serve as a bridge between recurrence computation and attention mechanisms: its gated units interpolate between linear recurrence operations and input-driven updates, filtering out uninformative signals while preserving salient contextual history.

## 5.3  Linear RNNs

To mitigate the sequential computational dependency, a natural intuitive approach is to eliminate nonlinearities from the recurrence update, thereby rendering it amenable to algebraic transformations for parallel training. This idea can be traced back to early investigations of *linear recurrent* formulations in neural networks for temporal smoothing and exponential moving average, and was subsequently instantiated in element-wise linear recurrences and convolutional token mixersboth of which eliminate nonlinear state updates to improve hardware computational efficiency (Balduzzi & Ghifary, 2016; Bradbury et al., 2017; Kalchbrenner et al., 2017; van den Oord et al., 2016). The core insight is that first-order linear recurrence relations can be computed via the *parallel scan* (prefix-sum) algorithm when appropriate algebraic conditions are satisfied (Blelloch, 1990), thereby enabling parallel forward and backward propagation across the sequence length dimension and yielding substantial computational speedups in practice deployments (Martin & Cundy, 2018; Lei et al., 2018). Formally, a first-order linear recurrence relation is formulated as:

$$h_t = \Lambda_t h_{t-1} + x_t, \tag{43}$$

where $\Lambda_t$ is typically a diagonal matrix (equivalently, $\Lambda_t = \text{Diag}(\lambda_t)$), reducing the recurrence relation to:

$$h_t = \lambda_t \odot h_{t-1} + x_t. \tag{44}$$

Such formulations satisfy the properties of *associativity*, *semi-associativity*, and *distributivity*, rendering them amenable to parallel scan operations (Blelloch, 1990; Martin & Cundy, 2018), while achieving performance comparable to that of standard nonlinear RNNs.

More recent research has modified linear recurrence relations to integrate the computational efficiency of parallel scans with the strong modeling capacity of deep sequence modeling architectures. The Linear Recurrent Unit (LRU) (Orvieto et al., 2023) eliminates nonlinearities from the state update process but interleaves linear recurrence operations with nonlinear projection modules (*e.g.*, MLP or GLU blocks). The linear recurrence layer is updated as follows:

$$h_t = \lambda_t \odot h_{t-1} + (1 - \lambda_t) \odot \phi(x_t), \tag{45}$$

where $\lambda_t$ denotes learnable decay coefficients and $\phi(\cdot)$ represents a pointwise nonlinear transformation. Inspired by LSTMs (Hochreiter & Schmidhuber, 1997) and GRUs (Cho et al., 2014), the RG-LRU (De et al., 2024) extends the Linear Recurrent Unit by incorporating two lightweight gating mechanisms while retaining a fully element-wise update mechanism. Its recurrence relation is defined as:

$$r_t = \sigma(W_a x_t + b_a), \quad i_t = \sigma(W_x x_t + b_x), \tag{46}$$

$$a_t = (\sigma(\Lambda))^{cr_t}, \quad h_t = a_t \odot h_{t-1} + \sqrt{1 - a_t^2} \odot (i_t \odot x_t), \tag{47}$$

where $\Lambda \in \mathbb{R}^d$ denotes a learnable decay coefficient matrix and $c$ is a fixed scalar constant. $r_t$ and $i_t$ correspond to the recurrence gate and input gate respectivelyboth of which depend solely on the current input $x_t$, thus ensuring efficient and numerically stable computation. Hierarchical architectural designs have been proposed to optimize memory allocation across network depth in linear RNNs. The HGRN (Qin et al., 2023b) model incorporates input-only gating mechanisms along with a layerwise lower bound constraint on the forget gate, enabling shallow layers to capture short-term sequence dynamics while deeper layers accumulate longer-term contextual information.

Recent theoretical work (Grazzi et al., 2025) demonstrates that constraining the state-transition eigenvalue spectrum to the interval $[0, 1]$ inherently limits the expressiveness of linear RNNs; such models cannot solve

even parity problem, and real triangular matrix structures impede modular counting tasks. Relaxing this constraint to allow negative (or complex) eigenvalues within the interval $[-1, 1]$ eliminates this limitation. In particular, products of generalized Householder matrices enable the modeling of group word problems (*e.g.*, $\mathbb{Z}_m$) andwith moderate network depthany regular language, while maintaining numerical stability (matrix norm $\leq 1$) and parallel scan efficiency.

Departing from conventional linear RNN formulations, RWKV-4 (Peng et al., 2023)inspired by AFT framework (Zhai et al., 2021)proposes an element-wise *weighted key-value* (WKV) aggregation mechanism to replace self-attention, and augments it with a lightweight *token shift* operation that interpolates $x_t$ and $x_{t-1}$ for both temporal and channel mixing processes. The WKV operator can be formulated in a recursive form as follows:

$$\mathrm{wkv}_t = \frac{a_{t-1} + e^{u+k_t} \odot v_t}{b_{t-1} + e^{u+k_t}}, \quad a_0, b_0 = 0, \tag{48}$$

$$a_t = e^{-w} \odot a_{t-1} + e^{k_t} \odot v_t, \tag{49}$$

$$b_t = e^{-w} \odot b_{t-1} + e^{k_t}, \tag{50}$$

where $(a_t, b_t)$ denote hidden state variables, $k_t, v_t$ represent the time-mixing key and value vectors, $w$ is a learnable decay coefficient, and $u$ is a positional bias term.

Beyond vector-valued hidden states, HGRN2 (Qin et al., 2024e) extends the state representation to a matrix form via outer-product updates (denoted by $\otimes$), formulated as:

$$H_t = H_{t-1} \mathrm{Diag}(f_t) + i_t^\top (1 - f_t) \in \mathbb{R}^{d \times d}, \tag{51}$$

where $f_t, i_t \in \mathbb{R}^{1 \times d}$ are the forget and input vectors, respectively, and $i_t^\top (1 - f_t)$ corresponds to an outer-product operation. This design expands the state dimensionality from $d$ to $d^2$ without increasing the number of trainable parameters, thereby enabling a significantly larger recurrent memory capacity. GateLoop (Katsch, 2024) introduces content-dependent diagonal state transition mechanisms. Its recurrence relation is defined as:

$$H_t = H_{t-1} A_t + x_t \otimes k_t, \tag{52}$$

where the hidden state is a matrix $H_t \in \mathbb{C}^{d \times d}$, updated via an input-dependent diagonal transition matrix $A_t \in \mathbb{C}^{d \times d}$ and an input-dependent gate vector $k_t \in \mathbb{C}^{1 \times d}$ applied to the input vector $x_t \in \mathbb{C}^d$. RWKV-5 (Eagle) (Peng et al., 2024) adopts a similar design philosophy, representing the hidden state as multiple matrix heads with stable decay parameterization schemes. Matrix-valued states enable richer associative dynamic behaviors, structured state updates, and multi-stream memory mechanismsideas that are central to the architectures discussed in the subsequent sections.

### 5.4   A Perspective of Fast Weight Programmers

**Classical fast weight programmers.**   The concept of using the hidden state as a *matrix-valued memory* long predates modern linear RNN architectures. *Fast Weight Programmers* (FWPs) first introduced in the early 1990s (Schmidhuber, 1992)formalize this concept by treating the hidden state itself as a dynamic *programmable associative memory.* At each time step, the model encodes a new association into a fast weight matrix $W_t$, and subsequent inputs retrieve information by querying this dynamically evolving memory. Thus, recurrence mechanism is reinterpreted as an online, content-addressable memory access process rather than a purely vector-valued state update operation.

Although initially proposed as a distinct independent framework, FWPs inherently unify a broad range of modern sequence modeling architectures. Their matrix-valued memory perspective subsumes linear RNNs with 2D hidden states (Qin et al., 2024e; Beck et al., 2024), structured SSMs (*e.g.*, Mamba2 (Dao & Gu, 2024)), and a broad family of linear attention mechanisms that dynamically maintain and update key-value associations over time (Neil et al., 2017; Sun et al., 2023b; Mao, 2022; Sun et al., 2025e; Yang et al., 2025b; Siems et al., 2025). This thus positions FWPs as a unifying theoretical perspective that bridges recurrent computation, associative memory, and linear attention mechanisms. Formally, given a sequence of inputs

$\{x_t\}_{t=1}^{T}$, an FWP performs the following computations:

$$a_t, b_t = W_a x_t,\ W_b x_t, \tag{53}$$

$$W_t = \sigma(W_{t-1} + a_t \otimes b_t), \tag{54}$$

$$y_t = W_t x_t, \tag{55}$$

where $W_a, W_b$ denote slow weight matrices. Each update step encodes the association $(a_t, b_t)$ into the fast memory matrix $W_t$, and information retrieval is achieved by applying this memory matrix to the current input $x_t$ (Schlag et al., 2021).

**Connection to linear attention.** We can reinterpret these association pairs through the lens of queries, keys, and values from modern attention mechanisms. The core insight is that $a_t$ and $b_t$ merely provide one parametrization scheme for the associations stored in fast weight matrix $W_t$, while the identical underlying mechanism can be re-expressed using standard Transformer notation. By introducing explicit query, key, and value projection layers, the slow network decouples the roles of memory writing and information retrieval: keys $k_t$ determine the storage location of information, values $v_t$ specify the content to be stored, and queries $q_t$ determine how relevant information is retrieved. When the query projection matrix $W_Q = \boldsymbol{I}_d \in \mathbb{R}^{d \times d}$ (where $\boldsymbol{I}_d$ denotes the $d$-dimensional identity matrix), the query vector $q_t$ degenerates to the raw input vector $x_t$, consistent with the vanilla FWP formulation. This notational transformation renders the equivalence between FWPs and linear Transformers fully transparent (Katharopoulos et al., 2020). At each step, the slow weight network generates the following projections:

$$q_t = W_Q x_t, \quad k_t = W_K x_t, \quad v_t = W_V x_t, \tag{56}$$

and the fast weight memory matrix is updated as follows:

$$W_t = W_{t-1} + v_t \otimes \phi(k_t), \quad y_t = W_t\, \phi(q_t), \tag{57}$$

where $\phi(\cdot)$ denotes a feature mapping function, and the activation function $\sigma$ is simplified to the identity function. This update rule encodes a keyvalue pair $(k_t, v_t)$ in the fast weight matrix $W_t$, while the readout operation generates an output vector conditioned on the query vector $q_t$.

**Updating mechanisms.** FWPs also support a diverse range of gated update rules for memory matrix refinement. The most fundamental of these is the Hebbian additive update rule in Eq. (57), which accumulates association pairs over time but is susceptible to information interferenceanalogous to a vanilla RNN without gating mechanisms. A more refined approach is DeltaNet (Neil et al., 2017), which introduces an error-corrected update rule formulated as:

$$W_t = W_{t-1} + \eta_t\big(v_t - W_{t-1}\phi(k_t)\big) \otimes \phi(k_t), \tag{58}$$

thereby encoding only the residual information that has not yet been captured by the existing memory matrix. This design directly parallels the gating mechanism of GRUs, thereby mitigating information conflicts and stabilizing the information retrieval process. Building on this insight, DeltaProduct (Siems et al., 2025) generalizes DeltaNet by performing multiple such error-corrected update steps per input token, formulated as:

$$W_t = W_{t-1}\left(\prod_{j=1}^{n_h}\big(I - \eta_{t,j}\,\phi(k_{t,j})\phi(k_{t,j})^{\top}\big)\right) + \sum_{j=1}^{n_h}\eta_{t,j}\,v_{t,j} \otimes \phi(k_{t,j}), \tag{59}$$

where $n_h$ denotes the number of error-correction update steps per input token. This yields a diagonal-plus-rank-$n_h$ update rule that achieves a trade-off between computational efficiency and model expressivity: increasing $n_h$ enhances state tracking and associative recall capabilities, while retaining the stability guarantees of products of Householder-like matrix updates. Another research direction introduces multiplicative decay mechanisms, as exemplified by mLSTM within the xLSTM (Beck et al., 2024) framework, which adopts a gated update rule of the form:

$$W_t = \text{Diag}(\lambda_t)\, W_{t-1} + (i_t \odot v_t) \otimes \phi(k_t), \tag{60}$$

where $\lambda_t$ and $i_t$ denote input-dependent forget gate coefficients and input gate vectors, respectively, which selectively discard obsolete memory content and regulate the encoding of new association pairs. In addition, mLSTM integrates other key architectural components of LSTMs (Hochreiter & Schmidhuber, 1997) into FWP framework, including a cell state, input gate, and output gate. Gated DeltaNet (Yang et al., 2025b) unifies the error-corrected delta update rule of DeltaNet and the input-driven multiplicative decay mechanism of mLSTM/Mamba2 into a single integrated update rule, formulated as:

$$W_t = \text{Diag}(\lambda_t)\, W_{t-1}\big(I - \eta_t\, \phi(k_t)\phi(k_t)^\top\big) + \eta_t\beta_t\, v_t \otimes \phi(k_t). \tag{61}$$

Here $\lambda_t$ governs the global decay rate, while $\eta_t$ performs key-specific residual error correction. This formulation enables rapid context resets when $\lambda_t$ takes small values and precise associative recall when $\lambda_t$ take large values, thereby enhancing information retrieval accuracy and long-context modeling capabilities. This mechanism has also been adopted in recent large-scale language models (*e.g.*, Qwen3-Next (Yang et al., 2025a)). Kimi Delta Attention (KDA) further extends $\lambda_t$ to a full diagonal gate matrix $\text{Diag}(\lambda_t)$, enabling more fine-grained control memory decay and positional awareness; this design has been incorporated into Kimi Linear model (Team et al., 2025a).

A series of extensions have further advanced the Fast Weight Programmer (FWP) framework. TPR-RNN (Schlag & Schmidhuber, 2018) extends FWPs to third-order tensors, replacing the fast weight matrix with a tensor that captures entity–relation–entity bindings. This enriched representation facilitates more robust compositional logical reasoning and enhances model interpretability. Self-referential architectures (Schmidhuber, 1993; Irie et al., 2022; 2023) abandon the slow-fast weight network distinction entirely—these systems directly modify their own weight matrices during the inference phase, thereby facilitating recursive self-improvement. Meanwhile, Recurrent FWPs (RFWPs) (Irie et al., 2021) incorporate temporal feedback mechanisms by routing the output of fast weight network back to the slow weight network, thus enabling multi-timescale dynamic behaviors.

The most prominent recent advancement is Test-Time Training (TTT) (Sun et al., 2025e), which fundamentally redefines the theoretical underpinnings of the FWP framework. As discussed in Sec. 3.3, TTT has evolved into a distinct research branch of linear attention mechanisms. Rather than relying on heuristic update rules, TTT frames the hidden state as a parametric model $f$ with weights matrix $W_t$, which is updated at every time step via gradient descent on a self-supervised loss function, formulated as:

$$W_t = W_{t-1} - \eta\nabla\ell(W_{t-1}; x_t). \tag{62}$$

The objective function $\ell(W; x_t)$ is typically defined as a self-supervised reconstruction loss, which forces the parametric model $f$ to predict the original input vector $x_t$ from a corrupted input variant $\tilde{x}_t$. By performing online optimization of this loss function, the hidden state dynamically adapts to capture the inherent structural patterns of the input sequence, demonstrating that FWPs implement associative memory via an implicit gradient-based adaptive mechanism.

## 5.5 Future Outlook

The scaling of recurrent neural network architectures remains a critical challenge. In the inference phase, RNNs process sequences with linear computational complexity and a constant memory footprint, with only a single state update operation required per time step. However, their training process is plagued by the dual issues of vanishing gradients and insufficient parallelizability, thus hindering their scalability to match that of attention-based models. Recent research efforts have tackled this challange from two complementary perspectives. One line of research simplifies the recurrent update mechanism by eliminating nonlinear components (*e.g.*, linear RNNs and FWPs with identity activation functions), rendering them algebraically equivalent to Linear Transformers architectures and compatible with parallel scan algorithms. The other research direction re-explores nonlinear recurrent dynamics: DEER (Lim et al., 2024) reformulates nonlinear recurrence as a fixed-point problem solved via parallel Newton update steps, while ELK (Gonzalez et al., 2024) stabilizes such iterative updates by establishing a theoretical connection between Newton's method and Kalman smoothing. Both of these approaches aim to retain the inherent computational efficiency of recurrent mechanisms while enabling training performance comparable to that of attention-based and state-space models.

# 6 Hybrid Architectures

Hybrid architectures can be read through the unified write/read memory view of Sec. 2. A linear-time module maintains a fixed-size state that is cheap to update and read but compresses history, which limits exact recall, whereas full softmax attention reads over the entire sequence exactly at quadratic cost. A hybrid therefore pairs a linear-time memory backbone with a limited amount of token-level softmax attention, and the three families below differ mainly in how that attention is allocated. Block-level Transformer–SSM hybrids interleave whole attention blocks with SSM or gated-recurrent blocks across depth; attention-level hybrids stay within a single Transformer framework and mix attention types, keeping a few softmax layers or heads among many linear or Delta-style ones; and post-hoc methods start from a full-attention model and decide which attention components to retain, approximate, or replace with linear-time modules. Reading the design space this way makes these otherwise separate lines of work easier to compare, and helps explain why they tend to share a common failure mode, the loss of fine-grained detail, together with a common remedy, a small and well-placed amount of exact attention. How much exact attention to keep, and where to place it, is the main design lever that large-scale models tune, and we survey these choices below.

The pursuit of computational efficiency has driven a paradigm shift in the architectural landscape of large language models (LLMs), shifting from the quadratic computational complexity of Transformers to linear-time computational alternatives such as State Space Models (SSMs) (Gu & Dao, 2024) and various linear attention mechanisms. While these $O(N)$ architectures successfully break through the scaling bottlenecks for long-context sequence modeling tasks, they inherently introduce new performance trade-offs: by design, purely linear-time models often act as lossy information compressors (De et al., 2024; Lenz et al., 2025), which can degrade high-fidelity, retrieval-intensive capabilities that rely on precise, token-level interactions.

This has led to a growing research consensus that the ultimate goal is not to abandon self-attention, but to deploy it in a targeted manner in tandem with more efficient sequence modeling operators. Hybrid architectures epitomize this design principle: they strive to reach a new Pareto frontier of performance and computational efficiency by integrating architectural components with complementary strengthsleveraging quadratic self-attention as a high-fidelity token retrieval and alignment mechanism, while delegating most long-range sequence memory and inference throughput tasks to linear-time modules. At the industrial scale, this paradigm has already been instantiated in models such as Jamba (Lenz et al., 2025), Nemotron-H (NVIDIA et al., 2025), MiniMax-01 (MiniMax et al., 2025), Kimi Linear (Team et al., 2025a), and Qwen3-Next (Qwen, 2025).

Broadly speaking, the existing literature on hybrid architectures can be categorized into the following three families:

1. **Block-level Transformer-SSM Hybrids**: These architectures interleave or integrate full-attention Transformer blocks with SSM or gated recurrent blocks across network depth or attention heads, where attention layers provide high-fidelity global context retrieval and refinement on top of a linear-time memory backbone (Lenz et al., 2025; Ren et al., 2025a; NVIDIA et al., 2025; Dong et al., 2025; Glorioso et al., 2024; De et al., 2024; Li et al., 2025e; Zuo et al., 2025; Bae et al., 2025).

2. **Attention-level Hybrids**: These models adhere to the Transformer architectural framework but replace the majority of softmax self-attention modules with linear or Delta-style attention mechanisms, incorporating a small number of full-attention layers or heads to periodically "refresh" fine-grained token-level interactions (MiniMax et al., 2025; Team et al., 2025b;a; Qwen, 2025).

3. **Post-hoc Hybridization of Pre-trained Transformers**: These methods start with a well-trained full-attention Transformer model and either distill it into a linear-time or hybrid student model or directly convert its attention weights into recurrent/SSM parameters, thereby preserving pre-trained knowledge while modifying the underlying inference engine (Mercat et al., 2024; Bick et al., 2024; Wang et al., 2024b; Lan et al., 2025; Zhang et al., 2025b; Kasai et al., 2021).

In the remainder of this section, we adopt this classification framework: we first elaborate on the architectural blueprints for block- and attention-level hybrids, then summarize practical post-hoc conversion methods and the key challenges encountered in large-scale industrial deployments.

### 6.1 Architectural Blueprints for Hybrid Models

The integration of diverse sequence modeling paradigms has spawned a suite of architectural design strategies, each leveraging the complementary strengths of its constituent components. The overarching objective is to leverage the computational efficiency of linear-time modules for long sequences processing, while harnessing quadratic self-attention to boost advanced capabilities including high-fidelity token retrieval and complex logical reasoning.

#### 6.1.1 Transformer-SSM Hybrids

A prominent hybridization approach entails the seamless fusion of Transformer and SSM blocks, which is rooted in the principle of functional specialization. SSM layers, characterized by their linear computational complexity, efficiently compress contextual information across lengthy input sequences, whereas Transformer layers execute more compute-intensive yet higher-fidelity global feature refinement operations.

A core architectural strategy is vertical stacking, where Transformer and SSM blocks are arranged in an alternating hierarchical sequence. This design enables the model to periodically apply the global contextual reasoning capability of self-attention to the context information efficiently compressed by the SSM layers. A seminal example is Jamba (Lenz et al., 2025), which pioneered a heterogeneous architecture that alternates layers of self-attention, Mamba, and MLP blocks to a carefully calibrated ratio. This architectural design directly alleviates the on-device hardware memory bottleneck, enabling a 12B parameter Jamba model to be deployed on a single 80GB GPU. The architecture has since been further refined in Jamba-1.5 (Team et al., 2024), whose capabilities are enhanced by integrating a Mixture-of-Experts (MoE) mechanism (Jacobs et al., 1991) into its MLP layersthis enables a substantial increase in active parameters during inference without a commensurate rise in computational overhead. Similarly, Samba (Ren et al., 2025a) proposes a Single-Stack Memory Block that alternates between Mamba and multi-head self-attention layers, achieving a favorable balance between memory efficiency and modeling capacity. The Nemotron-H (NVIDIA et al., 2025) model family adopts a relatively straightforward architectural pattern where Transformer and Mamba blocks are regularly interleaved, ensuring that input information is consistently processed through both modeling paradigms.

Another representative approach is the Parallel Heads strategy, where distinct modeling mechanisms operate concurrently within the same network layer. This strategy is exemplified by Hymba (Dong et al., 2025), which proposed a novel hybrid-head architecture inspired by multi-head self-attention mechanism. Within a Hymba layer, a subset of heads are standard self-attention heads, while the remainder are Mamba headsboth types of heads process the identical input tensor simultaneously. This design enables attention heads to focus on high-resolution token retrieval tasks, while Mamba heads efficiently aggregate contextual informationthis resolves the limitation of having to make a layer-level trade-off between the two mechanisms. To further enhance parameter efficiency, a Shared Attention mechanism is incorporated into the design. Zamba (Glorioso et al., 2024) presents a lightweight compact hybrid model that combines a Mamba backbone with a single shared self-attention module, which is deployed sparingly across the network. This architectural design delivers the crucial global context refinement capability of self-attention at a minimal parameter overhead.

Beyond simple block-wise alternation strategies, recent research efforts have explored more tightly integrated hybrid architectural designs. A notable example is the Griffin (De et al., 2024) architecture, which integrates linear recurrent mechanisms with sliding-window-based local attention within its residual blocks. In this model, a gated linear recurrent block maintains and updates a compact hidden state in a token-wise manner, providing an $O(N)$-time, constant-memory pathway for long-range context propagation, which is analogous to SSM/Mamba-style sequence modelswhile a subsequent residual block refines the feature representations via sliding-window-based local attention. TransMamba (Li et al., 2025e) proposes a flexible Transformer-Mamba hybrid architecture that operates at both the layer and sequence levels, which is grounded in the consistency between QKV matrices and CBx matrices.

Several recent studies have begun to treat architectural hybridization as a general design principle rather than an ad hoc palliative measure tailored to specific model instances. Falcon-H1 (Zuo et al., 2025) proposes a hybrid-head architecture where full self-attention heads and state-space/Mamba-style recurrent heads

operate in parallel within the same network blockrather than placing Transformer and SSM layers in separate sequential stages. This intra-layer (parallel) fusion scheme is scaled from sub-billion to tens-of-billions of parameters; for instance, Falcon-H1-34B achieves performance compared to that of substantially larger (70B-scale) pure-Transformer baselines, while supporting $\sim 10^5$-token contexts and delivering higher inference efficiency. Bae et al. (2025) adopt a more holistic perspective, contrasting two primary fusion paradigms: (i) inter-layer/sequential fusion, where Transformer-style attention blocks alternate with SSM or gated linear recurrent blocks (*e.g.*, Transformer-Mamba stacks); and (ii) intra-layer/parallel fusion, where attention-style and SSM-style heads are integrated within a single network layer (*e.g.*, Falcon-H1). This study quantifies the trade-offs between these two paradigms across multiple dimensions: language modeling performance, long-context retrieval accuracy, scaling behavior, and both training and inference costs, while providing practical guidance on the optimal proportion of quadratic attention to retain and its optimal placement within the network. Ring-linear models (Team et al., 2025b) draw a related conclusion for long-context reasoning tasks: by interleaving predominantly linear/Delta-style attention with occasional full softmax attention, their Ring-linear-2.0 models maintain near-linear memory and I/O costs during inference, while preserving fine-grained retrieval and multi-step reasoning capabilities, which tend to degrade in purely linear-time model stacks. Collectively, these findings reveal a universal design pattern: full self-attention is treated as a scarce, high-fidelity retrieval and alignment mechanism that is deployed sparingly, whereas SSM- or linear-attention-style modules provide scalable long-range memory capacity and inference throughput.

### 6.1.2 Transformer–Linear Attention Hybrids

A second category of hybrids architectures does not integrate different model families; instead, it combines multiple attention mechanisms within the Transformer architectural framework. The core objective is to retain the $O(N^2)$ computational flexibility of softmax self-attention in scenarios where precise token-to-token interactions are critical, while replacing most layers with $O(N)$ linear or Delta-style attention mechanisms to reduce memory and I/O overhead. Purely linear attention stacks tend to act as lossy information compressors, which degrade fine-grained retrieval and instruction following capabilities; these hybrid architectures explicitly treat full self-attention as a scarce resource that should be strategically allocated across network depth.

A representative example is MiniMax-01 (MiniMax et al., 2025), which constructs nearly all of its layers on Lightning Attentiona highly optimized, I/O-aware linear attention kerneland inserts a standard softmax self-attention block after every seven consecutive Lightning Attention layers. This 1:7 layer schedule ratio is determined via ablation experiments on long-context retrieval and instruction-following benchmarks: an insufficient number of full-attention layers fails to restore high-fidelity retrieval performance, whereas more frequent insertion yields diminishing returns in model performance while causing quadratic growth in KV-cache and memory traffic overhead. In essence, these sparse softmax layers act as periodic high-fidelity "refresh" steps that reconstruct precise token-to-token associations based on the compressed linear-attention feature state. With this design, MiniMax-01 scales training to million-token context lengths and achieves near-linear inference cost, while maintaining performance competitive with that of pure-Transformer baselines of comparable parameter sizes.

Similar scheduling principles have been adopted in recent industrial-grade models. Kimi Linear (Team et al., 2025a) replaces most softmax self-attention layers with Kimi Delta Attention (KDA)a gated Delta-style linear attention mechanismwhile retaining a small fraction of full Multi-Head Latent Attention (MLA) layers; Qwen3-Next (Qwen, 2025) combines standard gated self-attention with Gated DeltaNet-style attention mechanisms at a fixed ratio across network depth. In both cases, these models rely on expressive linear or Delta-based modules for the majority of long-context processing tasks, while preserving a small budget of quadratic self-attention to preserve high-precision retrieval and alignment capabilities. This demonstrates how attention-level hybrid architectures can achieve performance comparable to that of full self-attention models, while benefiting from substantially lower KV-cache requirements and higher inference throughput.

## 6.2 Post-Hoc Hybridization: Converting Pre-trained Transformers

An alternative paradigm for building hybrid models focuses not on designing a new architecture from scratch, but on transforming existing pre-trained Transformer models into more efficient linear-time inference architectures. This "full-to-linear" conversion strategy aims to preserve the extensive knowledge base and strong performance of well-established models (*e.g.*, Llama), while retrofitting them with the low-latency and memory-efficient properties of RNNs or SSMs. The core motivation behind this strategy is to accelerate inference latency without the need for expensive pre-training of an entirely new hybrid architecture.

One core technique is knowledge distillation. In this paradigm, a lightweight, efficient linear-time "student" model (*e.g.*, an SSM or RNN-based architecture) is trained to replace the output distributions of a large-scale pre-trained Transformer "teacher" model (Kasai et al., 2021). This process transfers the teacher model's capabilities to the student model. For instance, Mercat et al. (2024); Bick et al. (2024) demonstrate how a Mamba-based student model can be distilled from a Llama-family teacher model, inheriting its strong contextual language understanding capabilities while achieving significant speedups in autoregressive decoding latency. Wang et al. (2024b) further advances this paradigm by developing a hybrid student model that retains a small number of attention layers, effectively distilling a full Transformer architecture into a more efficient hybrid variant.

A more direct approach is architectural conversion (or linearization), where the weights of the pre-trained self-attention mechanisms are directly converted or mapped to initialize the parameters of a linear-time counterparts. This approach circumvents the need for a full distillation training phase. Methods like Liger (Lan et al., 2025) and LoLCATs (Zhang et al., 2025b) explore how to approximate the softmax self-attention matrix using structured low-rank approximations that can be reformulated as linear recurrent relations. Lan et al. (2025) specifically focuses on transforming Transformer layers into gated recurrent structures or Mamba blocks, which is often followed by a short fine-tuning phase to recover performance degradation incurred during the conversion process. An early exploration of this concept is presented in (Kasai et al., 2021), which laid the foundational groundwork for these modern techniques.

These methods represent a powerful, pragmatic paradigm for efficient LLM deployment, effectively yielding a "knowledge-architecture hybrid" where the knowledge is from a pre-trained Transformer and the inference engine is linear-time architectureoffering a compelling trade-off between preserving high-fidelity pre-trained model quality and achieving operational efficiency.

## 6.3 Representative Industrial Examples

In this section, we present several representative industrial-grade implementations to illustrate the diverse architectural design of hybrid models.

**Jamba Series** The Jamba series (Lenz et al., 2025; Team et al., 2024) adopts a structured hybrid architecture that interleaves standard Transformer self-attention layers, Mamba state-space layers, and Mixture-of-Experts (MoE) MLP blocks. The original Jamba model adopts a fixed 1:7 ratioone standard self-attention layer followed by seven Mamba layersenabling a 52B-parameter model to be deployed on a single 80GB GPU, with only 12B active parameters and a compact 4GB KV cache footprint. To maximize model capacity, every other layer replaces the dense MLP block with an MoE layer where only the top-2 experts are activated per token. Jamba-1.5 scales this architectural blueprint to 398B parameters (94B active), while retaining the identical layer scheduling rhythm.

Crucially, the integration of periodic self-attention layers serves a critical functional purpose beyond mere computational efficiency. Empirical results demonstrate that pure Mamba architectures exhibit suboptimal performance on in-context learning (ICL) tasks that demand strict format compliance (*e.g.*, labels quoting). Mechanistic analysis reveals that the hybrid self-attention heads function as "induction heads" (Olsson et al., 2022), enabling the model to retrieve and replicate exemplar labels from the input prompt-capabilities that are often diminished in purely recurrent dynamic systems. Thus, the hybrid architecture successfully recovers Transformer-level ICL performance, while retaining the high inference throughput and long-context modeling capabilities (up to 256K tokens) of SSMs.

**Nemotron-H**  The Nemotron-H family (NVIDIA et al., 2025) adopts a structured hybrid architecture that strategically replaces the majority of self-attention layers with more computationally efficient Mamba-2 (Dao & Gu, 2024) layers. Unlike the fixed, heterogeneous block structure of the Jamba series, Nemotron-H adopts a design where approximately 8% of the total layers are self-attention layers, evenly distributed across the network depth. The architecture is structured as a repeating three-stage modular framework: an initial segment comprising several Mamba-2 and FFN blocks; a main body consisting of modules that begin with a Mamba-2 layer, followed by a standard Transformer block (self-attention plus FFN), and additional Mamba-2 and FFN blocks; and a final Mamba-2 and FFN block. The number of such modules scales with model size, yielding a predictable yet highly effective architecture that balances the global context modeling capabilities of self-attention with the computational efficiency of state-space models.

In addition to 8B and 56B models, Nemotron-H-47B is derived from the compression of the 56B model via MiniPuzzlea two-stage distillation pipeline that integrates Minitron (Muralidharan et al., 2024) and Puzzle (Bercovich et al., 2025)achieving nearly 300× token savings compared with training from scratch. A conditional neural architecture search (NAS) first evaluated the importance of layers and neurons in the 56B teacher model and generates a pool of hardware-compatible candidate architectures (*e.g.*, architectures deployable on a 32GB GPU); a short "lightweight" distillation run then selects the optimal candidate, which is further refined via an extended knowledge distillation phase to recover performance degradation, ultimately yielding the final Nemotron-H-47B model.

The Nemotron-H series further demonstrates that hybrid architectures can serve as practical backbones for post-training and reasoning tasks. According to NVIDIA, these Mamba-2-dominant hybrid modelsdespite replacing most self-attention layers with state-space components and distilling the 56B model into smaller variantsremain fully compatible with reinforcement learning (RL)based reasoning and alignment tuning pipelines. In practice deployments, the hybrid backbone supports multi-step reasoning optimization and policy-based post-training without reverting to a full Transformer architecture, while maintaining higher inference throughput and significantly smaller KV cache footprints than comparable pure-Transformer baseline models (NVIDIA et al., 2025).

**Qwen3-Next**  The Qwen3-Next (Qwen, 2025) architecture is designed to scale both context length and total parameter count, while maximizing both training and inference efficiency. It introduces a hybrid attention mechanism that strategically integrates Gated DeltaNet (Yang et al., 2025b) with standard gated self-attention (Qiu et al., 2025) at a 3:1 ratio, leveraging the former's strong in-context learning capabilities for most layers and retaining the latter's high-precision recall capabilities for the remaining layers. These attention blocks are further enhanced with output gating mechanisms to improve numerical stability, and with expanded head dimensions paired with partially applied rotary position encodings (RoPE) to enhance extrapolation to longer sequence lengths. This attention backbone is complemented by an ultra-sparse Mixture-of-Experts (MoE) architecture with hundreds of experts, of which only a small subset (plus one shared expert) is activated per tokenensuring that only a small fraction of the model's total parameters are active during inference.

To ensure model robustness, Qwen3-Next incorporates stability-oriented designs strategies, such as zero-centered RMS normalization with weight decay applied to norm weights and normalized MoE router parameters. Finally, it integrates a native multi-token prediction (MTP) mechanism, optimized via multi-step training, to accelerate inference through speculative decoding and to further boost overall model performance. Collectively, these design choices demonstrate how attention-level hybrid architectures can be combined with sparse MoE and MTP mechanisms to maximize efficiency, while maintaining strong performance on both short- and long-context benchmark tasks.

**Kimi Linear**  Kimi Linear (Team et al., 2025a) presents an industrial-scale implementation of an Attention-Linear hybrid architecture. It replaces the majority of layers with Kimi Delta Attention (KDA)a lightweight gated Delta-style linear module with learnable gating and low-rank state parameterizationwhile retaining a small fraction of Multi-Head Latent Attention (MLA) layers. Adhering to the scheduling principles for efficient long-context processing, KDA layers form a linear-time computational backbone, while sparse MLA

layers act as high-precision retrieval anchors. Notably, the MLA blocks do not adopt RoPE (Su et al., 2024), as position information is inherently encoded by the linear attention layers.

When evaluated under matched training protocols, the Kimi Linear model family achieves performance comparable to that of full-MLA baseline models on both standard and million-token benchmark tasks. It significantly reduces memory bandwidth consumption and decoding latency, providing a concrete case study demonstrating that expressive linear moduleswhen combined with a carefully allocated budget of full self-attentioncan close the performance gap with pure Transformer models in terms of retrieval accuracy and instruction-following capabilities.

## 6.4 Limitations and Open Challenges of Hybrid Architectures

While hybrid architectures provide a promising pathway for balancing computational efficiency and model performance, their composite architectural nature introduces unique limitations that are far less pronounced in monolithic models. In particular, practitioners must carefully manage the trade-off between long-context modeling scalability and high-fidelity information retrieval, while ensuring stable optimization when integrating architectural components with very drastically different computational graphs and numerical behavior characteristics.

### 6.4.1 Balancing Long-Context Scalability and High-Fidelity Retrieval

A core challenge in hybrid models lies in managing the inherent tension between the information-compressive nature of SSMs or linear attention mechanisms and the high-precision retrieval capability of softmax self-attention. Linear-time components attain computational efficiency by aggregating and summarizing historical contextual information, which may lead to the loss of fine-grained details and degraded performance on "needle-in-a-haystack" retrieval evaluations. Hybrid architectures alleviate this issue by leveraging softmax self-attention as a specialized "retrieval expert" module: as demonstrated in models such as Jamba (Lenz et al., 2025) and MiniMax-01 (MiniMax et al., 2025), periodically inserting self-attention layers enables the model to re-access the full sequence information space and construct a high-resolution contextual index of the input sequence. These self-attention layers act as powerful corrective modules, ensuring that critical contextual details are not lost during information compression and delivering superior long-context retrieval performance compared to pure linear model stacks.

More recent models indicate that this trade-off can be pushed to a more optimal frontier. Kimi Linear (Team et al., 2025a) and Qwen3-Next (Qwen, 2025) employ expressive Delta-style or Gated DeltaNet-style linear attention modules for most layers, while reserving a small budget of full self-attention layers (*e.g.*, MLA or gated self-attention layers) to enhance retrieval accuracy and alignment quality. When evaluated under matched training protocols, these models can match or even outperform pure-Transformer baseline models on long-context perplexity and retrieval benchmark tasks, while benefiting from substantially smaller KV cache footprints and higher inference throughput. At the same time, these models also reveal a new limitation: the effective operating regime of a specific hybrid layer scheduling strategy (*e.g.*, linear-to-full attention layer ratio, chunk size, and gating configuration) is often narrow and highly dependent on both data distribution and target task characteristics, rendering it non-trivial to transfer a successful architecture and training recipe across different model scales or application domains without extensive re-tuning.

### 6.4.2 Training and Optimization Stability

Integrating architectural blocks with fundamentally distinct computational graphs and learning dynamics also presents non-trivial optimization challenges. Transformer blocks feature high parallelizability and well-characterized gradient flow properties, whereas SSM or recurrent blocks propagate information via sequential state transitions, introducing distinct numerical stability challenges. In principle, such architectural heterogeneity may give rise to exploding or vanishing activations values and unstable training dynamics. In practice, leading hybrid models such as Jamba (Lenz et al., 2025) and Griffin (De et al., 2024) demonstrate that standard training protocol (*e.g.*, AdamW optimizer paired with residual connections and layer normalization) exhibit surprisingly strong robustness: simple layer alternation patterns and per-block normalization strategies effectively mitigate many of these issues.

Newer hybrid model backbones, however, introduce additional sources of instability. Qwen3-Next (Qwen, 2025) integrates hybrid attention mechanisms with ultra-sparse MoE architectures, necessitating the adaption of zero-centered RMS normalization, explicit weight decay applied to normalization parameters, and router regularization strategies to avoid training divergence when handling extremely long contexts and performing reinforcement learning (RL)-style post-training. Kimi Linear (Team et al., 2025a) relies on specialized Kimi Delta Attention (KDA) computation kernels and chunk-wise diagonal-plus-low-rank state update mechanisms, whose numerical behavior must be meticulously controlled under low-precision training regimes to prevent parameter drift when processing million-token sequences. These examples indicate that, although hybrid models can often be trained using standard optimizers, their training stability increasingly hinges on system-level co-design of architecture, computation kernels, and normalization strategies, as well as dedicated protocols for downstream stages such as reasoning-focused RL and alignment tuning.

Read across the two subsections above, the three families tend to expose different bottlenecks. Attention-level hybrids with a fixed schedule (the 1:7 of MiniMax-01 (MiniMax et al., 2025), the 3:1 of Qwen3-Next (Qwen, 2025)) are most exposed to the narrow operating regime discussed above, since the small budget of exact attention makes its placement decisive, which motivates searching over the schedule. Expressive linear modules (Kimi Delta Attention (Team et al., 2025a), Gated DeltaNet (Yang et al., 2025b)) instead carry more of the optimization-stability burden, especially when paired with ultra-sparse MoE, and are commonly addressed through kernel- and normalization-level co-design; by contrast, block-level Transformer–SSM stacks with simple alternation (Jamba (Lenz et al., 2025), Griffin (De et al., 2024)) have often been reported to train robustly under standard recipes. Several reports also observe a gap relative to pure Transformers on the hardest reasoning benchmarks, which appears to narrow primarily through training—distillation and RL-style post-training—rather than through architecture, a point we take up in the following outlook.

## 6.5 Future Outlook

Hybrid sequence modeling architectureswhether integrating Transformers with State Space Models (SSMs) or fusing distinct linear attention mechanismsrepresent a critical evolutionary milestone in large language model architectural design. These architectures provide a pragmatic solution to the escalating computational capability demands and stringent constraints of real-world deployment scenarios. By adopting heterogeneous, functionally specialized structures instead of monolithic designs, this paradigm paves the way for the development of more powerful, computational efficient, and scalable AI systems.

Looking forward, research on hybrid models is poised to prioritize enhanced architectural adaptivity and structural refinement. Future research efforts may explore dynamic, input-aware routing mechanisms across heterogeneous architectural components, alongside the integration of novel functional modules (*e.g.*, retrieval mechanisms or explicit memory components). Nevertheless, a critical performance gap persists in current hybrid model designs. Despite their distinct advantages in memory footprint efficiency, computational throughput, and long-context modeling scalability, current hybrid models still fall short of leading pure-Transformer systems on highly challenging multi-step reasoning benchmarks. This indicates that merely relying on architectural modifications is insufficient: hybrid model backbones will need to adopt tailored training paradigmsincluding targeted chain-of-thought knowledge distillation from leading Transformer teacher models, RL-driven reasoning and alignment tuning protocols, and training curricula designed to explicitly enhance long-horizon planning capabilitiesto fully bridge the reasoning performance gap while retaining their computational efficiency advantages.

Taken together, these developments suggest a broader empirical principle: the field of hybrid models is still rapidly evolving, with no stable consensus on a single optimal formulation. In this setting, architectural progress is driven primarily by empirical scalability—a design is convincing only once it has been stress-tested across sequence length, parameter scale, and compute budget under reproducible conditions, since the narrow operating regimes noted above mean that small-scale wins often fail to transfer. Scalability in this sense is best treated as a primary evaluation axis for new hybrids, on par with perplexity and retrieval accuracy, rather than as an afterthought left to deployment.

# 7 Applications

Linear attention is adopted across downstream domains for two reasons: in perception domains the input is too large to attend over in full, and in sequence domains the history is too long to keep in an explicit cache. In perception domains, the costly step is not the modeling itself but doing it over the full input. A high-resolution image, an uncropped satellite scene, an hour-long video, or a 3D scan easily runs to hundreds of thousands of tokens; a single $64^3$ volume alone is on the order of $2.6 \times 10^5$ tokens, at which scale softmax attention becomes infeasible. The usual workarounds, such as downsampling, cropping into patches, sliding windows, or token compression, often weaken or approximate the global context the task needs, whereas a linear-complexity model can process the full-resolution image, the whole scene, or the long video directly while still mixing it globally. In sequence domains such as NLP and time series, the input is usually not as large but the history is long, and the decisive property is instead carrying that history in a fixed-size recurrent state with content-selective gating; audio sits between these two regimes, since its spectrograms and waveforms can themselves grow long enough that the input size, and not only the history, becomes the bottleneck. This distinction also accounts for several design patterns that recur below. Since the gain in perception domains is a global receptive field at linear cost, vision and 3D models tend to use bidirectional Mamba-style scans, as images have no causal order, and the main design choice becomes the scanning or serialization order. Since the fixed state is compressive and tends to lose fine detail, dense or detail-sensitive tasks such as restoration, segmentation, and high-fidelity generation usually pair the linear backbone with local convolution or a small amount of softmax attention. In sequence domains, the same compressive state favors gated and selective variants over purely additive linear attention, and the clearest gains appear on long-context rather than short-context tasks. We therefore organize this section by domain, and the major domain subsections open with the domain-specific version of this argument before reviewing representative work. We regard this task-grounded account of why linear attention pays off in each domain as complementary to, and as important as, the algorithm-level analysis of the preceding sections: it is what lets a practitioner judge where the efficiency gain becomes a deployable advantage. We survey these domains in turn:

1. **Language and Text Intelligence**: Applications in this domain leverage the models' efficiency in long-contexts processing to advance natural language understanding and generation capabilities. Core tasks include machine translation, long-document analysis and processing, text style transfer, and the enhancement of in-context learning (ICL) capabilities (Pitorro et al., 2024; DeGenaro & Lupicki, 2024; Zeng et al., 2025; Sarem et al., 2024; Zhang et al., 2024d; Xu, 2024; Do et al., 2025; Meng et al., 2025; Grazzi et al., 2024; Lu et al., 2023a).

2. **Vision and Multimodal Perception**: Methods in this field apply the powerful spatio-temporal modeling capabilities of linear attention to visual signals across multiple domains. Applications cover low-level vision tasks (*e.g.*, image restoration), visual content generation (including images and videos), 2D/3D visual understanding, medical image analysis, remote sensing, and audio-visual multimodal learning (Zheng & Wu, 2024; Zou et al., 2024; Wang et al., 2025c; Guo et al., 2024; Hu et al., 2024; Li et al., 2025c; Chen et al., 2024d; Xing et al., 2024; Wang et al., 2024a; Chen et al., 2024a; Li et al., 2024b; Jiang et al., 2024; Wang et al., 2024e; Zhao et al., 2024b; Chen et al., 2024c; Gong et al., 2025; Li et al., 2025d).

3. **Audio and Speech Processing**: This domain focuses on modeling one-dimensional temporal audio signals to support a wide range of auditory tasks. Applications include audio representation learning and classification, speech processing and enhancement, speaker separation and diarization, audio localization, and specialized auditory task modeling (Erol et al., 2024; Lin & Hu, 2024; Yadav & Tan, 2024; Shams et al., 2024; Sui et al., 2024; Zhang et al., 2025e; Gao & Chen, 2024; Avenstrup et al., 2025; Li et al., 2024a; Kuang et al., 2025; Xiao & Das, 2025).

4. **Time Series Analysis**: As a fundamental domain for sequence modeling, applications in this field specifically target real-valued temporal data across multiple disciplines. Research directions includes the development of foundational time series models, the tackling of multivariate and long-term forecasting challenges, and the enhancement of sequence modeling performance via bidirectional

processing and graph-based modeling approaches (Wang et al., 2025f; Patro & Agneeswaran, 2024a; Ma et al., 2024a; Liang et al., 2024; Zou et al., 2025; Ma et al., 2024d; Feng et al., 2025; Wu et al., 2024; Zhao et al., 2024a).

## 7.1 Natural Language Processing

In NLP the input is not spatially large but the history is long, so the cost falls on the KV cache and on quadratic attention over many tokens. What decides quality here is therefore associative recall (Arora et al., 2024a;c): keeping the few tokens that determine a translation, a retrieved passage, or a summary, which is exactly where purely additive linear attention is weakest and where gated and selective variants help. Because some recent large-scale linear and hybrid models have reported competitive results on general benchmarks such as MMLU and MATH without a distinctive advantage there, we focus below on the long-context tasks where the efficiency gain is decisive: machine translation, long-text processing and retrieval, and text generation.

### 7.1.1 Machine Translation

In the field of machine translation, recent research has rigorously evaluated the effectiveness of SSMs (*e.g.*, Mamba). Pitorro et al. (2024) conducted a comprehensive empirical study, benchmarking these SSM-based models against well-established Transformer-based architectures. Their findings indicate that while SSMs deliver notable computational efficiency gains, their performance is highly dependent on specific task characteristics and experimental setting. Further investigations into low-resource scenarios were conducted by (De-Genaro & Lupicki, 2024), who explored Mamba-based sequence modeling and fine-tuning strategies for multilingual translation tasksdemonstrating the potential applicability of such models even with limited training data. The integration of SSMs with other attention mechanisms is also being explored, as exemplified by (Zeng et al., 2025), who proposed SWAMambaa novel hybrid framework combining Sliding Window Attention (SWA) with Mamba for a translation-related specific prediction task.

### 7.1.2 Long-Text Processing and Information Retrieval

Beyond machine translation, the linear computational advantages of SSMs are being leveraged for **long-text processing** tasks. Sarem et al. (2024) specifically addressed the challenge of long-text classification by adopting a Mamba-based model, demonstrating performance improvements over traditional models that struggle to model long-context dependencies effectively. Similarly, in the field of information retrieval, Zhang et al. (2024d) proposed MambaRetrievera dedicated retriever that utilizes the Mamba architecture for dense retrieval tasks. Their work demonstrates that SSM-based models can achieve performance comparable to that of Transformer-based retrievers, while delivering superior computational efficiency. This finding is further complemented by the work of (Xu, 2024), who conducted comprehensive benchmarking of Mamba's performance on document ranking tasks.

### 7.1.3 Text Generation and Style Transfer

The domain of text generation and style transfer has also witnessed notable innovations. Do et al. (2025) proposed a discrete diffusion language model tailored for efficient text summarizationan approach that aligns with the pursuit of more efficient generation paradigms. Furthermore, Meng et al. (2025) proposed a hybrid Mamba-Transformer unsupervised framework for text style transfer tasks, demonstrating that SSMs can be effectively integrated with other architectural paradigms to leverage the complementary strengths of both.

## 7.2 Computer Vision

Vision faces a specific dilemma: convolutions are efficient but have a limited receptive field, whereas self-attention gives a global receptive field at a cost that grows quadratically with the number of patches and so becomes prohibitive at high resolution. Linear-complexity token mixers, whether kernel or feature-map attention (Katharopoulos et al., 2020) or bidirectional Mamba scans (Gu & Dao, 2024) (all linear attention in the sense of Sec. 2), are adopted because they offer a global receptive field at linear cost.

### 7.2.1 Image Classification

Here the obstacle is specific: early attempts to linearize ViT attention underperformed because naive linear attention loses focus and feature diversity. Later designs, such as focused (Han et al., 2023) and agent (Han et al., 2024b) attention and Mamba-style scans (Gu & Dao, 2024; Han et al., 2024a), largely close this accuracy gapin some cases surpassing softmax attentionwhile keeping linear cost. Early studies explored kernel approximation techniques and covariance transformation methods to achieve linear computational complexity. Lu et al. (2021) proposed a softmax-free Transformer architecture that leverages Gaussian kernel functions and low-rank matrix decomposition. Ali et al. (2021) proposed a Cross-Covariance Attention mechanism that operates on feature channels instead of tokens, thereby achieving linear complexity scaling. Shen et al. (2021) reformulated self-attention as a series of linear operations via feature dimension interactions, while Jeevan & Sethi (2021) conducted a systematic comparison of various linear attention variants for computer vision tasks.

Recent advances have focused on hybrid architectural designs and structural refinements. Sun et al. (2023a) integrated locality priors to simplify attention computation processes. Han et al. (2023) proposed a Focused Linear Attention mechanism that preserves critical visual information via channel-wise weighting strategies. Han et al. (2024b) integrated softmax and linear attention mechanisms via agent tokens, achieving a favorable balance between computational efficiency and feature representation capacity. The theoretical connection between state space models and linear attention was demystified by Han et al. (2024a), which revealed how selective state space mechanisms can be mathematically reformulated as linear attention operations.

These linear attention methods have demonstrated impressive efficiency-accuracy trade-offs on the ImageNet classification benchmark, often matching or even outperforming standard Transformer models while substantially reducing computational overhead. The evolution from pure approximation-based methods to sophisticated hybrid architectures underscores the growing maturity of linear attention-based approaches, rendering them increasingly practical for real-world image classification deployments in resource-constrained computational environments.

### 7.2.2 Low-Level Vision

Restoration tasks (dehazing, deraining, desnowing, super-resolution) must reconstruct fine detail using context from across the whole image at full resolution, which is exactly where the receptive-field-versus-cost dilemma bites: CNNs are local and Transformers are quadratic. SSM backbones such as MambaIR (Guo et al., 2024) are adopted because they activate all pixels with a global receptive field at linear cost, typically adding local convolution so that nearby-pixel detail is not lost.

Zheng & Wu (2024) proposed a U-shaped Vision Mamba (UViM) architecture that integrates Mamba blocks into a U-Net framework for single-image dehazing tasks. Similarly, Zou et al. (2024) proposed FreqMamba, a model that integrates frequency-domain analysis with Mamba for image deraining tasks. For image snow removal tasks, Wang et al. (2025c) developed SnowMamba, a dedicated model that employs multi-scale Mamba blocks to handle diverse snow patterns.

In the broader domain of image restoration, Mamba-based architectures have been widely adopted. Guo et al. (2024) proposed MambaIR and its enhanced variant MambaIRv2 (Guo et al., 2025), which leverage state space models for efficient long-range dependency modeling. Ding et al. (2025) designed a Cross-Modality Fusion Mamba (CMFM) model for all-in-one weather-degraded image restoration tasks. Other variants include CU-Mamba (Deng & Gu, 2024) for channel-wise feature processing, VMambaIR (Shi et al., 2025b) which incorporates visual inductive biases, and RetinexMamba (Bai et al., 2024) for Retinex-based low-light image enhancement tasks. Several approaches focus on improving computational efficiency and task specificity. MAIR (Li et al., 2025a) preserves local features while performing global context modeling, whereas RamIR (Tang et al., 2025a) employs prompting strategies to enable flexible image restoration. Peng et al. (2025b) proposed a texture-aware Mamba model to enhance fine-detail recovery, and Q-MambaIR (Chen et al., 2025d) introduced quantization techniques to facilitate efficient real-world deployment.

### 7.2.3 Visual Generation

High-resolution image, video, and 3D generation produce very long latent token sequences, over which the attention repeated at every diffusion or autoregressive step becomes the dominant cost. Linear-complexity backbones are used as scalable substitutes (Hu et al., 2024; Mo & Tian, 2024) that keep generation tractable at high resolution and long duration, often retaining a small amount of attention for fidelity. In the domain of image generation, several studies have integrated Mamba into diffusion modeling frameworks. Hu et al. (2024) proposed Zigmaa DiT-style zigzag Mamba diffusion model that alternates between spatial and frequency domains to enable efficient high-resolution image synthesis. Ergasti et al. (2025) developed U-Shape Mamba to accelerate diffusion sampling processes via state space model mechanisms. For autoregressive image generation tasks, Li et al. (2025c) proposed a scalable autoregressive image generation framework that leverages Mamba, whereas Chen et al. (2024d) proposed MaskMambaa hybrid Mamba-Transformer architecture tailored for masked image generation tasks. Video generation has also benefited significantly from Mamba's robust sequential modeling capabilities. Gao et al. (2024) proposed Mattena model that integrates Mamba with self-attention mechanisms for high-fidelity video synthesis tasks. Mo & Tian (2024); Mo (2025) explored scaling diffusion Mamba models with bidirectional SSMs to enable efficient video and 3D shape generation, demonstrating enhanced performance on complex generative tasks.

For motion and gesture generation tasks, Zhang et al. (2024i) proposed Motion Mamba for efficient long-sequence motion generation, which was later extended to InfiniMotion (Zhang et al., 2024h)a model that leverages Mamba to process arbitrarily long motion sequences. Fu et al. (2024) proposed MambaGesture, which enhances co-speech gesture generation via disentangled multi-modal fusion mechanisms integrated with Mamba. Several hybrid architectural approaches have also emerged. Fei et al. (2024) developed DiMBAa model that integrates Transformer and Mamba architectures into diffusion models to leverage the complementary strengths of both paradigms.

### 7.2.4 3D Vision

3D data, whether a point cloud, a voxel grid, or a volumetric scan, carries informative structure spread across the whole volume and has no natural reading order, so relating distant regions calls for a global receptive field. Softmax attention provides one but scales quadratically in the number of points or voxels, which is prohibitive at the resolutions 3D tasks demand. Linear-complexity SSM scans (Zhang et al., 2024c) are adopted because they retain this global view at a cost linear in the token count, and the central design question becomes the serialization order that maps unordered 3D structure onto a sequence (Wang et al., 2024c). As in 2D, dense tasks such as segmentation and reconstruction commonly add convolutional local modeling to recover fine geometric detail. In the field of 3D medical image segmentation, several Mamba-based approaches have been developed. Xing et al. (2024) proposed SegMamba for long-range sequential modeling in 3D medical image segmentation tasks, which was later extended to SegMamba-V2 (Xing et al., 2025) for general-purpose 3D medical image segmentation tasks. Wang et al. (2024a) proposed Tri-plane Mamba, a model that efficiently adapts the Segment Anything Model (SAM) (Kirillov et al., 2023) for 3D medical image segmentation tasks. Cao et al. (2024) presented MedSegMambaa 3D CNN-Mamba hybrid architecture tailored for 3D brain tissue segmentation tasks. Shi et al. (2024) developed ShapeMamba-EM, a model that fine-tunes 3D foundation models using local shape descriptors and Mamba blocks for 3D electron microscopy (EM) image segmentation tasks.

For point cloud-based 3D vision applications, Zhang et al. (2024c) proposed Voxel Mambaa group-free state space model tailored for 3D object detection tasks. Li et al. (2024c) proposed 3DET-Mamba for causal sequence modeling in end-to-end 3D object detection systems. Wang et al. (2024c) proposed Serialized Point Mambaa dedicated serialized point cloud segmentation model. Jin et al. (2025) developed UniMamba for unified spatial-channel representation learning in LiDAR-based 3D object detection tasks. Yu et al. (2024) explored cross-model knowledge distillation techniques to boost the performance of LiDAR-based 3D sparse detectors using Mamba.

In the domain of 3D reconstruction, Shen et al. (2025) proposed Gambaa model that integrates Gaussian Splatting with Mamba for single-view 3D reconstruction tasks. Dong et al. (2024) proposed Hamba for single-view 3D hand reconstruction tasks, which employs a graph-guided bi-scanning Mamba mechanism. For hyperspectral image classification tasks, He et al. (2024) developed 3DSS-Mambaa dedicated 3D-spectral-spatial Mamba architecture.

### 7.2.5 Video Understanding and Other Applications

Video compounds a long temporal axis with many spatial tokens per frame, so token counts grow quickly: transformer video models typically cap out at a few hundred frames or fall back on token compression that discards information. Linear-complexity temporal modeling raises this ceiling: reported systems encode on the order of a thousand frames on a single GPU with large memory savings (Li et al., 2024b), which is why long-video understanding increasingly relies on Mamba backbones or Mamba–Transformer hybrids (Ren et al., 2025b). Chen et al. (2024a) proposed Video Mamba Suite, demonstrating SSMs as a versatile architectural alternative for a broad range of video understanding tasks. Li et al. (2024b) proposed VideoMambaa dedicated model specifically designed for efficient video understanding tasks via state space model (SSM) mechanisms. For processing ultra-long video sequences, Ren et al. (2025b) developed Vambaa hybrid Mamba-Transformer architecture capable of efficiently understanding hour-long video clips. Several studies have explored multi-modal applications of Mamba in video understanding tasks. Chen et al. (2025b) proposed H-MBAa hierarchical Mamba-based adaptation framework for multi-modal video understanding in practical autonomous driving scenarios. Tang et al. (2025b) proposed MUSEa Mamba-based efficient multi-scale learning framework for cross-modal text-video retrieval tasks. Li et al. (2024e) developed SpikeMBAa multi-modal spiking saliency Mamba model for precise temporal video grounding tasks.

Mamba architectures have been adapted for a variety of specialized video tasks. Hu et al. (2025b) proposed VC-Mamba to ensure causal consistency of Mamba-based feature representation consistency in implicit video understanding tasks. Mi et al. (2025) proposed MVQA, which employs Mamba with a unified sampling strategy for efficient video quality assessment tasks. Liang & Zhang (2025) developed a Mamba-driven method for hierarchical temporal multimodal alignment in referring video object segmentation tasks. For video generation and enhancement applications, Shi et al. (2025a) proposed a self-supervised ControlNet framework integrated with spatio-temporal Mamba modules for real-world video super-resolution tasks. Kwak et al. (2025) explored endowing Mamba-based vision models with temporal modeling capabilities by leveraging Temporal Shift Module (TSM) for efficient video understanding.

### 7.2.6 Medical Image Analysis

A 3D medical scan is exactly the kind of volumetric input flagged at the outset, running to hundreds of thousands of tokens, so convolutions can struggle to capture whole-volume context, while full attention is expensive at full resolution. SSM backbones such as SegMamba (Xing et al., 2024) and U-Mamba (Ma et al., 2024b) are adopted because they model long-range dependencies across the entire volume at linear cost, and because segmentation is dense and detail-sensitive they are typically paired with convolutional local modeling rather than used alone.

Medical image segmentation has witnessed widespread adoption of Mamba-based architectures. Jiang et al. (2024) proposed MLLA-UNetan efficient U-shape model that integrates Mamba-like linear attention mechanisms. Similarly, Su et al. (2025) developed VMKLA-UNet, which fuses vision Mamba with KAN-based linear attention mechanisms. Several pure Mamba-based segmentation architectures have also been explored, including VM-UNet (Ruan et al.), Mamba-UNet (Wang et al., 2024e), and its enhanced variant VM-UNet-V2 (Zhang et al., 2024e). LightM-UNet (Liao et al., 2024) is designed for lightweight 3D medical image segmentation tasks, whereas U-Mamba (Ma et al., 2024b) specifically enhances long-range spatial dependency modeling capabilities. Additional contributions include H-VMUNet (Wu et al., 2025) for high-order vision Mamba-based segmentation, Selective and Multi-scale Fusion Mamba (Li et al., 2025b), and LoG-VMamba (Dang et al., 2024)which incorporate local-global feature fusion mechanisms. Advanced segmentation approaches include Semi-Mamba-UNet (Ma & Wang, 2024) for semi-supervised medical image segmentation tasks, which integrates pixel-level contrastive and cross-supervised learning strategies. Qiong et al. (2025) proposed a frequency-domain decomposition SVD-based linear attention approach for high-precision medical image segmentation tasks. Zhong et al. (2025b) integrated vision Mamba with xLSTM-UNet to achieve enhanced medical image segmentation performance.

For medical image classification tasks, Yue & Li (2024) proposed MedMambaa dedicated model for automated medical image classification. Poornam & Angelina (2024) proposed VITALT, which leverages vision transformers (ViTs) (Dosovitskiy et al., 2021) integrated with attention and linear transformation mecha-

nisms for robust brain tumor detection tasks. These approaches collectively demonstrate the effectiveness of linear attention mechanisms in clinical diagnostic applications. For medical image reconstruction and synthesis tasks, Huang et al. (2024a) developed MambaMIR for joint medical image reconstruction and uncertainty estimation. Atli et al. (2025) proposed I2I-Mamba for cross-modal medical image synthesis tasks, which relies on selective state space modeling mechanisms. Ju & Zhou (2024) proposed VM-DDPMa hybrid model that integrated vision Mamba with diffusion models for medical image synthesis tasks.

Beyond medical image analysis, Xu et al. (2023) developed a hybrid reinforced medical report generation system that integrates m-linear attention and repetition penalty mechanismsdemonstrating the versatility of linear attention mechanisms in medical natural language processing tasks. While this section focuses on linear attention mechanisms, it's worth noting that traditional self-attention mechanisms continue to play important complementary roles in medical imaging tasks. Lu et al. (2023b) proposed multi-attention segmentation networks integrated with the Sobel operator for medical images segmentation tasks, providing complementary technical approaches to linear attention-based methods.

### 7.2.7 Remote Sensing

Very-high-resolution remote-sensing scenes are far too large for quadratic attention, so transformer pipelines crop them into patches, which severs the long-range context the task depends on, such as the direction of a road running across the whole image. Mamba backbones such as RS-Mamba (Zhao et al., 2024b) are adopted because their linear cost lets the whole scene be processed without cropping; consistent with this, reported accuracy can even improve as the input image grows larger. Several foundational Mamba-based architectures have been developed for general-purpose remote sensing tasks. Zhao et al. (2024b) proposed RS-Mambaa dedicated model for large-scale remote sensing tasks. Chen et al. (2024c) proposed RSMamba for remote sensing image classification tasks, which leverages state space models mechanisms. Wang et al. (2025b) developed Romaa scalable Mamba-based foundation model specifically tailored for large-scale remote sensing applications. For remote sensing image semantic segmentation tasks, Ma et al. (2024e) proposed RS3Mambaa visual state space model tailored for remote sensing image semantic segmentation. Zhu et al. (2024) proposed UNetMambaan efficient UNet-like Mamba architecture for high-resolution remote sensing image semantic segmentation tasks. Liu et al. (2024) developed CM-UNeta hybrid CNN-Mamba UNet architecture for remote sensing image semantic segmentation tasks. Zhang et al. (2024b) proposed S2CrossMamba for spatial-spectral cross-modal remote sensing image classification tasks.

Land use change detection has witnessed significant advancements with Mamba-based architectures. Chen et al. (2024b) proposed ChangeMamba for remote sensing change detection tasks, which leverages spatiotemporal state space modeling mechanisms. Zhang et al. (2025a) developed CDMambaa model that integrates local spatial clues into Mamba for binary remote sensing change detection tasks. Paranjape et al. (2025) proposed a Mamba-based Siamese network architecture for remote sensing change detection applications. For remote sensing object detection applications, Zhou et al. (2025) developed DMMa disparity-guided multispectral Mamba model for oriented remote sensing object detection tasks. Ren et al. (2024) proposed RemoteDet-Mambaa hybrid Mamba-CNN network architecture for multi-modal remote sensing object detection tasks. Gao et al. (2025) proposed MSFMamba for multi-scale feature fusion in multi-source remote sensing image classification tasks.

Various remote sensing image restoration tasks have benefited significantly from Mamba-based approaches. Xiao et al. (2024) proposed Frequency-assisted Mamba for remote sensing image super-resolution tasks. Zhou et al. (2024) developed RSDehambaa lightweight vision Mamba for satellite image dehazing tasks. Zhang et al. (2024a) proposed Mamba-CR for remote sensing image cloud removal. Chi et al. (2025) proposed RSMamba for biologically plausible Retinex-based remote sensing image shadow removal tasks. Zhu et al. (2025) proposed a Mamba-based collaborative implicit neural representation framework for hyperspectral and multispectral remote sensing image fusion tasksdemonstrating the versatility of Mamba architectures in handling complex multi-modal remote sensing data fusion tasks.

### 7.3 Audio and Speech Processing

Audio spectrograms and waveforms are long one-dimensional sequences, and the dominant Audio Spectrogram Transformer scales quadratically in both compute and memory with that length. This raises a direct question, whether self-attention is even necessary here, and self-attention-free SSM backbones such as Audio Mamba (Erol et al., 2024) and SSAMBA (Shams et al., 2024) answer it by modeling long-range temporal context at linear cost and accommodating variable-length inputs, used bidirectionally for offline tasks and causally for streaming ones.

### 7.3.1 Audio Representation Learning and Classification

Several foundational studies have explored the application of Mamba to audio representation learning tasks. Erol et al. (2024) proposed Audio Mambaa bidirectional state space model specifically designed for audio representation learning. Lin & Hu (2024) developed a pretrained audio state space model for audio tagging tasks. Yadav & Tan (2024) proposed selective state spaces mechanisms for self-supervised audio representation learning, whereas Shams et al. (2024) proposed SSambaa dedicated framework for self-supervised audio representation learning that leverages Mamba state space model.

### 7.3.2 Speech Processing and Enhancement

Speech processing has witnessed widespread adoption of Mamba-based architectures across diverse tasks. For speech enhancement tasks, Qian et al. (2025) proposed a scene-aware audio-visual speech enhancement framework, whereas Groot et al. (2025) developed CleanUMamba for high-quality speech denoising. For speech super-resolution tasks, Sui et al. (2024) proposed Trambaa hybrid Transformer-Mamba architecture tailored for practical audio and bone conduction speech processing applications. For speech recognition tasks, Gao & Chen (2024) proposed Speech-Mamba for long-context speech recognition, which leverages selective state space models, and Zhang et al. (2025e) explored Mamba as a computationally efficient alternative to self-attention in speech signal processing.

For speaker separation and diarization tasks, Avenstrup et al. (2025) proposed SepMamba, which leverages state space models for high-accuracy speaker separation, and Li et al. (2024a) proposed SpMamba for speech separationcollectively demonstrating the effectiveness of state space models. Jiang et al. (2025) conducted a comprehensive empirical evaluation of Mamba for speech separation, recognition, and synthesis tasks, whereas Plaquet et al. (2025) developed a Mamba-based segmentation model for speaker diarization.

### 7.3.3 Multi-modal Audio Processing and Specialized Applications

Mamba-based architectures have demonstrated robust performance across multi-modal audio tasks and specialized application scenarios. In audio-visual processing domains, Gong et al. (2025) proposed AVS-Mamba for audio-visual segmentation tasks, whereas Huang et al. (2024b) proposed AV-Mamba and Yang et al. (2025d) developed SHMamba for audio-visual question answering tasks. Li et al. (2025d) designed a Mamba-enhanced network architecture for text-audio-video cross-modal emotion recognition tasks.

For audio localization and specialized tasks, Kuang et al. (2025) developed BAST-Mamba for binaural sound localization, and Xiao & Das (2025) proposed TF-Mamba for sound source localization, along with TAME (Xiao et al., 2025) for drone trajectory estimation tasks. Chen et al. (2024e) presented RawbMamba for audio deepfake detection tasks. Collectively, these studies highlight the versatility of Mamba-based architectures in handling multi-modal data integration, spatial audio localization, and specialized audio processing applications.

### 7.4 Time Series Analysis

For time series the choice is essentially three-way. Transformer forecasters capture temporal and inter-variate dependencies well but scale quadratically, which is costly to deploy and worsens as the number of variates grows; simple linear models are cheap but limited. Selective SSMs, closely related to gated linear-attention-style recurrences under the unified view (Gu & Dao, 2024; Dao & Gu, 2024), sit between the two, capturing long look-back windows and inter-variate correlations at near-linear cost. The same balance makes them attractive for AI4Science settings with ultra-long sequences (genomic, climate, dynamical-system data) (Ma

et al., 2025; Ramesh et al., 2025), although their compressive state can lose information, which is why several methods below add convolution or attention.

### 7.4.1 Foundational Mamba Models for Time Series

Wang et al. (2025f) conducted a comprehensive empirical study to investigate the effectiveness of Mamba for time series forecasting tasks, establishing its robust performance in this domain. Patro & Agneeswaran (2024a) proposed Simbaa lightweight simplified Mamba-based architecture tailored for both computer vision and multivariate time series forecasting tasks. Ma et al. (2024a) developed a dedicated Mamba foundation model specifically designed for time series forecasting, providing a high-performance baseline for various temporal prediction tasks. Several studies have explored bidirectional sequence processing strategies for time series. Liang et al. (2024) proposed Bi-Mamba+a bidirectional Mamba architecture for enhanced time series forecasting performance. Zou et al. (2025) proposed BiG-Mambaa hybrid model that integrates bidirectional graph modeling with Mamba for multivariate time series forecasting taskseffectively capturing both temporal dynamics and structural dependencies in sequential data.

### 7.4.2 Multivariate and Long-Term Time Series Forecasting

For multivariate time series forecasting applications, Ma et al. (2024d) developed FMambaa model that incorporates fast-attention mechanisms into Mamba for efficient multivariate time series forecasting tasks. Feng et al. (2025) proposed DecMamba, which leverages time series decomposition techniques within the Mamba framework to achieve improved multivariate time series forecasting performance. Li et al. (2024d) proposed CMMamba, which focuses on optimized channel mixing mechanisms within Mamba for enhanced multivariate time series analysis tasks. Long-term time series forecasting has witnessed several Mamba-based technical innovations. Wu et al. (2024) developed UMambaTSFa U-shaped multi-scale Mamba architecture tailored for long-term time series forecasting tasks. Zhao et al. (2024a) proposed ISMRNNa hybrid model that integrates implicitly segmented RNN methods with Mamba for long-term time series forecasting tasks. Weng et al. (2025) proposed a lightweight simplified Mamba variant with disentangled temporal dependency encoding, specifically tailored for long-term time series forecasting tasks.

### 7.4.3 Specialized Architectures and Hybrid Approaches

Various specialized Mamba-based architectures have been proposed for targeted time series tasks. Lee et al. (2024) developed Sequential Order-Robust Mamba for time series forecasting tasks, which exhibits strong robustness to input sequence order variations. Xu et al. (2025) proposed SSTa model that integrates multi-scale hybrid Mamba-Transformer expert mechanisms for both long-range and short-range time series forecasting tasks.

### 7.4.4 Sequence Modeling Enhancements

Yuan et al. (2025) developed ReMambaa dedicated variant specifically designed to equip Mamba with effective long-sequence modeling capabilities for extended temporal forecasting contexts. Zhang et al. (2024f) proposed MatrReca hybrid model that unites Mamba and Transformer architectures for sequential recommendation tasksdemonstrating the versatility of Mamba-based models in diverse sequential modeling applications.

## 8 Infrastructure

Many of the models surveyed so far, especially the linear-attention and SSD-style variants, can be expressed through the per-step recurrence introduced in Sec. 2: a memory state written by an outer-product-style update and read out by a query, with linear attention, its gated and delta variants, and structured SSMs differing mainly in the form of that update. The infrastructure problem is therefore common to all of them, and follows from a single property of this recurrence: it is cheap in arithmetic but sequential and memory-bound, so reducing asymptotic complexity does not by itself translate into faster execution on a GPU. Turning that complexity into real speedups requires a hardware-efficient implementation, and the

systems work surveyed here is what carries the algorithms of the preceding sections into practical training and deployment; we therefore treat it on the same footing as the model designs themselves.

This theory-practice mismatch is particularly pronounced during the training phase. While inference can be efficiently executed via recurrent state updates, training mandates parallel processing of entire sequences to maximize hardware utilization efficiency. However, the naive sequential state-updating mechanisms of these models involve low arithmetic intensity operations that fail to fully saturate the massive computational throughput of GPUs, resulting in severe computational resource underutilization.

To bridge the gap between linear-complexity algorithms and hardware execution efficiency, a substantial body of research has focused on reformulating these computational patterns into hardware-friendly implementations. Broadly speaking, the infrastructure-level optimizations for linear sequence models can be categorized into three core paradigms:

1. **Algorithmic Reformulation for Parallelism**: This paradigm reformulates sequential recurrent dependencies into parallelizable computational forms (*e.g.*, chunk-wise computations, parallel scans operations, or global convolutions layers) to fully leverage GPU parallel computing capabilities (Sun et al., 2023b; Qin et al., 2024a; Yang et al., 2024a; Qin et al., 2024d; Yang et al., 2024b; Smith et al., 2023b; Gu & Dao, 2024).

2. **IO-Aware Kernel Optimization**: This paradigm minimizes memory bandwidth bottleneck via operation fusion and memory hierarchy management, enabling efficient handling of large-scale state variables during the training phase (Katharopoulos et al., 2020; Schlag et al., 2021; Qin et al., 2024d; Yang et al., 2024a; Arora et al., 2024b; Gu & Dao, 2024; Dao & Gu, 2024).

3. **Distributed System Strategies**: This paradigm develops sequence parallelism techniques tailored for recurrent models, optimizing inter-device communication patterns to enable training on arbitrarily long sequences across multiple hardware devices (Yang et al., 2024a; Qin et al., 2024a; Dao & Gu, 2024; Sun et al., 2024a; 2025b).

## 8.1 Foundational Computational Patterns

An algorithm's computational execution patterns fundamentally determine its execution efficiency on modern parallel computing processors, particularly GPUs. The architecture of the NVIDIA H100 GPU, for example, is equipped with highly specialized hardware units tailored for different types of computational operations. Large-scale matrix multiplication operations are offloaded to Tensor Coreshardware components specifically optimized for dense matrix computationswhich can deliver a peak computational throughput of 495 TFLOPS under the TF32 precision regime. In contrast, general-purpose vector or element-wise operations are processed by CUDA Cores, which provide a peak throughput of only approximately 67 TFLOPS under the FP32 precision regimeleading to a performance gap of nearly one order of magnitude (Choquette & Gandhi, 2020). This indicates that merely reducing theoretical computational complexity to linear scaling is insufficient for realizing practical training acceleration for linear attention mechanisms. If the core computation of linear attention models remains dominated by memory-bound vector operations, the massive computational throughput of the GPU will be severely underutilized. To mitigate this critical bottleneck, current research efforts focus on reformulating the computational graphs of linear attention mechanisms and SSMs, converting them into hardware-friendly computational patterns that can be efficiently executed on GPU architectures.

### 8.1.1 Linear Attention

Linear attention circumvents quadratic computational complexity by eliminating the explicit softmax operation, enabling the computation to be reformulated as a recurrent neural network (RNN). While theoretically computationally efficient, this recurrent formulation introduces a new bottleneck for hardware execution performance. Its core mechanism is a state update rule that sequentially accumulates contextual information along the sequence length, which is formally defined by the following equation:

$$S_t = S_{t-1} + v_t k_t^\top, \quad o_t = S_t q_t, \tag{63}$$

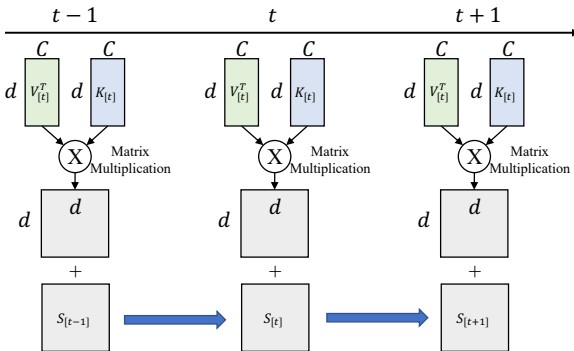

Figure 2: Chunkwise parallel form decomposes the sequence into chunks.

where at each timestep $t$, the terms $q_t, k_t, v_t \in \mathbb{R}^d$ denote the query, key, and value vectors of dimension $d$. The term $v_t k_t^\top$ represents a vector outer product operation, yielding a $d \times d$ matrix. This matrix is then added to the previous state matrix to generate the updated state $S_t \in \mathbb{R}^{d \times d}$. Finally, the output vector $o_t$ is computed via a matrix-vector multiplication operation between the state matrix $S_t$ and the query vector $q_t$.

While this recurrent structure reduces the overall theoretical computational complexity to $\mathcal{O}(Ld^2)$, its naive direct implementation poses a significant challenge for parallel training on GPU accelerators. Unlike inferencewhere sequential processing is inherently memory-efficienttraining mandates parallel processing of the entire sequence to maximize hardware computational throughput. However, the recurrent nature of linear attention mandates that the state at any timestep $t$ depends on the cumulative integration of all prior sequence history. This enforces a sequential execution path dominated by memory-bound vector operations, which fails to fully saturate the massive parallel computing capability of Tensor Cores.

To mitigate this mismatch between recurrent computation and parallel hardware architectures, a foundational optimization technique referred to as chunkwise parallel formulation has been widely adopted (Sun et al., 2023b; Qin et al., 2024a; Yang et al., 2024a; Qin et al., 2024d). This method partitions a sequence of length $L$ into $\frac{L}{C}$ chunks (each of size $C$) to hybridize sequential and parallel computation patterns.

It is important to highlight the evolutionary progression of this technique. Early iterationssuch as Lightning Attention-1 (Qin et al., 2024a)utilized chunking primarily to distribute computational workloads across parallel hardware. However, these early approaches computed intra-chunk attention via masked quadratic operations $(A_{ij} = (Q_i K_j^T) \odot M_{ij})$, meaning the theoretical computational complexity within each chunk remained quadratic. Subsequent advancementsmost notably Lightning Attention-2 (Qin et al., 2024d)strictly linearized the intra-chunk computation process. These advanced methods formalized inter-chunk state propagation as follows:

$$S_{[t+1]} = S_{[t]} + V_{[t]}^\top K_{[t]}, \tag{64}$$

where $S_{[t]}$ denotes the accumulated state matrix from all preceding chunks, as illustrated in Fig. 2. This operation corresponds to a dense matrix multiplication between $V_{[t]}^\top \in \mathbb{R}^{d \times C}$ and $K_{[t]} \in \mathbb{R}^{C \times d}$, which efficiently leverages Tensor Cores with a computational cost of $\mathcal{O}(Cd^2)$. Similarly, the intra-chunk computation step fuses historical information with local sequence context, defined as:

$$O_{[t]} = Q_{[t]} S_{[t]}^\top + (Q_{[t]} K_{[t]}^\top \odot M) V_{[t]}, \tag{65}$$

where $M$ denotes a causal attention mask matrix. By enforcing this strict state-space formulation, modern chunkwise linear attention algorithms achieve an overall computational complexity of $\mathcal{O}(Ld^2 + LCd)$, effectively converting memory-bound recurrent operations into compute-bound matrix multiplicationsoperations that are highly optimized in linear algebra libraries such as cuBLAS (NVIDIA Corporation, 2025).

Nevertheless, the expressive capacity of vanilla linear attention mechanisms is often constrained by their simple additive state update rules, which results in degraded performance on complex long-context tasks. To enhance the modeling capacity of linear attention and mitigate this limitation, recent research efforts have

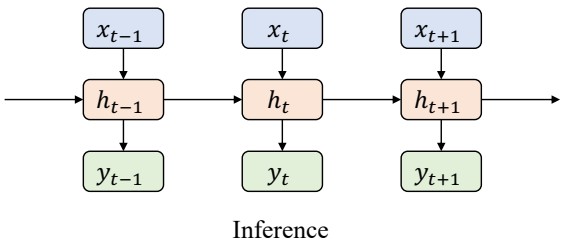 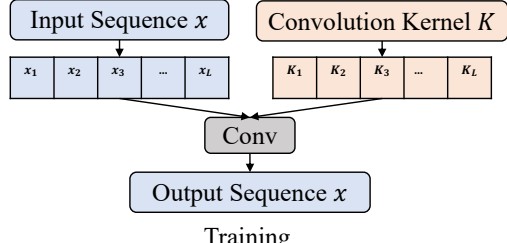

(a) Sequential recurrence mode, used for low-latency inference.

(b) Parallel global convolution mode, used for high-throughput training on full sequences.

Figure 3: Dual computational patterns of the S4 model.

proposed more sophisticated state update mechanisms. A notable example is the delta update rule (Yang et al., 2024b), which is formally defined as:

$$S_t = S_{t-1}(I - \beta_t k_t k_t^\top) + \beta_t v_t k_t^\top, \tag{66}$$

where $\beta_t \in \mathbb{R}$ is a data-dependent scalar coefficient that regulates the magnitude of the state update. The core innovation of this delta rule lies in the multiplicative update operation applied to the prior state matrix. However, a new computational challenge emerges when this recurrence is unrolled, which reveals an inherently sequential computation pattern:

$$S_t = \sum_{i=1}^{t} \left( \beta_i v_i k_i^\top \prod_{j=i+1}^{t} (I - \beta_j k_j k_j^\top) \right), \tag{67}$$

this formulation is incompatible with standard chunkwise linear attention computation kernels. The key insight for re-parallelizing this sequential computation is the WY representationa standard technique in numerical linear algebra for representing the product of multiple elementary matrices as a compact low-rank matrix update. Specifically, this technique transforms the sequential product of Householder-like elementary matrices back into an additive computational form:

$$\prod_{j=1}^{t} (I - \beta_j k_j k_j^\top) = I - \sum_{i=1}^{t} w_i k_i^\top, \tag{68}$$

where the newly introduced vectors $w_i$ can be computed efficiently in parallel. This mathematical transformation is critical because it converts the multiplicative recurrent operation back into an additive computational formulation. This restored additive computation pattern is once again compatible with chunkwise parallel computation kernels, thereby enabling efficient matrix multiplication operations on GPU accelerators.

### 8.1.2 State Space models

Distinct from linear attention mechanisms, State Space models (SSMs) represent another lineage of linear-complexity neural network architectures. Their hardware execution efficiency stems from a powerful computational duality: they can be formulated as either a stateful recurrent structure optimized for efficient autoregressive inference, or a global convolution representation tailored for highly parallelizable training.

The foundational S4 model (Gu et al., 2022b) is built upon a continuous-time linear dynamical system defined by parameter triplet $(A, B, C)$. This system is then discretized into a recurrent formulation, which is formally defined by the following state-space computation equations:

$$h_t = \overline{A}h_{t-1} + \overline{B}x_t, \quad y_t = \overline{C}h_t, \tag{69}$$

where the system maps the input vector $x_t \in \mathbb{R}^N$ to the output vector $y_t \in \mathbb{R}^N$ via a latent state vector $h_t$ at each timestep $t$. The parameter matrices $\overline{A}$, $\overline{B}$, and $\overline{C}$ are derived from the continuous-time system

parameters, which are fixed and input-independent. This recurrent formulation is highly efficient for autoregressive inference, since generating each new output step incurs a constant computational cost and memory overhead. However, its inherent sequential nature poses a significant bottleneck for model training, where the full input sequence is available in parallel and parallelized computation is favored on GPU accelerators.

To mitigate this bottleneck, S4 leverages its linear time-invariant (LTI) property to switch to a highly parallelizable convolutional representation:

$$y_t = \overline{C}(\overline{A}^t \overline{B} x_0 + \overline{A}^{t-1} \overline{B} x_1 + \cdots + \overline{AB} x_{t-1} + \overline{B} x_t). \tag{70}$$

By unrolling the recurrent computation across the full sequence length, it becomes evident that each output $y_t$ is a weighted sum of all prior input tokens, which can be reformulated as a single global convolution operation:

$$y = \bar{K} * x, \quad (\bar{K})_i = \overline{CA}^{i-1} \overline{B} \quad \text{for } i = 1, \dots, L, \tag{71}$$

where $*$ denotes the convolution operator. The convolution kernel $\bar{K}$ is a structured vector of length $L$ and can be precomputed in full prior to the forward pass computation. Since modern GPUs provide highly optimized linear algebra libraries for convolution operations, this convolutional formulation enables massive parallelization and substantial training acceleration.

While S4 leverages the LTI property for efficient training via global convolutions, Mamba (Gu & Dao, 2024) introduces input-dependent adaptive parameters to enable content-aware reasoning. This input-dependent selectivity violates the convolution theorem, shifting the computational pattern back to a time-varying recurrent formulation:

$$h_t = \bar{A}_t h_{t-1} + \bar{B}_t x_t. \tag{72}$$

To enable efficient parallel training, Mamba employs a parallel associative scan algorithm. Unlike naive sequential recurrent computation, this algorithm computes the prefix sum of state updates in logarithmic-time parallel steps. However, despite maintaining linear $\mathcal{O}(L)$ computational complexity, the scan operation exhibits low arithmetic intensity, resulting in a computational workload that is theoretically efficient but practically memory-bound on modern GPU accelerators.

To retain linear computational complexity while aligning with the hardware preference of dense matrix multiplications, Mamba-2 (Dao & Gu, 2024) introduces the Structured State Space Duality (SSD) framework. The SSD framework establishes that structured SSMs are mathematically equivalent to matrix multiplications involving semiseparable matrices. By constraining the state transition matrix to a scalar-identity structured form, Mamba-2 leverages a block matrix decomposition algorithm. This method partitions the input sequence into fixed-size chunks, where intra-chunk interactions are computed via a dual quadratic form operation:

$$Y = (L \circ (CB^\top))X, \tag{73}$$

which mirrors the matrix multiplication operations of linear attention mechanisms. Meanwhile, inter-chunk dependencies are managed via low-rank recurrent state propagation. This hybrid computation pattern effectively shifts the computational bottleneck from memory-bound scan operation back to compute-bound Tensor Core matrix multiplication operations, substantially boosting training throughput.

The two lineages thus converge on a similar hardware principle rather than an identical implementation: expose dense block-level matrix multiplications where possible, and propagate compact states across blocks to preserve linear or near-linear scaling. Concretely, whether one starts from linear attention or from a structured SSM, efficient training tends to partition the sequence into chunks, compute intra-chunk interactions as a dense, attention-like matrix product on Tensor Cores, and propagate a compact inter-chunk state. This is the hardware-level reading of the duality the SSD framework above makes explicit, so the choice between an SSM and a linear-attention presentation is often more a matter of formulation than of how the model executes on the GPU.

## 8.2 GPU Kernel-Level Optimization

A core challenge in training linear attention models is the excessive memory overhead incurred during the backward pass. A naive implementation relying on automatic differentiation would require caching

the hidden state matrix at every timestep, resulting in a severe memory bottleneck on GPU accelerators. Katharopoulos et al. (2020) developed a custom CUDA kernel for the backward pass that leverages a reverse-calculated cumulative sum to avoid storing intermediate state matrices. Based on this work, Schlag et al. (2021) further adapted this custom CUDA kernel to enable efficient training of their more complex delta network architecture. They recompute fast weights on-the-fly during the backward pass, trading a modest amount of redundant computation for substantial reductions in memory overhead. Gu & Dao (2024) also adopted the recomputation technique to reduce memory consumption in their Mamba model.

Beyond memory capacity optimizations during model training, another key trend in accelerating linear attention mechanisms is the development of I/O-aware GPU kernel designs. Many of these recent research efforts draw inspiration from FlashAttention (Dao et al., 2022), with the goal of mitigating the memory bandwidth bottleneck. The core idea is to minimize expensive data transfers between High-Bandwidth Memory (HBM) and on-chip Static Random-Access Memory (SRAM), which is accomplished by tiling input data into blocks for loading into SRAM and fusing kernel operations to maximize on-chip data reuse. This optimization mechanism has been effectively implemented in several recent studies. For example, Qin et al. (2024d) introduced Lightning Attentionan I/O-friendly kernel designthat addresses issues arising from cumulative summation while substantially reducing memory usage and runtime latency. Concurrently, Yang et al. (2024a) developed FlashLinearAttention for GLA, which provides two distinct I/O-aware variants enabling a trade-off between memory consumption and computational parallelism by deciding whether to materialize intermediate hidden states in HBM. Arora et al. (2024b) designed a custom CUDA kernel for their BASED architecture that explicitly fuses multiple operations in Taylor exponential linear attention, further optimizing data movement down to the register level to achieve substantial throughput improvements.

This focus on I/O-aware kernel optimization also extends to State Space Models (SSMs). The Mamba architecture (Gu & Dao, 2024) introduced input-dependent adaptive parameters, boosting model expressivity but violating the LTI propertycrucial for efficient convolutional-based training. To mitigate the resulting sequential bottleneck in recurrent training mode, the authors developed a hardware-aware parallel scan algorithm. Inspired by memory hierarchy optimizations analogous to those in FlashAttention (Dao et al., 2022), this algorithm employs a fused kernel operation: it loads parameters directly from HBM to SRAM, executes discretization and state recurrence entirely on-chip, and avoids materializing large intermediate state matrices in HBM. Furthermore, it leverages recomputation during the backward pass (instead of storing intermediate states), maintaining constant memory complexity and substantially accelerating training throughput compared to naive recurrent training approaches. Building on this work, Dao & Gu (2024) subsequently proposed the Structured State-Space Duality (SSD) framework. Recognizing that Mamba's scan operation underutilizes matrix multiplication hardware units, the SSD framework employs block matrix decomposition: diagonal blocks are computed in parallel via an attention-like quadratic form (efficiently mapped to matrix multiplication operations), while off-diagonal blocksrepresenting inter-chunk dependenciesutilize a lightweight, efficient scan operation. This hybrid computation approach substantially improves hardware utilization efficiency and delivers significant speedups over the original Mamba scan algorithm.

## 8.3 Distributed Training and Inference Systems

As sequence lengths for large language models (LLMs) exceed the memory capacity of a single hardware device, a crucial optimization direction lies in the development of distributed training and inference systems. For linear attention and state-space models (SSMs)whose core mechanism fundamentally enable sequence modeling with arbitrarily long lengthresearch efforts have focused on designing efficient Sequence Parallelism (SP) strategies that partition long input sequences across multiple hardware devices. This section surveys the evolutionary progression of these techniques, range from adaptations of existing parallelism paradigms to novel, communication-efficient algorithms specifically tailored for the unique properties of linear sequence models.

Early research approaches focused on adapting the chunkwise parallel formulation for multi-GPU distributed execution or leveraging standard distributed training paradigms. In their work on GLA, Yang et al. (2024a) proposed an I/O-aware kernel with a materialized variant specifically designed to enable efficient sequence-level parallelism. This method computes all segment-level hidden state matrices and stores them in HBM, thereby enabling parallel computation of outputs for all sequence segments across different devices. Sepa-

rately, to scale their linear attention model to longer sequences, Qin et al. (2024a) employed a combination of well-established techniques in TransNormerLLMincluding fully sharded data parallelism (FSDP) for distributing model parameters and states, and tensor parallelism for partitioning the computation of attention and MLP blocks within each compute node. Furthermore, Mamba-2 (Dao & Gu, 2024) modified its computational block structure to enable efficient tensor parallelism with minimized communication overhead, while the inherent recurrent nature of SSMs facilitates straightforward sequence parallelism via the transmission of compact intermediate states between devices assigned to handle distinct sequence segments.

Recent research has specifically tailored sequence parallelism strategies for linear attention mechanisms to reduce communication overhead by leveraging their unique algebraic properties. Sun et al. (2024a) introduced Linear Attention Sequence Parallelism (LASP), which employs a point-to-point ring communication mechanism. Instead of exchanging full key-value (KV) states whose size scales linearly with sequence length, LASP only transmits compact, sequence-length-independent intermediate memory states between participating devices. Building on this framework, they further developed LASP-2 (Sun et al., 2025b), and identified that LASP's ring communication mechanism introduces sequential communication dependencies that hinder parallel execution efficiency. LASP-2 replaces the ring communication mechanism with a single AllGather collective communication operation: participating devices first compute local intermediate states in parallel, then gather all local states concurrently via the AllGather operation, and finally compute the global final outputssubstantially enhancing the parallelism of both communication and computation processes. This AllGather-based sequence parallelism approach was further extended to LASP-2H, a variant tailored for hybrid models that integrate both linear attention and standard softmax attention layers.

## 9 Challenges, Analyses, and Insights

The mechanisms surveyed so far all realize the same memory write/read recurrence of Sec. 2, and the analyses in this section probe the single trade-off that recurrence forces: a fixed-size state makes the *write* lossy, so what the write compresses away is summarized rather than preserved token by token at *read* time. The three challenges below each expose one side of this trade-off: long-context retrieval concerns what survives compression; length extension concerns what happens once the state is driven past its trained regime; and scaling concerns whether the trade-off improves predictably with size. The unified-framework subsection then asks how the same recurrence has been re-derived across communities. For the retrieval and scaling challenges, we connect each limitation and its remedies back to the sub-families of Secs. 3, 5 and 6, keeping the analysis aligned with the taxonomy. We close with a synthesis of our key insights into linear attention.

### 9.1 Long-Context Retrieval

Linear attention has gained growing prominence in practical applications, particularly for tasks that demand long-context information retrieval and precise associative recall. For associative recall tasks, the model is required to retrieve a previously stored value token in response to a corresponding key token. However, because linear attention compresses contextual memory into a fixed-dimensional latent representationinstead of modeling pairwise interactions between all token pairs as softmax attention doesits performance in such tasks is often compromised.

**Challenging Tasks for Linear Attention**   Several studies have systematically analyzed the limitations of linear attention models, demonstrating that linear attention exhibits suboptimal performance on the following specific tasks (see Tab. 4 for details) (Arora et al., 2024a; Jelassi et al., 2024; Wen et al., 2025; Arora et al., 2024c; Waleffe et al., 2024; Park et al., 2024):

**Localization of Core Limitations**   Recent mechanistic interpretability studies have successfully identified the fundamental associative retrieval algorithm employed by Transformers and diagnosed the core failure points of linear attention within recurrent architectures. Transformers achieve robust associative recall via the uncompressed key-value (KV) cache and the development of specialized attention heads, which leverage the key-value associations to implement precise in-context retrieval. Bick et al. (2025) formalized this process

Table 4: Summary of task categories and retrieval mechanisms in SSMs.

| Task Category | Primary Objective | Core Retrieval Mechanism Tested | Typical Failure Mode in SSMs |
|---|---|---|---|
| Copying Task | Accurately retrieve a fixed, long sequence of tokens after a delay. | Raw Fixed Memory Capacity and Sequential Fidelity. | Loss of fidelity due to finite hidden state constraint/compression. |
| In-Context Retrieval | Locate and extract a specific, singular piece of information from a lengthy document. | Resistance to Recency Bias (Long-Range Dependency). | Degradation of token representation over long sequences. |
| Multi-Query Associative Recall (MQAR) | Dynamically infer and retrieve multiple associated values given non-sequential keys in context. | Content-based, Dynamic Key-Value Association. | Inability to execute sharp attention patterns; leakage and aggregation failure. |

into a unified associative retrieval mechanism termed Gather-and-Aggregate, and recurrent linear attention architectures realize this same capability with a smaller budget of sharp, head-like computation.

Additionally, visualization of attention patterns in Mamba-based models reveals that their attention maps are inherently smoother and significantly less spatially localized. This property causes linear attention to struggle to maintain precise token-level focus, often attending unnecessarily to adjacent tokens in addition to the target summary token. This attention dilution introduces extraneous noise, undermining the integrity and effectiveness of both the aggregated contextual memory and the associative lookup process. This attention continuity originates from the necessity of compressing the entire sequence history into a continuous hidden state $h_t$ (Qu et al., 2025).

A third, more fundamental factor is that the core efficiency advantage of linear attention and SSMs comes from compressing the entire input sequence history into a fixed-dimensional recurrent state. This finite-dimensional hidden state fundamentally limits the model's effective memory capacity, leading to inherent precision limitationsspecifically, it struggles to execute the local token shifts and pairwise comparisons that are essential for accurate associative recall (Arora et al., 2024a).

To summarize, sharp, non-average, token-to-token interaction patterns are essential for precise contextual aggregation. The performance degradation can be localized to the suboptimal execution quality of only a few critical attention heads, indicating that these efficient architectures possess the necessary structural components but lack the precision required to execute the associative retrieval algorithm effectively.

**Optimization Attempts and Solutions** Building on this localization, a line of work pins down where and why the gap arises, and this diagnostic evidence has shaped current understanding as much as any single fix. The Gather-and-Aggregate (G&A) analysis of Bick et al. (2025) is a representative example: inspecting attention maps on knowledge-intensive tasks such as MMLU, they separate retrieval into a *Gather* head that collects candidate spans and an *Aggregate* head that selects among them, and in hybrid Mamba models these functions concentrate in a small number of heads, so degrading those heads alone reproduces most of the recall gap. The reading that follows is that SSMs carry the structural components for retrieval and allocate a modest budget of sharp, head-like capacity to a localizable sub-computation. Complementary analyses triangulate the same conclusion from different angles: Jelassi et al. (2024) use controlled copying to show that the gap tracks the length of the string to be reproduced, pointing to finite state; Arora et al. (2024a) tie MQAR recall to content-based key–value association through local comparisons; and Wen et al. (2025) locate the effect in in-context retrieval by measuring how performance recovers as softmax layers are added. Together these studies place the limitation in a small, identifiable retrieval sub-circuit.

Read against this diagnosis, the proposed remedies each target a specific part of the sub-circuit. One family adds sharp, head-like computation, whether by inserting a few softmax layers or retrieval modules (Wen et al., 2025) or by building Mamba–attention hybrids such as MambaFormer and Mamba-2-Hybrid (Waleffe et al., 2024; Park et al., 2024), supplying G&A capacity directly. A second family strengthens the recurrent association itself, through delta-rule updates (Yang et al., 2025b) and BASED-style convolution–attention (Arora et al., 2024a), sharpening key–value bindings. A third family works at the training and prompt level, including JRT-Prompt and the non-causal JRT-RNN (Arora et al., 2024c), selective-copy objectives (Blouir et al., 2024), and retrieval-oriented curricula (Bick et al., 2025), improving how reliably the existing components run the retrieval algorithm. The interventions that help most are those aimed at the identified sub-circuit. Mapped onto the unified framework of Sec. 2, these remedies trace a clear ordering across sub-families, which also gives a practitioner a sequence of starting points when retrieval fidelity is the binding constraint. Purely additive linear attention, such as kernel or feature-map models like Performer (Choromanski et al., 2021), accumulates key–value bindings uniformly and offers the plainest baseline for fidelity (Sec. 3.1). Data-dependent gating (GLA, Sec. 3.2) and decay in linear RNNs (HGRN2, Sec. 5) add selectivity by down-weighting stale content. Delta-rule updates (DeltaNet, Gated DeltaNet (Yang et al., 2024b; 2025b)) sharpen the association further through explicit error correction, reinforcing the key–value bindings that the analyses above highlight. Finally, full/linear hybrids (Sec. 6) add a small budget of exact softmax read-out and close most of the remaining gap.

## 9.2 Length Extension Capability

Linear attention largely mitigates the sequence length scalability bottlenecks of Transformers and enables efficient autoregressive inference, yet this architecture introduces a fundamental constraint: its ability to retain and propagate long-range contextual information is limited by the adaptation capability of the fixed-dimensional latent state to sequences longer than the training length $L_{train}$. For linear attention mechanisms, the sequence length extension challenge mainly manifests in three key aspects: (a) Effective receptive field limitation: The model's ability to capture dependencies between distant tokens is bounded by $L_{train}$, which causes information from early tokens to decay or vanish in ultra-long sequences. (b) Fixed-size hidden state overflow: Fixed-dimensional latent states become corrupted by irrelevant noisy tokens when the evaluation sequence length $L_{eval}$ is excessively large, leading to associative retrieval errors or semantically incoherent generation outputs. (c) Polynomial extrapolation bias: Zero-initialized hidden states force the model to extrapolate polynomial-like memory kernels, which are unable to fit long-range sequence patterns. Several optimization approaches have been proposed to address these challenges, which can be categorized into three types of technical adjustments.

### 9.2.1 Training-Free Inference Adjustments

These methods generally filter out noisy tokens or redundant sequence chunks to convert a long-context task into equivalent shorter-context subtasks without additional training overhead. Specifically, Ye et al. (2025) first extract global channel-wise features using a decay threshold, then mark key tokens within these channels with state decay suppression and state update freezing to preserve critical hidden state information. Ben-Kish et al. (2025a) score token importance based on a hidden state filtering mechanism embedded within the *S6* layer (Gu & Dao, 2024). They retain only the top-ranked tokens in layers with a large Effective Receptive Field (ERF), and gradually reduce the number of retained tokens for deeper network layers. Ben-Kish et al. (2025b) also proposed a chunk-level sequence extension technique. This method partitions the input context into fixed-size chunks and feeds each chunk into the model independently. It then discards chunks where the model predicts an "Error" token, and retains chunks with the minimum entropy or maximum query matching probability to compress the original input context into a set of semantically critical chunks.

### 9.2.2 Training-Adjusted Techniques

These methods modify training-sequence configurations to address challenges associated with hidden state dynamics. Wang (2024) pointed out that zero-initialized hidden states in the memory kernel of SSMs are trained to fit short sequences, yet extrapolating this kernel to ultra-long sequences fails to capture long-range contextual dependencies. Thus, they replaced zero initialization with the final hidden state of the preceding

training batch to simulate long-context training without increasing GPU memory overhead. Buitrago & Gu (2025) extended the aforementioned method by proposing four variants to simulate initial hidden state configurations for long-context inference scenarios. These variants include: initializing hidden states with random noise or the variance of final state-related noise; passing the final hidden states of the current batch to the next batch; and partitioning long contexts into short chunks and propagating hidden states across consecutive chunks. Chen et al. (2025c) pointed out that Mamba's training process with oversized hidden states lacks an effective mechanism to "forget" irrelevant noisy tokens. When the inference sequence length increases, the hidden state becomes overloaded with noise, leading to associative retrieval errors. Therefore, they proposed a formula to calculate a forgetting threshold based on the hidden state dimension. They trained the model with sequence length exceeding the forgetting threshold to enforce the discarding of irrelevant tokens, thereby preventing hidden state overload.

### 9.2.3 Hybrid Fine-Tuning Approaches

Hybrid methods integrate training-free inference adjustments with lightweight fine-tuning strategies to balance model performance and deployment flexibility. Yuan et al. (2025) employed a two-stage forward pass strategy during the training process. The hidden states of the final layer are computed in the first stage. The hidden state of the final token is then used as a query vector to select the top-$k$ most relevant hidden states via cosine similarity matching. These selected states are ultimately aggregated to compute updated hidden states for optimization in the second stage, forcing the model to learn with a compact set of critical states.

### 9.3 Underexplored Scaling Laws for Linear Attention

A core open question in the field is whether linear attention architectures can comply with the empirical scaling laws of large language models (LLMs) (Kaplan et al., 2020). Shen et al. (2024) systematically investigated the scalability of several representative linear attention architectures, including TNL (linear attention with data-independent decay), HGRN2 (linear RNN with data-dependent decay), and cosFormer2 (linear attention without decay). By scaling model sizes from 70M to 7B parameters, training on a high-quality 300B-token corpus, and conducting comprehensive evaluations across a diverse range of downstream tasks-including validation perplexity, commonsense reasoning, and retrieval-augmented generation-they demonstrated that linear attention architectures exhibit predictable scaling behavior comparable to that of standard Transformer models. Notably, these linear attention models retain the core computational efficiency advantages of linear complexity, while achieving competitive language understanding capabilities and enhanced long-term knowledge retention. This empirical evidence indicates that linear attention models not only exhibit reliable performance scaling with increases in model parameter count and training data volume, but also establish well-defined scaling laws for this architectural paradigm.

Recent large-scale deployments further support this direction. MiniMax-01 (MiniMax et al., 2025) developed and deployed a large foundation model leveraging Lightning Attention-a highly optimized linear attention variant-as its core component within a hybrid architecture. This model supports long-context processing at million-token scale, and represents the first large-scale deployment of linear attention in a production foundation model. Similarly, the Ring-Linear model series (Team et al., 2025b) introduced a balanced hybrid architectural design that strategically fuses linear attention and standard softmax self-attention mechanisms. Systematic scaling experiments on this model series identified an optimal layer-wise fusion ratio that maintains state-of-the-art performance levels, while cutting long-context computational cost by up to an order of magnitude relative to a 32B-parameter dense Transformer baseline. Kimi Linear (Team et al., 2025a) reports a similar conclusion under matched-protocol pretraining, where its hybrid KDA design matches or surpasses a full-attention MLA baseline across short-context, long-context, and RL-scaling regimes. Together, these results indicate that linear attention is not only theoretically scalable but also practical for building large-scale language models.

This scaling evidence spans several sub-families and several levels of rigor, which are worth keeping distinct. As *controlled* evidence, the study of Shen et al. (2024) covers kernelized linear attention and linear RNNs through TNL, HGRN2, and cosFormer2 under a matched training protocol. As *large-scale deployment*

evidence, the MiniMax-01 release builds on Lightning Attention and Ring-Linear scales a linear–softmax hybrid, though such releases vary in how much is held fixed. As *matched-protocol model-family* evidence, the delta-rule family is compared against a full-attention baseline through Kimi Linear's KDA design, with Gated DeltaNet also deployed at scale in Qwen3-Next (Qwen, 2025). Selective SSMs such as Mamba are characterized in their own line of work (Sec. 4), and a systematic scaling account of test-time-training-style models remains an active and open direction. Reading the evidence at these three levels keeps each scaling result attached to both the design and the protocol it was measured under.

For practitioners, the dependable starting points are designs already validated at the intended scale, such as Gated DeltaNet in Qwen3-Next (Qwen, 2025) and Kimi Delta Attention (Team et al., 2025a), with the many newer variants marking promising directions to build on. A coarse linear-to-full attention ratio near 3:1 is a useful starting point in several studies and systems, but it is not universal: recent deployments also adopt different ratios, such as the 1:7 schedule of MiniMax-01 and Jamba, so finer gating and ratio choices are best confirmed at the intended scale and domain rather than transferred wholesale. As rough guidance, retrieval-heavy or instruction-following workloads benefit most from reserving a small budget of exact attention; throughput- or memory-constrained deployments favor a higher linear-to-full ratio; and dense multimodal perception tasks typically pair a linear backbone with local convolution.

## 9.4 Unified Frameworks for Linear Attention Mechanisms

Multiple research communities have independently developed linear attention mechanisms. However, discrepancies in terminology systems and mathematical formalism across different disciplines have posed substantial interdisciplinary barriers, hindering the clear understanding of their intrinsic relationships, fair performance comparison, and timely tracking of cutting-edge research developments. In this section, we review recent research efforts aimed at developing unified theoretical frameworks to bridge these fragmented research strands. Several studies (Katharopoulos et al., 2020; Schlag et al., 2021; Irie et al., 2021; Irie & Gershman, 2025) have formally established the mathematical equivalence between *2D-state linear RNNs* (*e.g.*, Fast Weight Programmers) and *linear attention* mechanisms. Concurrently, the SSM research community has uncovered a fundamental mathematical duality between *structured SSMs* and *linear attention mechanisms* (Ali et al., 2025; Dao & Gu, 2024), demonstrating that these two architectures are functionally coincident under appropriate parameter configurations. Furthermore, under specific mathematical conditions, SSMs can be exactly represented as linear RNNs (Gu et al., 2022b; Orvieto et al., 2024), further strengthening the intrinsic connections among these model families. Beyond these pairwise equivalence relationships, recent research has proposed more comprehensive unifying theoretical perspectives that encompass the entire family of linear-complexity sequence modeling architectures.

### 9.4.1 Mechanism Framework

This framework encompasses three core analytical perspectives: the associative memory perspective, the dynamical system perspective, and the computational complexity perspective.

**Associative Memory Perspective.** Associative memory is a fundamental cognitive mechanism for storing and retrieving relational associations between input items. Fast Weight Programmers (FWP, see Sec. 5.4 for more details) were originally proposed to model biological associative memory systems, and consist of two complementary components: slow weights and fast weights. Slow weights encode long-term, context-independent knowledge, whereas fast weights capture rapidly updated, context-dependent associations modulated by slow weight parameters. In linear attention mechanisms, slow weights correspond to the learned projection matrices that generate query, key, and value vectors ($W_Q$, $W_K$, $W_V$), whereas fast weights correspond to the dynamic key-value (KV) memory representations accumulated during sequence processing. Neurobiological evidence corroborates this dual-memory architecture: slow weights correspond to stable synaptic connections in biological neural systems, whereas fast weights reflect rapid, context-dependent neural activity patterns modulated by slow weight parameters (Irie & Gershman, 2025). The EOS framework (Qin et al., 2024c) provides a unified analytical paradigm by decomposing linear-complexity sequence models into three sequential stages: (1) *Expand*: projecting input tokens into a high-dimensional latent memory space; (2) *Oscillation*: recursively integrating current and historical memory states; (3) *Shrink*:

mapping the aggregated memory representation back to a low-dimensional output space. Miras (Behrouz et al., 2025b) draws inspiration from attentional biasthe innate human cognitive tendency to prioritize specific stimuli over others. It reframes linear-complexity sequence models as associative memory systems equipped with attentional bias, and formalizes this bias as an *internal memory optimization objective* that regulates keyvalue mapping and retention mechanismsrevealing a universal shared associative structure across diverse linear sequence models.

**Dynamical System Perspective.** A dynamical system refers to any system whose internal state evolves continuously or discretely over time. The *Dynamical Systems Framework* (DSF) (Sieber et al., 2024) demonstrates that linear attention mechanisms, linear RNNs, and SSMs can all be formulated as linear time-varying (LTV) dynamical systems, differing solely in the parameterization of their state transition matrices and input projection matrices. For instance, linear attention corresponds to a scalar gating mechanism applied uniformly across all hidden dimensions, whereas SSMs adopt dimension-wise selective update strategies with more sophisticated temporal dynamics. This framework also explains key empirical patterns observed across these architecturessuch as the superior long-range memory capacity of SSMs and the parallelizability of attention-based modelsby correlating these properties with the mathematical characteristics of their underlying state evolution operators.

**Computational Complexity Perspective.** The Prefix-Scannable Model (PSM) framework (Yau et al., 2025) unifies efficient sequence models based on their inherent sequential-parallel duality: parallelizable training with a computational complexity of $O(n)$, and sequential inference with an amortized time complexity of $O(1)$ per token. Models with affine state update operations achieve SPD-$(n, 1)$ complexity, sharing a universal common associative structure while differing in their specific gating mechanisms. The framework can be extended to non-associative operators (*e.g.*, softmax attention), enabling the construction of SPD-$(n, \log n)$ models with a memory complexity of $O(\log n)$.

### 9.4.2 Empirical Framework

This empirical framework primarily comprises two core analysis perspectives: the architectural design perspective and the state update mechanism perspective.

**Architectural Design Perspective.** Large-scale empirical studies have uncovered systematic performance patterns across diverse hybrid architectural designs for linear-complexity sequence models. STAR (Thomas et al., 2025) explores a unified operator search space encompassing linear attention, recurrence, and convolution, and automatically discovers optimal hybrid architectural topologies that incorporate only a small number of full-attention layers. Wang et al. (2025a) systematically evaluate 72 hybrid models, demonstrating that while language modeling performance remains largely stable across different attention ratio configurations, associative recall performance improves with an increasing proportion of full-attention and saturates at an approximate ratio of 3:1. Attention ratios ranging from 3:1 to 6:1, when combined with gated and hierarchical architectural designs, deliver optimal efficiencyrecall trade-offs. This 3:1 ratio has also been adopted and validated in recent large-scale industrial models, including Qwen3-Next (Qwen, 2025) and Kimi Linear (Team et al., 2025a). At the module level, Sun et al. (2025c) integrate all variants of linear attention modules as drop-in token-mixing layers within MoE blocks, interleaving them with standard TransformerMoE layers. Under the perspective of a unified recurrent framework, experimental results show that these hybrid model stacks retain the baseline performance of pure Transformer models while enhancing long-context processing efficiency via the LASP-2 (Sun et al., 2025b) sequence parallelism strategy, achieving stable training throughput and memory usage across model scales ranging from 0.3B to 2B and 1B to 7B.

**State Update Mechanism Perspective.** Differences in state update rules also exert a significant impact on model performance. Qin et al. (2025) investigated decay mechanisms across representative linear-complexity models including Mamba (Gu & Dao, 2024), GLA (Yang et al., 2024a), and HGRN2 (Qin et al., 2024e), and found that optimal decay values consistently cluster around 0.8 regardless of the underlying architectureindicating the existence of universal operating regimes for this model family. Gated architectural extensions, such as HGRN2 (Qin et al., 2024e) and Gated DeltaNet (Yang et al., 2025b), introduce per-token selective forgetting mechanisms, which effectively enhance model stability and associative recall capabilities in long-context scenarios.

**Future Research Directions.** Future research directions include two key avenues: first, developing algebraic and operator-theoretic frameworks to unify the theoretical foundations of linear-complexity sequence models; second, conducting large-scale empirical analyses to identify universal design patterns for hybrid model optimization. These unified theoretical frameworks can guide the design of interpretable gating and decay modules, and motivate the development of standardized benchmarks that better bridge the gap between theoretical research and practical deployment.

### 9.5 Insights

Through a systematic review of linear attention mechanisms from four core perspectivesmodule-level design, hybrid architecture design, infrastructure optimization, and downstream application deploymentwe distill core design principles for linear attention models and articulate practical strategies for their effective deployment and integration into state-of-the-art architectural frameworks. Looking ahead, we synthesize key insights and promising research directions to guide future advancements in this field:

#### 9.5.1 Retrieval Ability

Both standalone and hybrid linear attention models have demonstrated performance comparable to that of pure softmax attention models on mainstream language modeling benchmarks. Yet several studies (Arora et al., 2024a; Jelassi et al., 2024; Wen et al., 2025; Arora et al., 2024c; Waleffe et al., 2024; Park et al., 2024) have reported that linear attention can exhibit suboptimal performance relative to softmax attention on retrieval-centric tasks; specifically, linear attention often fails to retrieve fine-grained details from long-range historical contexta limitation that may stem from the compressive nature of its state update mechanism. If this performance gap persists, it would pose a non-trivial barrier to the deployment of production-scale large language models (LLMs) built exclusively on linear attention architectures. Encouragingly, a growing body of recent studies (Behrouz et al., 2025c;a; Team et al., 2025a) has reported enhanced retrieval performance by leveraging advanced techniquessuch as the delta rule (Yang et al., 2024b)and more sophisticated gating mechanisms. Nevertheless, a principled understanding of the core factors governing retrieval ability in linear attention mechanisms remains elusive. Future research should aim to elucidate the underlying cause of this retrieval performance gap and develop targeted mitigation strategies. Promising research directions include: (1) developing richer state representations and hybrid attention schemes; (2) integrating explicit memory-augmentation mechanisms; (3) devising training objectives tailored to preserving retrievable information; and (4) establishing retrieval-centric benchmarks and evaluation metrics that emphasize long-horizon information recovery and fidelity.

#### 9.5.2 Engineering Infrastructure & Deployment

Extensive efforts have been devoted to optimizing the efficiency and accuracy of traditional softmax attention across both training and inference stages. However, practical engineering support for linear attention variants remains insufficient. Although linear attention boasts theoretically superior computational complexity compared to softmax attention, the latter benefits from mature infrastructure optimizations (*e.g.*, FlashAttention (Dao et al., 2022)) and a rich ecosystem of highly optimized computational kernels. Notable infrastructure initiatives for linear attention do exist (*e.g.*, FLA (Yang & Zhang, 2024)); however, widespread inference support for many linear attention variants in modern inference enginessuch as vLLM (Kwon et al., 2023) and SGLang (Zheng et al., 2024)remains nascent, posing non-trivial practical challenges for large-scale deployment. Moreover, advanced rollout-based reinforcement learning (RL) training and other long-horizon applications require robust, low-latency, and memory-efficient long-sequence inference capabilities. Future work should prioritize the development of comprehensive, production-grade infrastructure for linear attentionencompassing optimized GPU/TPU kernels, fused computational operators, memory-efficient implementations, quantization-aware optimization, and seamless integration with mainstream training and inference frameworksto enable large-scale research and real-world deployment.

### 9.5.3 Benchmarking

Most existing sequence modeling benchmarks are tailored for evaluating models employing standard soft-max self-attention mechanisms. However, linear-attention architectures have demonstrated capabilities that exceed the performance envelope of conventional Transformer models, indicating that linear attention is not merely a secondary variant of softmax self-attention. For instance, linear attention achieves superior performance on state-tracking tasks (Zhong et al., 2025a; Siems et al., 2025) and universal context-free language recognition tasks (Merrill & Sabharwal, 2023). The practical task domains where linear attention excels require further systematic exploration, and its adaptation to diverse downstream tasks deserves more in-depth consideration. Accordingly, a promising research direction is to develop more diverse, architecture-specific benchmarks designed to probe the intrinsic functional differences between linear attention and softmax self-attention.

Furthermore, several recent studies have integrated key design elements derived from state-of-the-art linear-attention methods to enhance softmax-attention models (Bick et al., 2025; Lin et al., 2025; Qiu et al., 2025). This highlights another fruitful research avenue: developing novel sequence understanding paradigms that fuse the complementary strengths of linear attention and softmax self-attention mechanisms.

## 10 Conclusion

In this survey, we presented a comprehensive review of the evolutionary trajectory of diverse linear attention mechanismsincluding classical linear attention methods, State Space Models (SSMs), and linear recurrent neural network (RNN) models. Furthermore, we explored and synthesized the core factors critical to realworld deployment: hybrid architectures (the mainstream paradigm for integrating linear attention with conventional self-attention in LLMs), infrastructure (ensuring that theoretical efficiency advantages translate into practical performance gains), and practical application scenarios (aligning the inherent strengths and limitations of linear attention with realscenario task requirements). Building on this foundation, we presented a detailed analysis of the inherent characteristics, core advantages, key limitations, and unresolved challenges of linear attention mechanisms. We also proposed targeted suggestions for future research directionsincluding long-context retrieval capability enhancement, engineering infrastructure optimization, evaluation benchmark development, and real-world task adaptation. We hope that these discussions will accelerate future research progress and promote the practical deployment of linear attention mechanisms.

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

# A    Notation

Table 5 summarizes the core symbols used throughout this survey. Unless otherwise specified, vectors are columns, $\odot$ denotes the Hadamard (element-wise) product, and $\otimes$ denotes the outer product. The unified linear recurrent model (Eq. (14)) is written with state orientation $S_t \in \mathbb{R}^{d_v \times d_k}$, so that the outer-product write $\beta_t v_t k_t^\top \in \mathbb{R}^{d_v \times d_k}$ and the read-out $o_t = S_t q_t \in \mathbb{R}^{d_v}$ are dimensionally consistent for arbitrary $d_k$ and $d_v$. We adopt this orientation as the default; some individual sections quote a model in its original (possibly transposed) convention, and the matrix-valued state is variously written $S_t$, $W_t$, $H_t$, or $M_t$ depending on the source architecture.

Table 5: Summary of key notation used throughout the survey.

| Symbol | Space / Type | Meaning |
|---|---|---|
| *Dimensions and indices* | | |
| $N, L$ | $\mathbb{N}$ | Sequence length (number of tokens). |
| $d$ | $\mathbb{N}$ | Model / hidden dimension. |
| $d_k$ | $\mathbb{N}$ | Key (and query) dimension. |
| $d_v$ | $\mathbb{N}$ | Value dimension. |
| $t, i, j$ | $\mathbb{N}$ | Time-step / token indices. |
| $C$ | $\mathbb{N}$ | Chunk size in chunkwise-parallel computation (Sec. 8). |
| *Token projections* | | |
| $x_t$ | $\mathbb{R}^d$ | Input hidden state at step $t$; in Sec. 4 $x_t/x_k$ instead denotes the SSM latent state $\in \mathbb{R}^N$. |
| $u_t$ | $\mathbb{R}^M$ | SSM input signal (Sec. 4); elsewhere the token input is written $x_t$. |
| $q_t$ | $\mathbb{R}^{d_k}$ | Query vector (read-out address). |
| $k_t$ | $\mathbb{R}^{d_k}$ | Key vector (write address). |
| $v_t$ | $\mathbb{R}^{d_v}$ | Value vector (content written to memory). |
| $o_t, y_t$ | $\mathbb{R}^{d_v}$ | Output of the layer at step $t$ (in the linear-attention convention; in Sec. 5 and Sec. 4 the output $y_t$ may have its own dimension). |
| $Q, K, V$ | $\mathbb{R}^{N \times \cdot}$ | Stacked query / key / value matrices over a sequence. |
| *Memory, gates, and write strength* | | |
| $S_t$ | $\mathbb{R}^{d_v \times d_k}$ | Recurrent state (memory) at step $t$; also written $W_t$ (FWP, Sec. 5), $H_t$, or $M_t$ in individual sections. |
| $h_t$ | $\mathbb{R}^d$ / $\mathbb{R}^N$ | Recurrent hidden state (Secs. 2, 3, 5); SSM latent state $\in \mathbb{R}^N$ in Sec. 4. |
| $A_t$ | $\mathbb{R}^{d_v \times d_v}$ | Left (forget) gate acting on the value side of the state. |
| $B_t$ | $\mathbb{R}^{d_k \times d_k}$ | Right (forget) gate acting on the key side of the state. |
| $\beta_t$ | $\mathbb{R}$ | Write strength: how strongly the input is written to memory. |
| $\alpha_t, \gamma_t$ | $\mathbb{R}$ or $\mathbb{R}^d$ | Decay coefficient(s); scalar (data-independent or data-dependent) or vector-valued. |
| $\lambda_t$ | $\mathbb{R}^d$ | Diagonal / vector forget-gate values (e.g. $\mathrm{diag}(\lambda_t)$). |
| $\eta_t$ | $\mathbb{R}$ | Learning-rate / step size in delta-rule and test-time updates; plays the role of the write strength $\beta_t$ in delta-rule form. |
| $z_t$ | $\mathbb{R}^{d_k}$ | Normalizer state accumulating keys (denominator term). |
| *State-space parameters* (Sec. 4) | | |
| $A, B, C, D$ | — | Continuous-time SSM state / input / output / skip matrices. |
| $\bar{A}, \bar{B}$ | — | Discretized SSM matrices (the read-out matrix $C$ is left undiscretized). |
| $\Delta$ | $\mathbb{R}_{>0}$ | Discretization step size. |
| $\bar{K}$ | $\mathbb{R}^L$ | SSM convolution kernel, $\bar{K}_i = C\bar{A}^i\bar{B}$. |
| *Maps and operators* | | |
| $\phi(\cdot)$ | — | Feature map / kernel function on queries and keys. |
| $\sigma(\cdot)$ | — | Nonlinear activation (e.g. sigmoid, tanh). |

*Table 5 (continued)*

| Symbol | Space / Type | Meaning |
|---|---|---|
| $M$ | $\mathbb{R}^{N \times N}$ | Causal attention mask. |
| $\odot$, $\otimes$, $*$ | — | Hadamard product, outer product, convolution. |
| $\nabla$ | — | Gradient operator (test-time / delta-rule objectives). |
| $\mathcal{L}$, $\ell$ | — | Loss / objective minimized during state update. |

