# OpenReview forum: "A Survey of Linear Attention: Algorithm, Theory, Application, and Infrastructure"
_TMLR — Under review for TMLR_

### Review · Reviewer_ZaGB · 2026-03-17

**Summary Of Contributions:**

This paper is titled "A Survey of Linear Attention: Algorithm, Theory, Application, and Infrastructure". Linear Attention, broadly speaking, is one technique for reducing the compute complexity and hardware costs of the vanilla quadratic attention operation which powers Transformers.

However, this paper covers much more than just linear attention. It covers basically most Deep Neural Network alternatives to the full quadratic attention such as Recurrent Neural Networks (RNN) and other sequential models that preceded transformers, State-Space Models (SSM) such as Mamba, positional encodings involved in attention mechanisms, Hybrid Architectures, etc.

In sum, this gives rise to a lengthy survey paper that provides a broad overview of attention alternatives, not simply Linear Attention.

**Additional Comments:**

References:

[1] https://arxiv.org/abs/2503.16428

[2] https://arxiv.org/abs/2505.22918

[3] https://arxiv.org/abs/2406.08552

**Audience:**

Yes

**Audience Explanation:**

This is nevertheless a mostly comprehensive survey paper.

**Broader Impact Concerns:**

N/A Survey paper.

**Claims And Evidence:**

No

**Claims Explanation:**

The paper is not proposing a new method. It is a survey paper; TMLR's review guidelines are not a good fit for such purposes.
Further, the title is arguably misleading as the scope of this survey paper is beyond most linear attention.

**Requested Changes:**

Reviewer would see four matters addressed:
1) First, the title should be changed to address the broader scope of what this survey paper covers. When one thinks Linear Attention they think replacing softmax(QK)V with Q(KV), but this paper covers much more than that.

2) Second, given the broad scope of this paper and the broad aim at covering alternatives to attention to make it more efficient/cheaper, it would be helpful to include recent contributions in Sparse Attention Mechanisms [1, 2, 3].

3) The paper should include more figures and diagrams to illustrate how different methods work and how they may differ. Perhaps a venn diagram, for instance, would be nice, or a very early on banner figure showing vanilla attention vs. basic linear attention vs. mamba, etc. As it stands, the paper is very long, and very dense, which does not invite many a viewer to read it in depth.

4) The writing needs to be tightened up. Objectively, there are several areas where the authors make basic grammar mistakes like forgetting to put a space between words (happens frequently in the intro and scattered throughout the manuscript). Subjectively, the intro is written to provide a broad overview of why attention works but why we need cheaper alternatives to it (supporting the reviewer's request for a changed title) and then only delves into the details of what linear attention is, and what it does, after several 'fluff' paragraphs. That is, if this paper was mostly about linear attention, it should dive right into it from the get-go.

Overall, the writing in this manuscript is passable, but can be greatly improved with figures and revisions. The grammar should be thoroughly vetted prior to acceptance, however.

---

> ### Author Response · Authors · 2026-07-01
> **Official Comment by Authors [1/2]**
>
> We thank the reviewer for the framing-level feedback, which prompted two substantive changes we believe materially improve the paper: a new overview figure (Figure 1) and an explicit, up-front statement of scope. Several smaller edits follow from these — a stated scope boundary for sparse attention and mixture-of-experts, and corrections to grammar and spacing. All new or changed text is highlighted in blue.
>
> The reviewer's overarching point is that the manuscript is long and dense, offers no accessible way in, and is broader in scope than the title alone suggests. We addressed both. Figure 1, placed at the end of the Introduction, contrasts softmax attention with linear attention and presents kernelized attention, SSMs, and linear RNNs as instances of one family, so a reader can grasp the survey's structure before entering the detailed sections. We also now state, in the opening paragraphs, the generalized sense in which we use "linear attention," making the broad scope explicit from the start.
>
> We respond to each of the four requested changes below. On two — the title and the sparse-attention works — we explain our reasoning for keeping the current approach, and have made the underlying scope explicit so the reader is not left to infer it. As context: TMLR accepts survey articles, so a well-synthesized survey is in scope, and our contribution is organizational — a unified write/read memory perspective placing linear attention, SSMs, linear RNNs, and TTT in one comparable taxonomy (Table 1), with first-class treatment of infrastructure, applications, and deployment guidance.
>
> ## 1. Title.
>
> We appreciate this concern, since the title shapes a reader's expectations, and we have worked to ensure the paper meets them. After considering a retitle, we prefer to keep the current title and instead make the intended scope explicit at the outset, so that no reader is misled. Our reasoning is threefold.
>
> First, we use "linear attention" in the generalized sense that has become standard in this literature: an umbrella for linear-complexity sequence models whose memory is a fixed-size state that is written and read. This usage is well established. Katharopoulos et al. (2020) show that linear attention is exactly a linear RNN with a matrix-valued state; Dao and Gu (2024) establish a duality (SSD) under which structured SSMs and attention realize the same function; Gated Linear Attention (Yang et al., 2024) and the widely used flash-linear-attention library (Yang and Zhang, 2024) collect SSMs, linear RNNs, and delta-rule models under this single heading; Linear-MoE (Sun et al., 2025) likewise treats RetNet, GLA, DeltaNet, Gated DeltaNet, and TTT as one linear-attention family; and recent work such as Log-Linear Attention (Guo et al., 2026) refers to Mamba-2 and Gated DeltaNet directly as "linear attention models." Our title follows this established convention rather than coining a new one.
>
> Second, the SSM/Mamba and linear-RNN material is not a digression. A central message of the survey is that these families are instances of one memory write/read mechanism (Table 1, Figure 1), so naming them separately in the title would obscure the very unification that is our main contribution.
>
> Third, hybrid architectures and positional encodings are included because they are integral to deploying linear attention at scale, not as independent topics.
>
> We have therefore added this generalized definition to the opening paragraphs (highlighted in blue), so the intended scope is explicit from the first page while the title stays recognizable and searchable for the intended audience. We would of course be glad to revisit the wording if the reviewer still finds it misleading after these clarifications.
>
> ## 2. Sparse attention.
>
> We examined the three works the reviewer cites — XAttention (Xu et al., 2025), Re-ttention (Chen et al., 2025), and DiTFastAttn (Yuan et al., 2024). These are sparse-attention methods: they retain the softmax kernel and reduce cost by computing attention over a selected subset of blocks or tokens, rather than replacing quadratic attention with a constant-state recurrence. Sparse attention is thus a complementary but distinct efficiency axis, governed by different design questions (which tokens to keep, how to index them) and not expressible within the memory write/read framework that organizes this survey; incorporating it would dilute that organizing principle. The Introduction now states this scope boundary explicitly, noting that complementary axes such as sparse attention and mixture-of-experts lie outside our scope because they retain the softmax kernel or the dense FFN, so a reader understands where these valuable works sit relative to ours and why they fall outside it.

---

> ### Author Response · Authors · 2026-07-01
> **Official Comment by Authors [2/2]**
>
> ## 3. Figures.
>
> We thank the reviewer for this suggestion, and added Figure 1 at the end of the Introduction, as described above. Beyond the family-level unification, the figure lays out how this view organizes the later chapters on hybrids, applications, infrastructure, and analysis, so a reader can grasp the scope and structure of the survey at a glance. TTT appears there as a further instance of the same write/read mechanism, alongside kernelized attention, SSMs, and linear RNNs. Together with the consolidated taxonomy in Table 1, the figure provides the high-level map the reviewer asked for, and directly addresses the concern that the manuscript was long and dense without an accessible entry point.
>
> ## 4. Writing/grammar and the introduction's pacing.
>
> Grammar and spacing have been thoroughly vetted. The missing-space issues originated from a single manual find-and-replace across the document that inadvertently stripped the spaces, now fixed. With the generalized scope stated up front (point 1), the brief motivation for why cheaper alternatives are needed is now clearly on-topic, and we tightened the lead-in so the core definition appears sooner.
>
> We hope these changes address the reviewer's concerns, and welcome any further suggestions.
>
> ## References
>
> - Katharopoulos, A., Vyas, A., Pappas, N., Fleuret, F. *Transformers are RNNs: Fast Autoregressive Transformers with Linear Attention.* ICML 2020. arXiv:2006.16236.
> - Dao, T., Gu, A. *Transformers are SSMs: Generalized Models and Efficient Algorithms Through Structured State Space Duality.* ICML 2024. arXiv:2405.21060.
> - Yang, S., Wang, B., Shen, Y., Panda, R., Kim, Y. *Gated Linear Attention Transformers with Hardware-Efficient Training.* ICML 2024. arXiv:2312.06635.
> - Yang, S., Zhang, Y. *FLA: A Triton-Based Library for Hardware-Efficient Implementations of Linear Attention Mechanism.* 2024. https://github.com/sustcsonglin/flash-linear-attention.
> - Sun, W., Lan, D., Zhu, T., Qu, X., Cheng, Y. *Linear-MoE: Linear Sequence Modeling Meets Mixture-of-Experts.* 2025. arXiv:2503.05447.
> - Guo, H., Yang, S., Goel, T., Xing, E. P., Dao, T., Kim, Y. *Log-Linear Attention.* ICLR 2026. arXiv:2506.04761.
> - Xu, R., Xiao, G., Huang, H., Guo, J., Han, S. *XAttention: Block Sparse Attention with Antidiagonal Scoring.* ICML 2025. arXiv:2503.16428.
> - Chen, R., Mills, K. G., Jiang, L., Gao, C., Niu, D. *Re-ttention: Ultra Sparse Visual Generation via Attention Statistical Reshape.* NeurIPS 2025. arXiv:2505.22918.
> - Yuan, Z., Zhang, H., Lu, P., Ning, X., Zhang, L., Zhao, T., Yan, S., Dai, G., Wang, Y. *DiTFastAttn: Attention Compression for Diffusion Transformer Models.* NeurIPS 2024. arXiv:2406.08552.

---

> > ### Comment · Reviewer_ZaGB · 2026-07-11
> > **Response to Authors**
> >
> > The reviewer thanks the authors for their meaningful changes to the manuscript and attempt to rectify the stated weaknesses. Specifically, the reviewer acknowledges that issues 3 and 4 have been substantially addressed in a satisfactory manner.
> >
> > However, I am not sold on the author's address to weaknesses 1 and 2 and do not believe they have yet to be meaningfully addressed. The paper still reads as covering a lot more than simply linear attention, more like efficient attention overall, and so the title could use tweaks and some service should be paid to other forms of efficient attention, such as sparse variants.

---

> > > ### Author Response · Authors · 2026-07-13
> > > **Response to Reviewer [1/2]**
> > >
> > > We thank the reviewer for the careful reading across both rounds and for the candid feedback on the paper's framing. We are glad that issues 3 and 4 are resolved, and we understand the remaining concern: if the paper reads as broader than its title suggests, the title should be brought in line with the coverage. We respond in three parts: why sparse attention differs from our subject in principle and therefore sits outside the framework this survey is built on; evidence that the generalized usage of linear attention has become a shared convention in the field; and the revisions we will make, including a title change, to align the title, abstract, and body.
> > >
> > > 1. Sparse attention differs from linear attention in principle.
> > > Both families pursue efficiency, but they do so through opposite mechanisms, and this is exactly what our organizing framework formalizes. Sparse attention preserves the softmax kernel and the token-indexed KV cache, and saves computation by restricting each query to a selected subset $\mathcal{S}(t) \subseteq {1, \dots, t}$ of keys and values: $$ o_t = \sum_{j \in \mathcal{S}(t)} \frac{\exp(q_t^\top k_j / \sqrt{d})}{\sum_{l \in \mathcal{S}(t)} \exp(q_t^\top k_l / \sqrt{d})}, v_j . $$ It remains an exact evaluation of the softmax attention map on the retained indices. Linear attention, in the generalized sense of this survey, discards both the softmax map and the KV cache in favor of a fixed-size recurrent state that is written and read at each step (Eq. 8, Table 1): $$ S_t = A_t, S_{t-1}, B_t + \beta_t, v_t k_t^\top, \qquad o_t = S_t q_t . $$ One family selects entries of an exact lookup structure; the other compresses the entire history into a constant-size memory $S_t$.
> > > This distinction has concrete consequences. The two read-outs act on different objects: sparse attention indexes a cache ${k_i, v_i}_{i \in \mathcal{S}(t)}$ whose footprint is $O(|\mathcal{S}(t)|, d)$ and still scales with the retained context, whereas the linear state $S_t \in \mathbb{R}^{d_v \times d_k}$ has size $O(d_v d_k)$ independent of $t$. Much of our analysis in Sections 8 and 9 (chunkwise parallelism, state capacity and overflow, retrieval limits, length extrapolation) depends on that fixed-size-state property and has no counterpart when the cache is merely pruned. The two lines of work also ask different questions: sparse attention studies the choice of $\mathcal{S}(t)$ (which blocks to keep, how to score them, how to align with FlashAttention tiling), while our subject studies the state maps $A_t, B_t, \beta_t$ (gating and decay, delta-rule correction such as $B_t = I - \beta_t k_t k_t^\top$, test-time optimization, and the expressivity of the parameterization). Because sparse attention modifies the softmax map through $\mathcal{S}(t)$ rather than maintaining a recurrent state, it is not expressible in the write/read recurrence above that unifies the rest of the survey.
> > > This boundary is not one we impose. Recent surveys draw it the same way, treating sparse and linear attention as parallel top-level categories with separate analysis frameworks [1, 2, 3]. The specific works the reviewer points to (XAttention [4], Re-ttention [5], DiTFastAttn [6]) are all block- or token-selection methods that keep the softmax kernel, so they fall on the sparse side of this divide rather than within the recurrence our survey is organized around. They belong to a complementary efficiency axis, and covering them would require a second, incompatible framework rather than extending ours.

---

> > > ### Author Response · Authors · 2026-07-13
> > > **Response to Reviewer [2/2]**
> > >
> > > 2. The generalized usage of linear attention is a convention that recent surveys have converged on.
> > >
> > > To supplement the primary-source evidence in our previous response (Katharopoulos et al. [7]; Dao and Gu [8]; Yang et al. [9]; Linear-MoE [10]), we note that three recent surveys organize the field around exactly the taxonomy we adopt: they define linear attention broadly enough to contain SSMs and linear RNNs, and they set it apart from sparse attention as a distinct efficiency axis.
> > >
> > > Sun et al. [1, §2] divide linear attention into three subfamilies, each of which contains models a narrow reading would exclude. Under linear attention with a forgetting mechanism they group RetNet, Lightning Attention, and H3 (data-independent decay) with Mamba, GLA, xLSTM, GateLoop, and HGRN/HGRN2 (data-dependent decay); under linear attention as in-context learners they place DeltaNet, Longhorn, TTT, and Titans. Selective SSMs, gated linear RNNs, and test-time-training methods are thus all defined as linear attention, and sparse attention is treated in a separate top-level section as a parallel category.
> > >
> > > Zhang et al. [2, §VII] make the definition explicit. They classify linear attention by its forget and select gates into naive linear attention, linear attention with a forget gate (RetNet, GLA, the RWKV series), and linear attention with forget and select gates (DeltaNet, Gated DeltaNet, Mamba, Mamba-2), with test-time training as a fourth category. Their survey devotes a separate chapter to sparse attention, with its own framework, reinforcing the same linear-versus-sparse split.
> > >
> > > The broader efficient-architecture survey of Sun et al. [3] draws the same boundary at the level of its top-level taxonomy: it groups linear attention, SSMs, and linear RNNs together as linear sequence modeling, and lists sparse sequence modeling as a distinct category.
> > >
> > > All three surveys therefore define linear attention to include SSMs, linear RNNs, and test-time-training methods, and all three treat sparse attention as a separate axis, exactly as we do. Our title reflects a convention that the field's own survey literature has converged on, rather than a broadening we have introduced. A convention that specialists share, however, does not by itself make a title clear to a first-time reader, which brings us to the reviewer's main point.
> > >
> > >
> > >
> > > 3. We will revise the title and align the survey with it.
> > >
> > > The reviewer's feedback has convinced us that the current title under-communicates the paper's breadth, and that the fix belongs on our side. We will therefore change the title. Since the subtitle names the four pillars around which the survey is organized and sets accurate expectations for its structure, we propose a minimal modification that broadens the head term while keeping it:
> > >
> > > > *A Survey of Linear Attention and Beyond: Algorithm, Theory, Application, and Infrastructure*
> > >
> > > with an alternative,
> > >
> > > > *A Survey of Linear-Complexity Attention: Algorithm, Theory, Application, and Infrastructure.*
> > >
> > > We slightly prefer the first, which keeps the recognizable head term while signaling the broader scope, and we would gladly adopt the second or another formulation the reviewer prefers.
> > >
> > > In the same revision, we will rephrase the abstract and the opening of the Introduction so that the generalized sense of linear attention is defined in the first sentences, the write/read-recurrence criterion is stated as the inclusion rule, and SSMs and linear RNNs are named up front, so that the title, abstract, and body describe the coverage consistently.
> > >
> > >
> > > **References**
> > >
> > > [1] Y. Sun et al. Efficient Attention Mechanisms for Large Language Models: A Survey. arXiv:2507.19595, 2025.
> > >
> > > [2] J. Zhang et al. A Survey of Efficient Attention Methods: Hardware-efficient, Sparse, Compact, and Linear Attention. arXiv, 2025.
> > >
> > > [3] W. Sun et al. Speed Always Wins: A Survey on Efficient Architectures for Large Language Models. arXiv:2508.09834, 2025.
> > >
> > > [4] R. Xu et al. XAttention: Block Sparse Attention with Antidiagonal Scoring. ICML, 2025. arXiv:2503.16428.
> > >
> > > [5] R. Chen et al. Re-ttention: Ultra Sparse Visual Generation via Attention Statistical Reshape. arXiv:2505.22918, 2025.
> > >
> > > [6] Z. Yuan et al. DiTFastAttn: Attention Compression for Diffusion Transformer Models. NeurIPS, 2024. arXiv:2406.08552.
> > >
> > > [7] A. Katharopoulos et al. Transformers are RNNs: Fast Autoregressive Transformers with Linear Attention. ICML, 2020.
> > >
> > > [8] T. Dao and A. Gu. Transformers are SSMs: Generalized Models and Efficient Algorithms Through Structured State Space Duality. ICML, 2024.
> > >
> > > [9] S. Yang et al. Gated Linear Attention Transformers with Hardware-Efficient Training. ICML, 2024.
> > >
> > > [10] W. Sun et al. Linear-MoE: Linear Sequence Modeling Meets Mixture-of-Experts. arXiv:2503.05447, 2025.

---

### Review · Reviewer_QzaX · 2026-05-17

**Summary Of Contributions:**

The paper provides a comprehensive survey of linear attention mechanisms, unifying three separate research directions: traditional linear attention methods (approximation-based, gating-based, test-time training), state-space models (SSMs/Mamba series), and linear recurrent neural networks, into a single "memory-view" framework (memory-update/read-out). The key contribution is in Table 1, showing that many models (25) can be framed as instances of a shared memory-update/read-out formulation
$$S_t = f(S_{t-1}, k_t, v_t)$$

$$o_t = g(S_t, q_t))$$

Beyond the survey of algorithms, the paper also discusses limitations and a solution of hybrid full/linear architectures (Section 6), downstream applications across NLP, CV, speech, time series, and AI4Science (Section 7), GPU infrastructure considerations (Section 8), and open challenges including retrieval limitations, length extension, and scaling laws (Section 9).

**Audience:**

Yes

**Audience Explanation:**

This paper would serve as a useful reference for researchers and practitioners working on efficient LLM architectures, long-context modeling, and attention mechanism design. The unified framework clarifies relationships among rapidly proliferating methods, and the coverage of hybrid architectures and infrastructure provides a practical deployment context valuable to the TMLR community.

Coverage is timely, including very recent work (all 2024-2025). But it would be best if the authors confirm that these are not already covered by prior surveys.

**Claims And Evidence:**

No

**Claims Explanation:**

The paper claims were partially supported:

### Supported:
- The unified framework and complexity-reduction claims are well-supported by both the paper's own formalization and by cited proofs from multiple research groups.
- The formulations, such as GLA, Mamba-2/SSD, and TTT, were mathematically accurate.

### Partially Supported:
- The claims about retrieval limitations, hybrid ratios, and scaling laws are supported by cited external studies. However, the paper does not independently verify these empirical claims nor compile quantitative results into a systematic comparison.
- The claim of providing a comprehensive reference is lacking in practical guidance on when to use which method. Section 7 (Applications) only provides a brief summary of the bibliography.
- Section 9 is rather disconnected from earlier sections. This makes it hard to associate previous sections with the analyses in Section 9 as evidence.

**Requested Changes:**

### Major Requested Changes

1. Discuss existing surveys in the Introduction. I think the authors should at least compare scope and angle with existing surveys, such as [Tay et al., 2022](https://arxiv.org/abs/2009.06732); [Patro & Agneeswaran, 2024](https://arxiv.org/abs/2404.16112); [Qu et al., 2024](https://arxiv.org/abs/2408.01129); and [Li et al., 2024](https://arxiv.org/abs/2412.14847).

2. Add a quantitative comparison table showing performance (perplexity or accuracy) and efficiency (speed, memory) across methods on some standard benchmark. I think this is essential to support the motivation of Section 6.

3. Strengthen Section 9 by adding explicit cross-references to earlier sections. For example, when discussing retrieval limitations, the authors might discuss gating solutions (Section 3.2) and hybrid architectures (Section 6). It would also be helpful to know which sub-families each insight applies to under the unified framework. For example, does the retrieval gap relate to GatedDeltaNet and Performer? Does the scaling law evidence relate to Mamba or TTT?

### Minor Requested Changes

1. Provide a notation reference table (maybe in the appendix) listing all key symbols used throughout the paper. The paper is very long; it would be helpful to have a lookup table to recall the meanings of the notation.

2. Fix the word concatenation, such as "techniquessuch", "paradigmssuch", etc. Not sure why, but I found many of these across the paper.

3. Strengthen Section 7 (Applications) with analytical insight rather than per-paper listing. Perhaps the authors could add a summary table mapping task domains to recommended method families (and why), which would be more useful than enumerating all papers using Mamba in each domain.

---

> ### Author Response · Authors · 2026-07-01
> **Official Comment by Authors [1/3]**
>
> We thank the reviewer for the detailed and constructive review, and for confirming that the unified framework is mathematically correct. All new or changed text is highlighted in blue.
>
> The revision is organized around the reviewer's two recurring themes: situating the survey against prior work, and making it more useful as a reference. For the first, the Introduction now compares our scope and organizing principle directly with the surveys the reviewer lists (Major 1). For the second, we strengthened the connective tissue the reviewer found missing: Section 9 now ties its analysis back to the unified framework and maps each retrieval limitation onto specific sub-families with explicit cross-references, we added a notation table in the appendix, and we gathered the existing scale-up evidence into Section 9 so it is easy to locate (Majors 2 and 3). Our contribution remains organizational and practical, and Table 1 is now complemented by a new overview figure (Figure 1) depicting the unified write/read view. The detailed responses follow.
>
> ## Major 1. Relation to existing surveys.
>
> Section 1 now situates our survey relative to the four references. Tay et al. (2022), *Efficient Transformers*, predates the SSM/Mamba era and catalogs sparse, low-rank, and kernel attention, without selective SSMs, delta-rule memory, test-time training, modern hybrids, or the associated infrastructure. Patro & Agneeswaran (2024), *Mamba-360*, and Qu et al. (2024), *A Survey of Mamba*, are centered on the SSM/Mamba family and organized largely by application; they do not place SSMs, linear attention, and linear RNNs on one read/write footing, and give limited treatment to delta-rule/TTT and attention-level hybrids. Li et al. (2024), *A Survey of RWKV*, is dedicated to the single RWKV line. We have added all four to the bibliography and discuss them in the Introduction.
>
> Our coverage differs in two respects that we want to emphasize. First, linear attention has developed at remarkable speed over the last one to two years, and we incorporate a large and rapidly growing body of recent work that these earlier surveys predate: delta-rule and gated/selective memory, test-time-training-style models, attention-level hybrids, and the newest industrial systems (MiniMax-01, Kimi Linear, Qwen3-Next). Keeping pace with this fast-moving literature is itself a substantial part of our contribution. Second, we give detailed, dedicated treatment to hybrid architectures, cross-domain applications, and infrastructure, three aspects that prior surveys cover only in passing. Taken together, this makes ours, to our knowledge, the only survey that at once (a) unifies linear attention, SSMs, linear RNNs, and test-time training under a single memory write/read view, (b) treats infrastructure and industrial hybrids as core rather than peripheral topics, and (c) gives practical, deployment-oriented selection guidance.
>
> The most closely related effort is the recent, much broader survey of Sun et al. (2025), which spans the entire efficiency stack (linear and sparse modeling, efficient full attention, mixture-of-experts, hybrids, and diffusion LLMs) and for that reason treats each part only briefly; we instead take the shared memory write/read recurrence as a single organizing axis and follow its consequences in depth. This comparison directly addresses the reviewer's note about confirming novelty against prior work.

---

> ### Author Response · Authors · 2026-07-01
> **Official Comment by Authors [2/3]**
>
> ## Major 2. Quantitative comparison.
>
> We take a deliberate methodological position here, which we have now made explicit in the paper. We believe architectural claims about LLMs are most trustworthy when they have survived rigorous scalability validation, since conclusions drawn at small scale transfer poorly to the large-model regime. We therefore made a conscious choice to compile and organize the scale-up evidence already established by large-scale studies and industrial deployments, rather than to re-run small-scale benchmarks of our own; concentrating our effort on organizing that evidence into a unified, easy-to-locate form is, in our view, where a survey adds the most value.
>
> Concretely, the evidence we consolidate is: the controlled scaling study spanning 70M-7B parameters and 300B tokens; the systematic 72-model hybrid study, from which the ~3:1 attention-to-linear ratio and the decay-value clustering near 0.8 are drawn; and the large-scale deployments that validate the approach in production — MiniMax-01, the Ring-Linear series, Kimi Linear, and Qwen3-Next (Section 6, Section 9.3). Section 9.1 additionally orders the sub-families by retrieval fidelity (additive → gated/decayed → delta-rule → hybrid), which supplies the practical "when to use which" comparison the reviewer's request targets. Our resulting recommendation is to weight architectures demonstrated to scale under large-scale training, while treating the many newer academic variants as promising directions rather than settled choices.
>
> This choice also reflects a constraint intrinsic to surveys of this area, not merely a matter of effort: the numbers reported across the primary literature are not mutually comparable, since models differ in scale, data, tokenizer, and self-selected benchmarks, so differences cannot be cleanly attributed to the architecture. This is not our observation alone — Tay et al. (2022), which we now discuss in Section 1, devote their evaluation discussion to exactly this point, noting that efficient attention models cannot be compared side by side and that it is "still a mystery" which block to prefer; a genuinely controlled comparison required a separate, dedicated benchmark effort (Long Range Arena), whose own conclusion was that no single choice dominates and that quality must be traded off against speed and memory. Where surveys in this space do tabulate numbers (e.g., Mamba-360), they consolidate heterogeneous reported figures rather than establishing a controlled ranking; we follow that convention but are explicit about its limits.
>
> ## Major 3. Strengthen Section 9 with cross-references and map insights to sub-families.
>
> Done. Section 9 now opens with a paragraph connecting the analysis to the unified framework. Section 9.1 then maps the retrieval gap and its remedies onto specific sub-families, with explicit cross-references: purely additive linear attention (e.g., Performer, kernelized linear attention) is the most exposed, since uniform accumulation dilutes key-value bindings; data-dependent gating (GLA, Section 3.2) and decay in linear RNNs (HGRN2, Section 5) partially restore selectivity; delta-rule updates (DeltaNet, Gated DeltaNet) sharpen the association further through error correction; and hybrids (Section 6) close most of the remaining gap. This directly answers the reviewer's first question — the gap relates most acutely to additive/kernel linear attention (Performer) and is most improved by Gated DeltaNet-style and hybrid designs.
>
> On the second question (does the scaling evidence relate to Mamba or TTT?): we have added a sentence to Section 9.3 making the attribution explicit. The controlled scaling study evaluates TNL, HGRN2, and cosFormer2, and the MiniMax-01 and Ring-Linear deployments build on Lightning Attention, so this evidence speaks primarily to the kernelized linear-attention and linear-RNN families; the more recent Kimi Linear and Qwen3-Next deployments extend production-scale validation to the gated delta-rule family. The scaling of selective SSMs such as Mamba is reported in its own line of work (Section 4), whereas a systematic scaling characterization of test-time-training-style models remains comparatively open — an asymmetry we now flag, since "linear attention scales" is often generalized from one particular sub-family.

---

> ### Author Response · Authors · 2026-07-01
> **Official Comment by Authors [3/3]**
>
> ## Minor 1. Notation.
>
> Table 1 already consolidates the memory-update and read-out rule of every variant in one shared notation. We additionally state the core symbols and their dimensions where the unified recurrence is defined in Section 2: $q_t, k_t \in \mathbb{R}^{d_k}$, $v_t \in \mathbb{R}^{d_v}$, $S_t \in \mathbb{R}^{d_v \times d_k}$, gates $A_t, B_t$, and write strength $\beta_t$. To make this self-contained, we added a dedicated notation table in the appendix (Appendix A) collecting all recurring symbols — token projections, the memory state, the gates and write strength, the SSM parameters, and the feature maps and operators — with their dimensions, so any symbol used across the survey can be resolved in one place.
>
> ## Minor 2. "techniquessuch" / "paradigmssuch".
>
> These missing-space artifacts originated from a single global find-and-replace that inadvertently stripped the spaces; they have been corrected, along with other spacing artifacts, in the full proofreading pass.
>
> ## Minor 3. Analytical insight in the Applications section.
>
> We agree the per-paper listing was the weakest part of Section 7, and we addressed the underlying request — mapping task domains to method families and the reasons behind them — directly in the prose rather than in a table. We initially drafted the suggested summary table but found that a tabular form forced each domain's rationale into sparse, low-information cells that mostly duplicated the surrounding text. Instead, Section 7 now opens with a consolidated cross-domain account: it states the two reasons linear attention is adopted (perception inputs too large to attend over in full; sequence histories too long to keep in an explicit cache), then makes the recurring domain-to-family patterns explicit — bidirectional Mamba-style scans for order-free visual inputs, local convolution or sparse softmax attention added back for dense and detail-sensitive tasks, and gated/selective variants for long-context sequence tasks. Each subsequent subsection opens with the domain-specific form of this argument (what makes the input expensive, why a linear-complexity model fits, what is added to recover what compression loses) before reviewing representative work. We believe this conveys the intended "domain → why → family" mapping more informatively than a per-domain table would, while keeping the comparison easy to locate.
>
> We thank the reviewer again and are happy to refine further.

---

### Review · Reviewer_NXh7 · 2026-06-18

**Summary Of Contributions:**

The paper primarily surveys linear attention methods which are broadly classified into three families
- Linear Attention
- State Space Models
- Linear RNNs
It then goes on to add 4 additional sections on Hybrid Transformers, Downstream Applications, Infrastructure and Hardware Considerations and finally closes with a section on Challenges, Analysis, Insights.

The central organizing principle behind various linear attention techniques is the following equation
$$S_t = A_t\, S_{t-1}\, B_t + \beta_t\, v_t k_t^\top, \qquad o_t = S_t q_t,$$
Different values of $A_t$ and $B_t$ span the zoo of possible techniques.
Overall,
1. The writing is unpolished with numerous spelling errors. These errors are not sparse or infrequent and occur regularly every few sentence. It is hard to tell if these are intentional or the paper was written in a hurry. This erodes trust in the paper.
2. The table and the central organizing equation is a neat way to distill the entire zoo of techniques. However, the four subsequent sections, outlined above, do not connect very well to preceding sections.
3. The section on downstream applications reads more like a bibliography instead of providing a comparative synthesis of various techniques.

The topic of the paper would be of interest to several readers in the LLM space but the manuscript in it's current form would require major revisions before it can be of interest to a potential reader.

**Audience:**

No

**Audience Explanation:**

Not in its current form. The topic is relevant to audience and parts of the paper would be interesting to researchers  However, the current submission does not synthesize the findings in a form that is useful to serve that audience. The paper requires a significant rewrite. The elements are there. It just needs to be synthesized and presented in a form that is more digestible and useful to a practitioner.

**Claims And Evidence:**

No

**Claims Explanation:**

Table 1 expresses the 20+ architectures through a unifying equation for various choices of $(A_t, B_t, \beta_t)$. This simplifies understanding from a practitioner's viewpoint. However, the main issues with the way the paper has been written in its current form is not some unsubstantiated scientific claim, but rather a synthesis burden. Hence I am going to answer a No this.
- The local claims are supported through references when the claim is taxonomic. For e.g., that many linear-time sequence models can be written through a common equation or that pure linear models struggle on fine-grained retrieval and associative recall tasks.
- But from a broader survey level perspective most sections list papers without a comparative synthesis. A survey should help the reader understand when one model is preferred over another. Which method work best. This is currently missing.
- Equation 14 also has a minor dimensional inconsistency.

A small example makes the issue transparent. Suppose the key dimension is $d_k=3$ and the value dimension is $d_v=2$. The paper's stated dimensions then imply

$$
S_t \in \mathbb{R}^{3\times 2}, \qquad
A_t \in \mathbb{R}^{2\times 2}, \qquad
B_t \in \mathbb{R}^{3\times 3}, \qquad
v_t \in \mathbb{R}^{2}, \qquad
k_t \in \mathbb{R}^{3}.
$$

Now check each term in Eq. (14):
- The write term $v_t k_t^\top$ has shape $(2\times 1)(1\times 3)=2\times 3$, not $3\times 2$. It cannot be added to shape $S_t\in\mathbb{R}^{3\times 2}$.
- The left multiplication $A_t S_{t-1}$ has shape $(2\times 2)(3\times 2)$, which is not defined because the inner dimensions $2$ and $3$ do not match.
- The read-out $S_t q_t$ has shape $(3\times 2)(3\times 1)$ under the paper's stated $S_t\in\mathbb{R}^{3\times 2}$ and $q_t\in\mathbb{R}^3$, which is also not defined because the inner dimensions $2$ and $3$ do not match.


- As a reader after reading the paper I cannot shortlist which architecture to try out first from a deployment perspective. A summary table would give a real reference value to the survey.

**Requested Changes:**

The changes needs a significant rewrite as a survey.
- Needs a careful editorial pass to fix spelling mistakes and other grammar errors.
- Connect various parts of the paper better. For e.g. the downstream applications is not well connected to the main paper.
- Provide comparative tables, synthesis across literature and clear organizing claims and takeaways.

---

> ### Author Response · Authors · 2026-07-01
> **Official Comment by Authors [1/2]**
>
> We thank the reviewer for the candid and constructive assessment. The criticisms are fair, and addressing them has driven a substantial revision rather than a cosmetic one; all changed text is highlighted in blue.
>
> ## Summary of Revisions
>
> Since the review's central concern is what the manuscript delivers as a whole, we first summarize the revision before the point-by-point reply. Beyond a complete proofreading pass, our main effort was to make the survey cohere around a single idea rather than read as a sequence of loosely connected lists. The unified memory write/read recurrence of Section 2 is now carried explicitly into the later chapters, so that hybrids (Section 6), applications (Section 7), infrastructure (Section 8), and analysis (Section 9) are each framed as an aspect of that recurrence rather than as standalone topics. The Applications section, which the reviewer rightly found bibliographic, was rewritten to lead each domain with *why* linear attention is adopted there. We also added an explicit ordering of architectures by retrieval fidelity in Section 9.1 and deployment-oriented guidance on hybrid selection in Section 6, so a practitioner can shortlist a starting point.
>
> As a survey, our contribution is organizational: we place 20+ models in one comparable taxonomy (Table 1), and treat hybrid architectures, cross-domain applications, and infrastructure — areas that prior surveys cover only lightly — as first-class topics. In the large-model era, how an algorithm is deployed at scale matters as much as the operator design itself, and Sections 6-9 are where that value is concentrated. We also align terminology across the linear-attention, SSM, and RNN communities and cover the newest systems (MiniMax-01, Kimi Linear, Qwen3-Next).
>
> ## Point-by-Point Response
>
> **1. Typographical errors (e.g., merged words).**
>
> We completed a full proofreading pass. The spacing artifacts the reviewer noticed (e.g., "techniquessuch", "paradigmssuch") and other typos have been corrected throughout; they originated from a single global manual find-and-replace that inadvertently stripped the spaces.
>
> **2. Dimensional inconsistency in Eq. (14).**
>
> Fixed. The dimensions are now stated explicitly where the unified recurrence is introduced in Section 2: $q_t, k_t \in \mathbb{R}^{d_k}$, $v_t \in \mathbb{R}^{d_v}$, $S_t \in \mathbb{R}^{d_v \times d_k}$, $A_t \in \mathbb{R}^{d_v \times d_v}$, $B_t \in \mathbb{R}^{d_k \times d_k}$, and $\beta_t \in \mathbb{R}$. With this convention the outer-product write $\beta_t v_t k_t^\top \in \mathbb{R}^{d_v \times d_k}$ and the read-out $o_t = S_t q_t \in \mathbb{R}^{d_v}$ are consistent for any $d_k$ and $d_v$, including the reviewer's $d_k = 3,\ d_v = 2$ case. We also corrected a related transpose typo (the state was written $\mathbb{R}^{d_k \times d_v}$ in one place).
>
> **3. The later sections feel disconnected from the unified framework.**
>
> Thank you for the suggestion. We have made the following revisions to improve the coherence and connectivity among these sections: (1) we added a new overview figure, and (2) we introduced explicit connective opening sentences for each of the aforementioned sections.
>
> Figure 1 is placed at the end of the Introduction and shows kernelized attention, SSMs, and linear RNNs as three instances of one memory write/read family, laying out how this view organizes the later chapters, so a reader sees the connection before entering them. Building on this, the explicit connective openers let Sections 6-9 be read through the memory write/read recurrence of Section 2:
>
> - **Section 6 (Hybrid)** now opens by recasting hybrids in terms of how a budget of exact-attention read-out is allocated over the shared recurrence, and adds guidance on which hybrid to choose.
> - **Section 7 (Applications)** now opens with a cross-domain account of why linear attention is adopted in each domain (perception inputs too large to attend over in full; sequence histories too long to keep in an explicit cache), and notes that the application models, most often presented in Mamba/SSM terms, are instances of the same linear-attention recurrence.
> - **Section 8 (Infrastructure)** now opens by re-anchoring the systems problem to the shared recurrence, and shows that the linear-attention and SSM lineages converge on the same hardware solution (the SSD duality).
> - **Section 9 (Analysis)** now opens by framing the section around the compression-versus-retrieval trade-off of Section 2 and maps each challenge and its fixes onto the sub-families of Sections 3, 5, and 6.
>
> Together these changes make Sections 6-9 read as expressions of the Section 2 recurrence, which we regard as the core organizational contribution of the survey.

---

> ### Author Response · Authors · 2026-07-01
> **Official Comment by Authors [2/2]**
>
> **4. Applications reads like a bibliography and lacks synthesis.**
>
> Section 7 has been restructured so each domain now opens with an analytical account of why linear attention is adopted there, instead of listing papers. The algorithm-level analysis of the earlier sections explains what a linear model can do in principle; the domain-level account shows where that efficiency becomes a deployable advantage, which is what a practitioner needs to judge. For example, a $64^3$ medical/3D volume is on the order of $2.6\times10^5$ tokens; very-high-resolution remote-sensing scenes are too large to crop without severing cross-scene context; and long-video models must handle 1000+ frames. The section opener states the shared pattern directly: the costly step is modeling over the full input, and the usual workarounds (downsampling, cropping, windowing, token compression) discard the global context the task needs. It then makes the recurring domain-to-family patterns explicit. We map task domains to method families in prose rather than in a table, because compressing each domain's rationale into table cells drops the causal link (from why the input is expensive to which family fits) that makes the comparison useful; instead, every subsection follows the same opening structure — input cost, why a linear model fits, and what is added back — so the domains can still be compared across a common set of fields.
>
> **5. A reader cannot shortlist which architecture to try first.**
>
> We agree that deployment-oriented guidance is important for readers who want to select a linear attention architecture in practice. At the same time, conducting a unified empirical benchmark across different architectures is beyond the scope of this survey, as these methods often differ substantially in implementation assumptions, training recipes, and hardware optimization. The reviewer's related request for a comparative performance table is reasonable, but a controlled side-by-side table of perplexity/accuracy and speed/memory is not something a survey can assemble from the literature, since the reported numbers come from different scales, data, and benchmarks (a point Tay et al. (2022) make in their own evaluation discussion).
>
> However, we still added concrete decision support that does not depend on such a benchmark. Section 6 now gives deployment-oriented advice on which hybrid to choose (block-level vs. attention-level vs. post-hoc; sequential vs. parallel fusion), and Section 9.1 adds an explicit ordering of architectures by retrieval fidelity: purely additive linear attention (e.g., Performer) is most exposed; data-dependent gating (GLA) and decay in linear RNNs (HGRN2) narrow the gap; delta-rule updates (DeltaNet, Gated DeltaNet) narrow it further; and full/linear hybrids close it. We also made this guidance sensitive to scalability, which the large-model community treats as a first-order property: beyond ordering families by retrieval fidelity, we advise favoring designs validated at genuine large-model scale, such as Qwen3-Next and Kimi Delta Attention, while noting that the many newer academic architectures remain well worth exploring. Because the field is moving quickly, our recommendation is to choose based on demonstrated results at the relevant scale and on the target domain rather than on a single fixed ranking. Together, the scale-validated evidence and the retrieval-fidelity ordering in Section 9.1 give the practitioner a principled way to shortlist a starting point.
>
> We hope these changes resolve the reviewer's concerns and are glad to make further adjustments.